# Unraveling the electronegativity-dominated intermediate adsorption on high-entropy alloy electrocatalysts

Jiace Hao[1,6], Zechao Zhuang[2,6], Kecheng Cao[3], Guohua Gao[4], Chan Wang[1], Feili Lai[5], Shuanglong Lu[1], Piming Ma[1], Weifu Dong[1], Tianxi Liu[1], Mingliang Du [1] & Han Zhu [1✉]

High-entropy alloys have received considerable attention in the field of catalysis due to their exceptional properties. However, few studies hitherto focus on the origin of their outstanding performance and the accurate identification of active centers. Herein, we report a conceptual and experimental approach to overcome the limitations of single-element catalysts by designing a FeCoNiXRu (X: Cu, Cr, and Mn) High-entropy alloys system with various active sites that have different adsorption capacities for multiple intermediates. The electronegativity differences between mixed elements in HEA induce significant charge redistribution and create highly active Co and Ru sites with optimized energy barriers for simultaneously stabilizing OH* and H* intermediates, which greatly enhances the efficiency of water dissociation in alkaline conditions. This work provides an in-depth understanding of the interactions between specific active sites and intermediates, which opens up a fascinating direction for breaking scaling relation issues for multistep reactions.

---

[1] Key Laboratory of Synthetic and Biological Colloids, Ministry of Education, School of Chemical and Material Engineering, Jiangnan University, Wuxi 214122, China. [2] Department of Chemistry, Tsinghua University, Beijing 100084, China. [3] School of Physical Science and Technology, ShanghaiTech University, Shanghai 201210, China. [4] Shanghai Key Laboratory of Special Artificial Microstructure Materials and Technology, Key Laboratory of Road and Traffic Engineering of the Ministry of Education, Tongji University, Shanghai 200092, China. [5] Department of Chemistry, KU Leuven, Celestijnenlaan 200F, Leuven 3001, Belgium. [6] These authors contributed equally: Jiace Hao, Zechao Zhuang. ✉email: zhysw@jiangnan.edu.cn

Electrocatalytic reactions hold the key to the efficiency of many energy conversion and storage systems, which highly require effective and stable electrocatalysts to tune the rates (selectivity) of reaction pathways during catalytic process. A great challenge facing materials scientists arises from the ongoing need for advanced electrocatalysts with reasonable activities that meet the needs of rapid development[1–3]. Recently, high-entropy alloy (HEA) catalysts with near-equiatomic proportions, have attracted wide attention due to their unprecedented properties caused by high configuration entropy, lattice distortion, sluggish diffusion, and cocktail effects[4–8]. A series of nanoscale HEA catalysts have been applied for ammonia oxidation, water splitting, methanol oxidation, $CO_2$ reduction, and dye degradation, and they deliver superior performance than conventional alloys[7–13]. Infinite elemental combinations and unconventional compositions offer many possibilities for regulating catalytic performance and overcoming the limitations of single-element catalysts, making HEAs an intriguing option in the field of electrocatalysis[11].

The catalytic activities of monometallic catalysts depend substantially on the adsorption of reactants on surface-active sites[14]. Scaling relation issues applicable to many multistep reactions require the stabilization of multiple intermediates during the reaction, but the limited number of active sites with monometallic catalysts cannot achieve simultaneous stabilization of all intermediates on the available active sites with optimal binding energies[15,16]. Nano-catalysts have been rapidly developed in recent years with effectively explored strategies (including alloying, nanostructuring, defect, heterostructure, and lattice strain) to reveal the active sites and cooperation with each other (Fig. 1a)[17–23]. Single atom catalysts (SACs) have attracted tremendous effort to exploit effective routes with good control over coordination environment, composition (dimer), metal loading, and substrate, which are needed for the discovery of the correlations between compositional engineering and optimization of catalytic behavior (Fig. 1b)[24]. Numerous studies have been devoted to focusing on active sites and mechanisms of nano catalysts and SACs, and however, as an equally important catalyst system, accurate identification of active centers and their activity origins in HEA catalysts are greatly neglected. Furthermore, the relationships between multiple active sites in HEAs and reaction intermediates are still obscure with the lack of guidelines for designing reasonable active sites.

Herein we propose a conceptual and experimental approach to overcome the limitations of single-element catalysts by designing a FeCoNiXRu (X: Cu, Cr, and Mn) HEA system with two kinds of active sites that have different adsorption capacities for multiple intermediates. HEA NPs are synthesized in electrospun carbon nanofibers (CNFs) and displayed a thermodynamically driven phase transition, as revealed by in situ characterization. We employ the HEAs as electrocatalysts for the alkaline hydrogen evolution reaction (HER) and successfully identify electronegativity-dependent preferences for active site adsorption of the intermediates OH* and H* during $H_2O$ dissociation and $H_2$ production steps (Fig. 1c). In the FeCoNiMnRu NP, the Co sites are the most active centers with the lowest energy barrier (0.34 eV) for water dissociation. During the

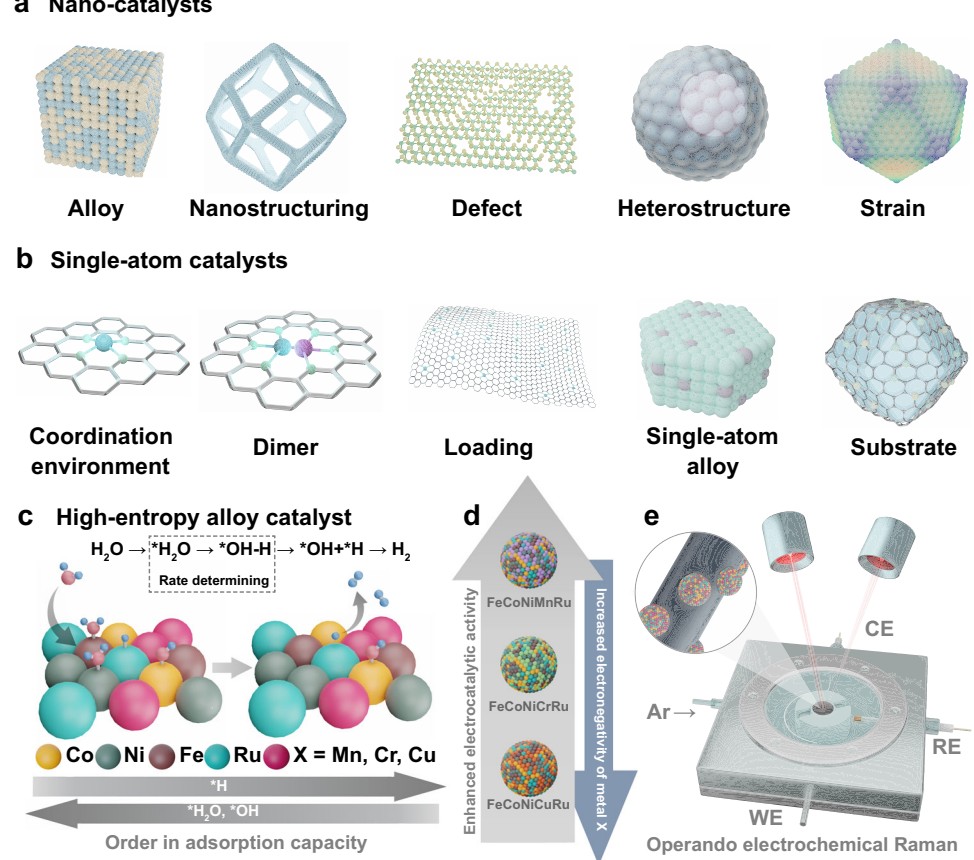

**Fig. 1 Illustration of the concepts for designing nano catalysts, SACs and HEA catalysts.** Designed strategies for creating active sites in (**a**) nano catalysts and (**b**) SACs. **c** Schematic illustration of HEA electrocatalysts with identified electronegativity-dependent preferences for active site adsorption of the intermediates OH* and H* during $H_2O$ dissociation and $H_2$ production steps. **d** Adjustments of the HER activities of HEA electrocatalysts by tailoring the electronegativity of the composition. **e** Identifying active sites of HEA for stabilization of intermediates by operando electrochemical Raman spectra.

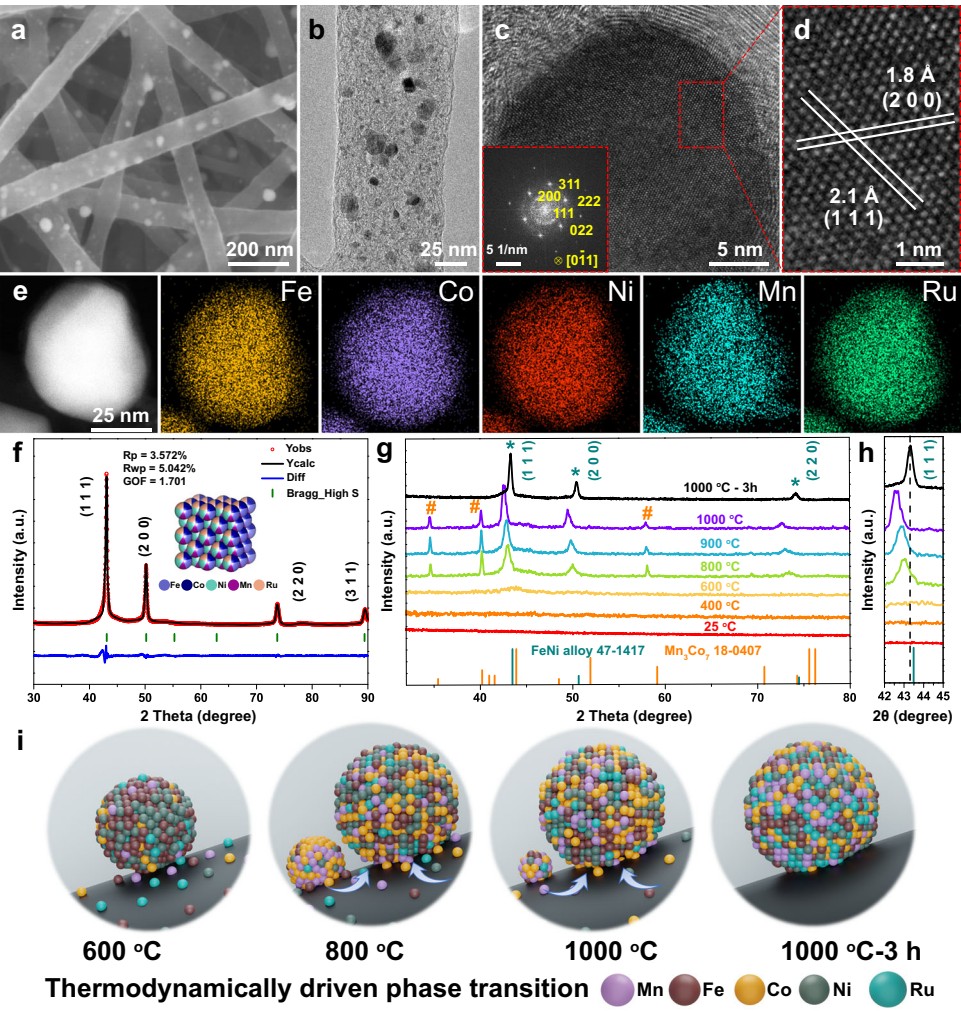

**Fig. 2 Morphological and structural characterization of FeCoNiMnRu/CNFs. a** FE-SEM, **b** TEM and **c, d** HRTEM images of FeCoNiMnRu/CNFs. The inset in (**c**) is the corresponding FFT pattern of a FeCoNiMnRu HEA NP. **e** HAADF-STEM and the corresponding STEM-EDX mapping images of a FeCoNiMnRu HEA NP supported on CNFs. **f** XRD patterns of FeCoNiMnRu/CNFs with detailed Rietveld refinements. **g** Real-time in situ XRD patterns for FeCoNiMnRu/CNFs with temperatures ranging from 25 to 1000 °C. **h** The corresponding enlarged in situ XRD patterns of FeCoNiMnRu/CNFs. The heating rate was kept at 30 °C min$^{-1}$ during the whole process. **i** The thermodynamically driven phase transition of FeCoNiMnRu NPs in CNF nanofiber reactors.

subsequent H$^*$ adsorption/desorption process, H$^*$ leaves Co and is absorbed on Ru sites, which exhibit the lowest $\Delta G_{H^*}$ of $-0.07$ eV. Adjustments of the HER activities of HEA catalysts are shown experimentally and theoretically by tailoring the electronegativity of the composition (Fig. 1d and e). Constructing controllable multifunctional active sites on HEA surfaces allows for different interactions with related intermediates, which is a powerful option for solving this scaling relation challenge.

## Results

**Synthesis and materials characterization.** The model electrocatalysts, single-phase FeCoNiXRu solid solution HEA NPs (X = Cr, Mn, and Cu), were synthesized in carbon nanofibers (FeCoNiXRu/CNFs) through a polymer nanofiber reactor strategy by combining the electrospinning technology and graphitization process. The schematic illustration of the synthetic procedure for FeCoNiXRu/CNFs (X = Cr, Mn, and Cu) are shown in Supplementary Fig. 1 and the details are described in the Experimental section.

As revealed by the field emission scanning electron microscopy (FE-SEM) image in Fig. 2a, large amounts of FeCoNiMnRu NPs

were densely and uniformly anchored in CNFs, and intertwined CNFs with diameters ranging from 100 to 200 nm exhibited porous three-dimensional (3D) networks. The transmission electron microscopy (TEM) image (Fig. 2b) also displays the uniform distribution of FeCoNiMnRu NPs with an average diameter of approximately $14.2 \pm 9.1$ nm (Supplementary Fig. 2). Figure 2c and d display distinctly visible lattice fringes with interplanar crystal spacings of 2.1 and 1.8 Å, corresponding to the (111) and (200) facets. All interplanar spacings were determined by measuring the total distances of 20 successive corresponding planes (Supplementary Fig. 3). In addition, the fast Fourier transform (FFT) pattern (inset in Fig. 2c) further reveals the face-centered cubic (*fcc*) crystal structures of FeCoNiMnRu HEA NPs and exhibits the presence of typical (111), (200), and (220) planes. High-angle annular dark-field STEM (HAADF-STEM) and STEM energy dispersive X-ray (STEM-EDX) elemental mapping images (Fig. 2e) show the homogeneous distribution of elements Fe, Co, Ni, Mn, and Ru in a single FeCoNiMnRu HEA NP. Additionally, the line-scan EDX spectra (Supplementary Fig. 4) also reveal the distribution of Mn, Fe, Co, Ni, and Ru elements throughout the whole HEA NP, further demonstrating the formation of homogeneous structures in FeCoNiMnRu HEA.

The mapping area contains 10 HEA NPs (Supplementary Fig. 5) and the line-scan STEM-EDX of 3 HEA NPs (Supplementary Fig. 6) have also been conducted. The results exhibit the uniform distribution of Fe, Co, Ni, Mn, and Ru elements among all the HEA NPs, further suggesting the repeatability of HEA NPs with uniform composition distribution. The content of each element in FeCoNiMnRu/CNFs was measured by inductively coupled plasma-optical emission spectrometry (ICP-OES). The composition of HEA was calculated to be $Fe_{0.23}Co_{0.22}Ni_{0.22}Mn_{0.14}Ru_{0.19}$ (Supplementary Table 1). A calculated mixing entropy of $\Delta S > 1.59$ R was determined from ICP-OES results, suggesting the intrinsic nature of HEAs without phase separation. The FeCoNiXRu/CNFs (X = Cr and Cu) (Supplementary Fig. 7 and Supplementary Tables 2, 3) with similar Ru contents and control samples of FeCoNi/CNFs, FeCoNiMn/CNFs, and FeCoNiRu/CNFs (Supplementary Fig. 8) were prepared using the same approach and they all exhibited morphologies similar to that of FeCoNiMnRu/CNFs.

The crystalline structures of FeCoNiMnRu/CNFs (Fig. 2f) prepared at 1000 °C under 3 h treatment and the corresponding control samples (Supplementary Fig. 9) were investigated with X-ray diffraction (XRD) patterns. As shown in Fig. 2f, the FeCoNiMnRu/CNFs exhibits three main diffraction peaks at $2\theta = 43°, 50°$ and 74°, which can be indexed to the (111), (200), and (220) planes of the *fcc* phases (PDF#47-1417), respectively. No separated XRD peaks from Fe, Co, Ni, Mn, Ru, or metal oxides were observed, suggesting the formation of a single-phase HEA. In addition, the corresponding XRD detailed Rietveld refinements also confirm the single phase HEA structure of FeCoNiMnRu NPs. Compared with the standard line patterns of the FeNi alloy (PDF#47-1417), the *fcc* diffraction peaks of the FeCoNiMnRu HEA shifted slightly to lower angles due to the lattice distortions caused by the incorporation of Ru, Mn and Fe atoms and the resulted high entropy[4].

All of the XRD patterns of FeCoNi/CNFs, FeCoNiMn/CNFs and FeCoNiRu/CNFs (Supplementary Fig. 9) exhibit the characteristic peaks for the (111), (200), and (220) planes of the *fcc* phase, suggesting that the *fcc* crystal structures can be well maintained after changing in the numbers of elements. Furthermore, with the incorporation of Mn and Ru atoms along with Fe, Co and Ni atoms, the peak positions of the (111) planes for FeCoNiMn/CNFs, FeCoNiRu/CNFs, and FeCoNiMnRu/CNFs gradually move to lower angles, suggesting a strong high −entropy effects[6]. As shown in Supplementary Fig. 10, the XRD patterns of FeCoNiXRu/CNFs (X = Cr, Mn and Cu) confirm the *fcc* structures, and the diffraction peaks exhibit slight differences caused by the different compositions.

The temperature-dependent in situ XRD patterns are illustrated in Fig. 2g with temperatures range from 25 to 1000 °C. As shown in Fig. 2g, only taenite (FeNi alloy, PDF#47−1417) is observed after treated after 600 °C, with (111) planes at $2\theta = 43.8°$. The FeCoNiMnRu HEA *fcc* phase and $Mn_3Co_7$ phase (marked as #, PDF#18−0407) coexisted between 800 and 1000 °C, where the HEA becomes the dominant phase. The fraction of the HEA *fcc* phase increases with the increase of equilibration temperature, while the fraction of the $Mn_3Co_7$ phase decreased from 800 to 1000 °C, suggesting that more Mn and Co atoms diffused into the HEA *fcc* crystal lattice to produce a near-equimolar mixture of component by way of an effect driven by thermodynamics. During annealing at 1000 °C for 3 h, full conversion to single-phase HEA occurred without observation of additional peaks, suggesting the complete formation of FeCoNiMnRu HEA. Figure 2h clearly shows that the peak positions of (111) planes for HEA initially shifted to lower $2\theta$ angles between 800 and 1000 °C and the asymmetry of diffraction peak strengthened with increased temperatures, suggesting the

generation of a larger lattice distortion caused by differences in the atomic radii[10]. Then, the prolonged annealing treatment can reduce the lattice distortion of the HEA crystal, as evidenced by the positively shifted peak position and the enhanced symmetry of diffraction peaks. In regard of (200) and (220) planes (Supplementary Fig. 11), the peak shifts display the same trend as that of (111) planes.

We proposed a possible growth process of HEA NPs and the thermodynamically driven phase transition of FeCoNiMnRu NPs in CNF nanofiber reactors is illustrated in Fig. 2i. During graphitization, the Fe/Co/Ni/Mn/Ru mixed metal precursors decomposed first, and then the reduced metal clusters were bonded and confined within the PAN-derived CNFs. At relative low temperature 600–800 °C, the metal elements with small atom radii differences prefer to form alloy phase and insufficient heating energy at low temperature cause the slightly atom diffusion, which make both of HEA and $Mn_3Co_7$ phases co-exist. At high temperature 1000 °C, sufficient dynamic energy caused the metal atoms to diffuse dramatically, leading to homogeneous formation of single-phase HEA alloy. It is concluded that the high temperature coupled with prolonged heating treatment provided the activation energy that drove complete mixing of multiple metal element atoms. The XRD patterns of FeCoNiMnRu/CNFs synthesized at 800, 900, and 1000 °C under prolonged heat treatment for 3 h were also performed. As shown in Supplementary Fig. 12, compared with the in situ XRD patterns of FeCoNiMnRu/CNFs without prolonged heat treatment, all the diffraction peaks for *fcc* HEA NPs ((111), (200), (220) planes) exhibit positively shifts to high values, suggesting the reduced lattice parameters and lattice distortion. It is indicated that after the prolonged heat treatment for 3 h, all of the diffraction peaks for $Mn_3Co_7$ phase vanished, suggesting the complete formation of FeCoNiMnRu HEA. Figure 2f and Supplementary Fig. 13 show the XRD patterns of FeCoNiMnRu/CNFs-800-3h and FeCoNiMnRu/CNFs-1000-3h with detailed Rietveld refinements. Both of the FeCoNiMnRu HEA NPs obtained at 800 °C and 1000 °C under 3 h prolonged treatment was single phase structure with similar component. Therefore, negatively shifted peak position through in situ XRD results from 800 to 1000 °C suggest the growth process of single-phase HEA (Fig. 2g). The XRD results of FeCoNiMnRu/CNFs with prolonged heat treatment from 1000 °C to 1000-3h exhibit a positively shifted peaks position, further show a structural symmetry optimization by reducing the lattice distortion in HEA NPs.

The X-ray absorption spectroscopy (XAS) (Fig. 3a–d) was performed to investigate the chemical states of FeCoNiMnRu/CNFs. The X-ray absorption near-edge structure (XANES) results (Fig. 3a and c) indicate that the pre-absorption edge features for Co and the absorption edge for Ru both are metallicity by comparing with reference metal Co, CoO and $Co_3O_4$ foils, demonstrating that the Co and Ru elements in HEA NPs are in metallic state. The post-edge for Co and Ru in HEA exhibits slight deviation in the shape and intensity when compared with the reference metal Co and Ru foils. These features indicate the alloy formation rather than elemental segregation into pure metals, which would show the same length as metal Co and Ru foils. The extended X-ray absorption fine structure (EXAFS) of Co and Ru were determined through the fitting of the Fourier transform (FT) spectra. As shown in Fig. 3b and d, the FT-EXAFS spectra indicate that the average bond length of Ru and Co in HEA NPs is quite different from the metallic bond in bulk Co and Ru references, suggesting that the Co and Ru elements are surrounded by different metallic species (Fe, Mn and Ni). The bond structures of Co and Ru in HEA reveal the similar average bond length without any oxidation when compared CoO and $Co_3O_4$ foils, further confirming the metallic states of Co and

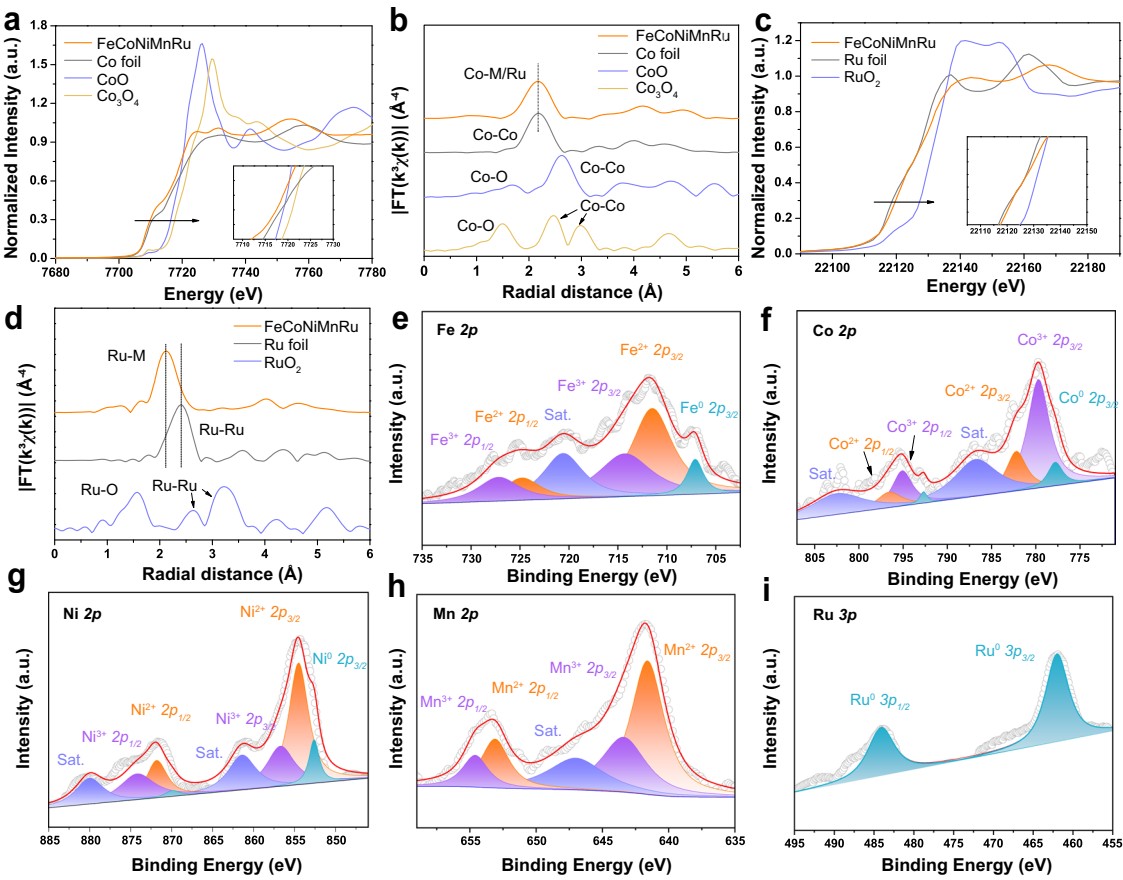

**Fig. 3 Surface chemical states characterized by X-ray absorption spectroscopy (XAS) and X-ray photoelectron spectroscopy (XPS). a** The Co K-edge XANES spectra and **b** FT-EXAFS spectra of FeCoNiMnRu/CNFs, Co foil, CoO foil, and $Co_3O_4$ foil. **c** The Ru K-edge XANES spectra and **d** FT-EXAFS spectra of FeCoNiMnRu/CNFs, Ru foil, and $RuO_2$ foil. High-resolution XPS spectra of as-prepared FeCoNiMnRu/CNFs: **e** Fe 2p, **f** Co 2p, **g** Ni 2p, **h** Mn 2p, and **i** Ru 3p.

Ru in HEA after stability test. According to the EXAFS fitting (Supplementary Fig. 14), the bond length (R) and coordination numbers of each bond type in the HEA were summarized in Supplementary Table 4. The reliability of the fitting method is supported by smaller R factors.

X-ray photoelectron spectroscopy (XPS) was utilized to investigate the surface compositions and electronic effects of the as-prepared FeCoNiMnRu/CNFs. The XPS survey spectrum of the as-prepared FeCoNiMnRu/CNFs is shown in Supplementary Fig. 15. Figure 3e shows the presence of $Fe^0$, $Fe^{2+}$ and $Fe^{3+}$ species, and the BEs at 707.1, 711.5, 724.8 eV, 714.3 and 727.2 eV are ascribed to $Fe^0$ $2p_{3/2}$, $Fe^{2+}$ $2p_{3/2}$, $Fe^{2+}$ $2p_{1/2}$, $Fe^{3+}$ $2p_{3/2}$ and $Fe^{3+}$ $2p_{1/2}$, respectively. Another peak with a BE at 720.6 eV is attributed to the satellite peaks of Fe 2p[25]. For the Co 2p spectrum (Fig. 3f), the main doublets at 777.8, 792.7, 779.7, 795.1, 782.2 and 796.6 eV correspond to $Co^0$ $2p_{3/2}$, $Co^0$ $2p_{1/2}$, $Co^{3+}$ $2p_{3/2}$, $Co^{3+}$ $2p_{1/2}$, $Co^{2+}$ $2p_{3/2}$ and $Co^{2+}$ $2p_{1/2}$, respectively[26]. The Ni 2p spectrum (Fig. 3g) displays the coexistence of metallic $Ni^0$ (852.6 and 869.9 eV for $Ni^0$ $2p_{3/2}$ and $Ni^0$ $2p_{1/2}$), $Ni^{2+}$ (854.5 and 871.8 eV for $Ni^{2+}$ $2p_{3/2}$ and $Ni^{2+}$ $2p_{1/2}$), and $Ni^{3+}$ (856.7 and 874.2 eV for $Ni^{3+}$ $2p_{3/2}$ and $Ni^{3+}$ $2p_{1/2}$)[27]. The high-resolution Mn 2p spectrum (Fig. 3h) suggests the coexistence of $Mn^{2+}$ $2p_{3/2}$ (641.6 eV), $Mn^{2+}$ $2p_{1/2}$ (653.1 eV), $Mn^{3+}$ $2p_{3/2}$ (643.4 eV) and $Mn^{3+}$ $2p_{1/2}$ (654.6 eV). Two satellite peaks (marked as "Sat.") appear at binding energies (BEs) of 648.5 and 659.7 eV[28].

The Ru 3p XPS spectrum of FeCoNiMnRu/CNFs (Fig. 3i) exhibit only metallic Ru states (462.0 and 484.1 eV for $Ru^0$ $3p_{3/2}$ and $Ru^0$ $3p_{1/2}$)[29]. The O 1 s spectrum (Supplementary Fig. 16) reveals peaks at 531.8 and 532.8 eV, which are ascribed to hydroxyl groups and residual oxygen-containing groups on the surface of FeCoNiMnRu/CNFs, respectively[30]. There are no peaks observed for lattice $O^{2-}$ of metal oxides, and the strong adsorption of hydroxyl groups would be beneficial for water splitting. The Fe 2p, Co 2p, Ni 2p, Mn 2p and Ru 2p XPS spectra of the controlled samples including Ru/CNFs, FeCoNi/CNFs, FeCoNiMn/CNFs, and FeCoNiRu/CNFs were performed to provide more electron interaction information among the metal elements in HEA (Supplementary Fig. 17). The corresponding BE information were summarized in Supplementary Table 5. The BEs for Co, Ni, Fe and Mn in FeCoNiMnRu/CNFs show negative shifts when compared with FeCoNi/CNFs, FeCoNiRu/CNFs and FeCoNiMn/CNFs, respectively (Supplementary Fig. 17a–d). As shown in Supplementary Fig. 17e, the BEs for Ru in FeCoNiMnRu/CNFs exhibit positive shift when compared with the FeCoNiRu/CNFs and Ru/CNFs, suggesting that the Ru in HEA NPs served as the electron acceptor. The results strongly demonstrate the electron transfers from Fe, Co, Ni and Mn atoms to Ru atoms in HEA NPs that is due to the higher electronegativity of Ru (2.20) than those of Ni (1.91), Co (1.88), Fe (1.83), and Mn (1.55)[31].

**Evaluation of electrochemical performance.** To evaluate the electrochemical performance, all of the as-prepared electrocatalysts were used as the working electrodes in a typical three-electrode system. A saturated calomel electrode (SCE) electrode and graphite rod were used as reference and counter electrodes, respectively. The HER, oxygen evolution reaction (OER) and overall water splitting (OWS) reaction were performed in Ar-saturated 1.0 M

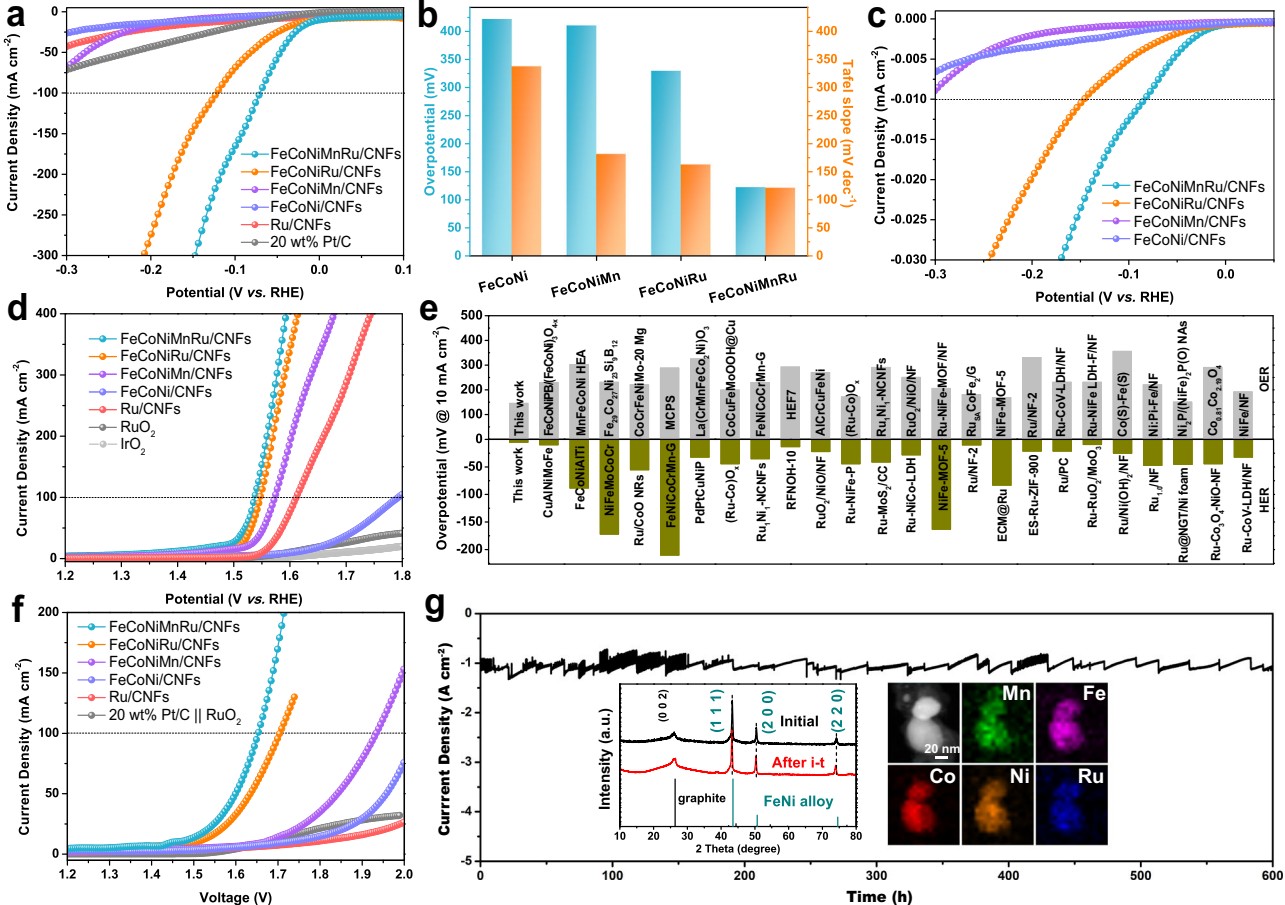

**Fig. 4 Electrochemical performance of HEA electrocatalysts. a** HER polarization curves, **b** the corresponding histogram for overpotentials at 100 mA cm$^{-2}$ and Tafel slopes obtained for Ru/CNFs, FeCoNi/CNFs, FeCoNiMn/CNFs, FeCoNiRu/CNFs, FeCoNiMnRu/CNFs, and Pt/C in 1.0 M KOH electrolyte. **c** ECSA-normalized HER polarization curves of the as-prepared electrocatalysts. **d** OER polarization curves for the as-prepared electrocatalysts, commercial RuO$_2$ and IrO$_2$ in 1.0 M KOH electrolyte. **e** Comparison of HER and OER overpotentials at 10 mA cm$^{-2}$ in 1.0 M KOH for different catalysts. Those recently reported literatures for HER and OER electrocatalysts were shown in Supplementary Tables 6 and 7. **f** Polarization curves for full water splitting by the as-prepared electrocatalysts in a two-electrode configuration at a scan rate of 2 mV s$^{-1}$. **g** Long-term stability measurement of the FeCoNiMnRu/CNFs electrode at −1.16 V vs. RHE in 1 M KOH electrolyte for 600 h. The insets in (**g**) are the XRD patterns (left) and STEM-EDS mapping images (right) of FeCoNiMnRu/CNFs after the long-term stability test.

KOH electrolyte, and all potentials were calibrated with a reversible hydrogen electrode (RHE). As shown in Fig. 4a, the FeCoNiMnRu/CNFs achieves the lowest overpotentials of 71 mV to produce a current density of 100 mA cm$^{-2}$ and a Tafel slope of 67.4 mV dec$^{-1}$ (Supplementary Fig. 18), which are much more excellent than the indicated values for Ru/CNFs (421 mV and 337.1 mV dec$^{-1}$), FeCoNi/CNFs (410 mV and 180.9 mV dec$^{-1}$), FeCoNiMn/CNFs (329 mV and 162.3 mV dec$^{-1}$), and FeCoNiRu/CNFs (122 mV and 120.9 mV dec$^{-1}$). The commercial Pt/C cannot support a current density of 100 mA cm$^{-2}$ below −0.2 V, suggesting the excellent HER activity of HEAs under alkaline conditions. The low Tafel slope of FeCoNiMnRu/CNFs (67.4 mV dec$^{-1}$) indicates the operation of the Volmer–Heyrovsky mechanism[22]. The over-potentials at 100 mA cm$^{-2}$ and Tafel slopes of the as-prepared electrocatalysts are summarized in Fig. 4b. The remarkable elec-trocatalytic activity is further supported by electrochemical impe-dance spectroscopy (EIS) performed at an overpotential of 50 mV. As shown in Supplementary Fig. 19, the Nyquist plots of the as-prepared electrodes exhibit the characteristic semicircles. The FeCoNiMnRu/CNFs presents the smallest charge transfer resistance (R$_{ct}$) value of 11.4 Ω compared to the FeCoNi/CNFs (637.9 Ω), FeCoNiMn (950.1 Ω) and FeCoNiRu/CNFs (18.6 Ω), which served to accelerate the sluggish reaction kinetics.

To evaluate the active sites and the intrinsic activities of the as-prepared electrocatalysts, electrochemical surface areas (ECSAs) were measured by a double-layer capacitance (C$_{dl}$) method[32]. As shown in Supplementary Fig. 20, the C$_{dl}$ values for FeCoNi/CNFs, FeCoNiMn/CNFs, FeCoNiRu/CNFs, and FeCoNiMnRu/CNFs are 16.4, 62.1, 110.5, and 104.6 mF. The ECSA–normalized LSV curves (Fig. 4c) show that the intrinsic activity of FeCoNiMnRu/CNFs at 0.01 mA cm$^{-2}$ (ECSA) is 85 mV, which is remarkably enhanced relative to the values of FeCoNi/CNFs (341 mV), FeCoNiMn/CNFs (309 mV), and FeCoNiRu/CNFs (146 mV), suggesting the high intrinsic HER activity of HEA. Figure 4d shows the OER LSV curves of the as-prepared electrodes. The FeCoNiMnRu/CNFs requires the lowest overpotential of 308 mV to reach 100 mA cm$^{-2}$, as shown by the indicated values for Ru/CNFs (564 mV), FeCoNi/CNFs (382 mV), FeCoNiMn/CNFs (344 mV), and FeCoNiRu/CNFs (318 mV). The remarkably enhanced OER kinetics of the FeCoNiMnRu HEA are reflected by the low Tafel slope of 61.3 mV dec$^{-1}$ (Supplementary Fig. 21).

Comparisons of the overpotentials and Tafel slopes for different electrocatalysts are summarized in Supplementary Fig. 22. The OER results exhibit a trend similar to that for HER activity in that HEA shows superior activity. As shown in Supplementary Fig. 23, the HER and OER LSV curves normalized

by geometric area and mass loading of noble metal indicate that the FeCoNiMnRu/CNFs exhibit outstanding electrocatalytic activities and even prominently surpass the state-of-art Pt/C (20 wt%), $IrO_2$ and $RuO_2$ electrocatalysts at high current density. At overpotential of 300 mV for HER the FeCoNiMnRu/CNFs show higher mass activity of 2666 than that of Pt/C (1688 mA $mg^{-1}_{Pt}$). In addition, at overpotential of 450 mV for OER, the FeCoNiMnRu/CNFs obtain the highest mass activity of 487 mA $mg^{-1}_{Ru}$, which is significantly higher than $RuO_2$ (146 mA $mg^{-1}_{Ru}$), and $IrO_2$ (49 mA $mg^{-1}_{Ir}$). Furthermore, the overpotentials required for a current density of 10 mA $cm^{-2}$ are compared with those recently reported HER, OER, and OWS electrocatalysts in alkaline electrolytes (Fig. 4e, Supplementary Tables 6–8), which demonstrate the excellent activity of FeCoNiMnRu/CNFs.

In addition, the electrocatalytic activities of Ru-containing electrocatalysts have been further evaluated by comparing a series of FeCoNiMnRu/CNFs with different Ru contents. The FeCoNiMnRu/CNFs with different Ru contents were denoted as FeCoNiMnRu$_{0.5}$/CNFs, FeCoNiMnRu/CNFs, and FeCoNiMnRu$_2$/CNFs, and the corresponding Ru contents were 2.41, 4.23 and 7.13 wt%, respectively, which were measured by ICP-OES. As shown in Supplementary Fig. 24, it is shown that the FeCoNiMnRu/CNFs and FeCoNiMnRu$_2$/CNFs display very similar values of overpotentials and Tafel slopes at 100 mA $cm^{-2}$, while FeCoNiMnRu$_{0.5}$/CNFs show obviously inferior activity. The ECSA values of FeCoNiMnRu/CNFs, FeCoNiMnRu$_{0.5}$/CNFs and FeCoNiMnRu$_2$/CNFs were calculated to be 2614, 2818, and 2093 $cm^2$ for FeCoNiMnRu/CNFs, FeCoNiMnRu$_{0.5}$/CNFs, and FeCoNiMnRu$_2$/CNFs, respectively. The $R_{ct}$ values of FeCoNiMnRu/CNFs, FeCoNiMnRu$_{0.5}$/CNFs, and FeCoNiMnRu$_2$/CNFs were measured to be 11.4, 16.3, and 5.1 $\Omega$, respectively. The OER activities were also evaluated by LSV curves. The overpotentials at 100 mA $cm^{-2}$ of FeCoNiMnRu/CNFs, FeCoNiMnRu$_{0.5}$/CNFs, and FeCoNiMnRu$_2$/CNFs were determined to be 308, 324, and 299 mV, respectively. The results indicate that FeCoNiMnRu/CNFs achieved higher current density than that of FeCoNiMnRu$_2$/CNFs at high overpotential (>300 mV). Therefore, at relative low Ru content range (2–4 wt%), high Ru contents would lead to enhanced activity for water splitting. When the Ru contents increased to very high values (>4 wt%), the Ru contents could pose negligible or negative effects on improving the electrocatalytic activity. Inspired by the above results, we further fabricated an alkaline electrolyzer by employing FeCoNiMnRu/CNFs as both the anode and cathode to explore practical electrolytic applications (Fig. 4f). Interestingly, the FeCoNiMnRu/CNFs||FeCoNiMnRu/CNFs system requires only a low voltage of 1.65 V at 100 mA $cm^{-2}$, which is lower than those of FeCoNiMn/CNFs||FeCoNiMn/CNFs (1.93 V), FeCoNiRu/CNFs||FeCoNiRu/CNFs (1.71 V), and Pt/C||$RuO_2$ (Supplementary Fig. 25).

The electrochemical durability of FeCoNiMnRu/CNFs was characterized by LSV, CV cycles, and chronoamperometry measurements. Supplementary Fig. 26 shows the LSV curves of FeCoNiMnRu/CNFs before and after 10000 CV cycles; they nearly overlap with each other, suggesting the superior stability of FeCoNiMnRu/CNFs. The chronoamperometric curve (Fig. 4g) for FeCoNiMnRu/CNFs was measured at −1.16 V vs. RHE for more than 600 h in 1.0 M KOH. The current density at 1 A $cm^{-2}$ displays no evident changes, also suggesting its remarkable stability. This is ascribed to the wondrous corrosion resistance of HEA structures. The OER stability of FeCiNiMnRu/CNFs was conducted at 1.55 V vs. RHE for 10 h (Supplementary Fig. 27). The current density showed no obvious decay during the test, and furthermore, the LSV curves before and after stability display negligible change. In addition, as shown in Supplementary Fig. 28,

at 60 °C, the FeCoNiMnRu/CNFs can maintain a high current density of 1000 mA $cm^{-2}$ at −2.22 V vs. RHE for 100 h, suggesting no obvious degradation in current density. In 10 M KOH, the FeCoNiMnRu/CNFs also can afford a high current density of 1000 mA $cm^{-2}$ at −0.77 V vs. RHE for 100 h without current density degradation (Supplementary Fig. 29).

As illustrated in the FE-SEM, TEM and HRTEM images of the FeCoNiMnRu/CNFs electrode after the long-term stability test (Supplementary Fig. 30), the electrode can well maintain its initial nanoparticle morphology comprising HEA and 3D nanofiber networks. XRD patterns (inset in Fig. 4g) of FeCoNiMnRu/CNFs obtained before and after the stability test show that the HEA can also retain the same *fcc* structure seen for the initial structure without any newly formed phases, suggesting ultrastable HER performance with an alkaline electrolyte. STEM-EDS element mapping images (inset in Fig. 4g) confirm the lack of phase separation and the homogeneous distribution of Fe, Co, Ni, Mn, and Ru in HEA NPs after the stability test.

We further used the XAS to investigate the chemical states of FeCoNiMnRu/CNFs before and after the HER stability test. As shown in Fig. 5a–f, the pre-absorption edge features for Co and the absorption edge for Ru both are metallicity by comparing with reference metal Co and Ru foils, demonstrating that the Co and Ru elements in HEA NPs are in metallic state. Meanwhile, after the stability test, the XANES of Co and Ru in HEA NPs also keep metallic state, suggesting the excellent oxidation resistance during the long-term HER stability test. The FT-EXAFS spectra (Fig. 5b and e) indicate that the bond structure of Co and Ru in HEA before and after stability test reveal the similar average bond length without any oxidation when compared CoO and $Co_3O_4$ foils (Fig. 3a–d), further confirming the metallic states of Co and Ru in HEA after stability test. According to the EXAFS fitting (Fig. 5c and f), the bond length (R) and coordination numbers of each bond type in the HEA before and after stability test were summarized in Supplementary Table 4. The FT-EXAFS and WT-EXAFS spectra (Fig. 5g–i) of Co and Ru in HEA before and after stability tests have negligible mismatch, which means that the FeCoNiMnRu HEA keep metallic states in long-term stability tests, exhibiting extraordinary durability. The reliability of the fitting method is supported by smaller R factors.

**The relationship between active sites and intermediates.** These discriminating enhancements of HER, OER, and OWS activities imply key roles for multiple metals serving as active centers, and we further used density functional theory (DFT) calculations to determine the cooperation of multiple metal active sites in the alkaline HER. The Tafel slope of the FeCoNiMnRu/CNFs (67.4 mV $dec^{-1}$) suggested the Volmer–Heyrovsky reaction pathway. Figure 6a illustrates the atomic configurations at catalytic sites of FeCoNiMnRu HEA in the four different stages. The $H_2O^*$ molecule absorbed on the HEA surface (stage 1) is destabilized at the H−OH bond (stage 2), which is then dissociated to generate coadsorption of $H^*$ and $OH^*$ intermediates (stage 3). The $H^*$ intermediate will be detached from the surfaces after combining with another $H^*$ to give $H_2$ production (stage 4). The water dissociation into $H^*$ and $OH^*$ and adsorption of the $H^*$ are potential-determining steps (PDS), which determine the water dissociation rates. The chemical structures of the FeCoNiMnRu HEA were shown in Supplementary Fig. 31. Figure 6b shows the energy profile for water dissociation on Fe, Co, Ni, and Ru sites of the FeCoNiMnRu HEA surface at four states. In addition, the chemical structures and atomic configurations of Fe, Co, Ni, and Ru sites of FeCoNiMnRu HEA during $H_2O$ dissociation are shown in Supplementary Figs. 32–35. Interestingly, the energy barrier for breaking the H−OH bond (stage 1 → stage 2) on Co

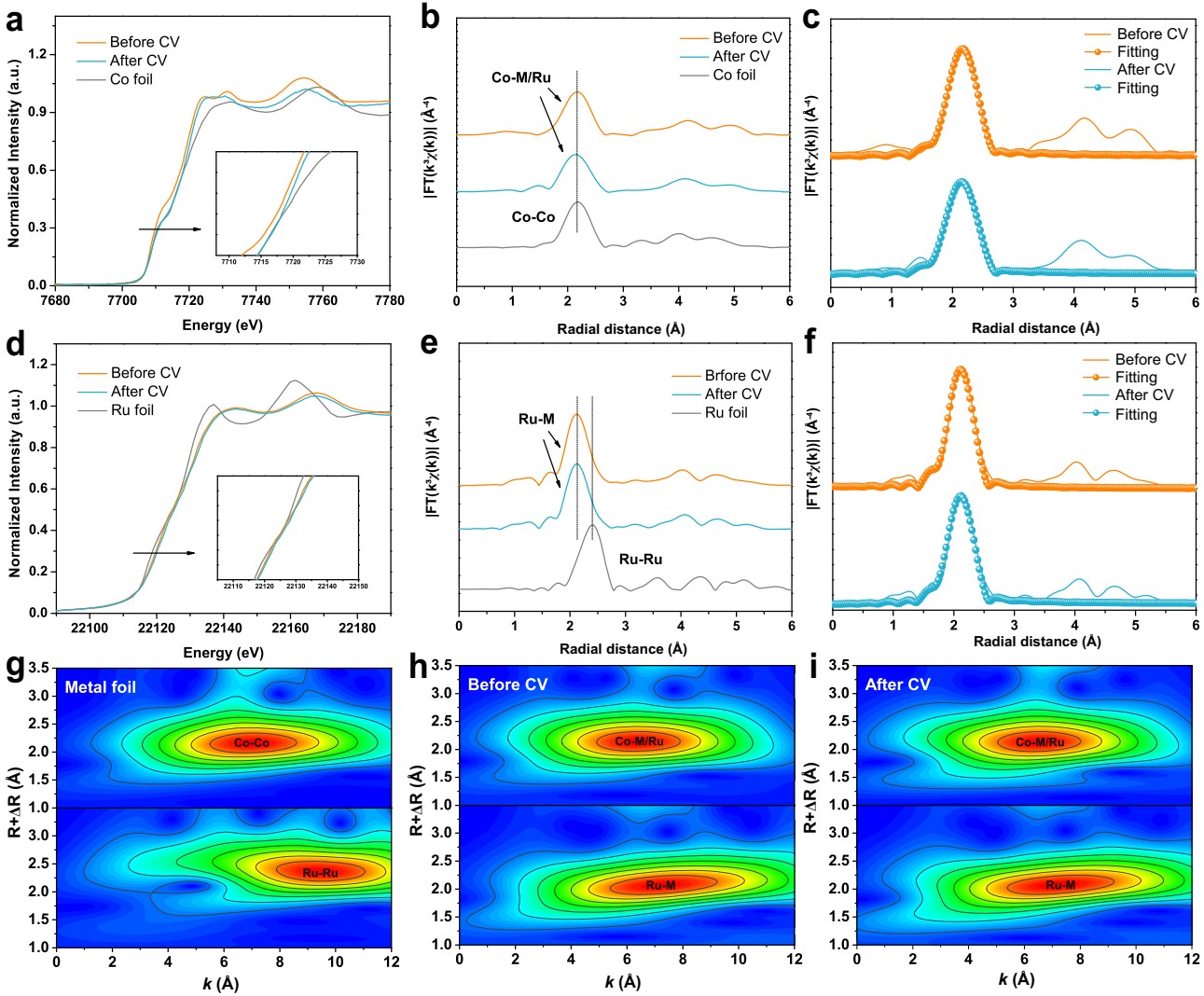

**Fig. 5 Stability characterization by XAS. a** The Co K-edge XANES spectra and **b** FT-EXAFS spectra of FeCoNiMnRu/CNFs before and after stability test, and Co foil. **c** The corresponding FT-EXAFS fitting curves of FeCoNiMnRu/CNFs before and after stability test. **d** The Ru K-edge XANES spectra and **e** FT-EXAFS spectra of FeCoNiMnRu/CNFs before and after stability test, and Ru foil. **f** The corresponding FT-EXAFS fitting curves of FeCoNiMnRu/CNFs before and after stability test. The WT-EXAFS spectra of Co and Ru in (**g**) metal foils and **h**, **i** FeCoNiMnRu/CNFs before and after stability tests.

site of HEA is the lowest as 0.34 eV in comparison with the Fe site (0.70 eV), Ni site (0.63 eV), and Ru site (0.67 eV). These results suggest that the $H_2O$ adsorption and dissociation are more favorable at Co site, which is beneficial to accelerating water dissociation for the generation of $H^*$ intermediates.

We further calculated the Gibbs free energies of atomic hydrogen adsorbed ($\Delta G_{H^*}$) (Fig. 6c) at four catalytic sites of HEA to reveal the influence of different metal sites on hydrogen adsorption. Atomic configurations of the FeCoNiMnRu HEA at the $H^*$ adsorption stage on Fe, Co, Ni, and Ru sites are shown in Supplementary Fig. 36. The DFT results indicated that the Ru sites achieve the most appealing $\Delta G_{H^*}$ of −0.07 eV, as compared to those of Fe (−0.13 eV), Ni (−0.27 eV), and Co (−0.43 eV), which suggests that $H^*$ is preferentially stabilized at the Ru sites. Therefore, during the whole electrocatalytic water splitting process, the Co and Ru sites function to simultaneously accelerate the $H_2O$ disassociation and $H^*$ adsorption with the lowest energies, and this remarkable cooperation avoids active site blocking and accelerates the whole water dissociation process.

The active sites in HEA for the stabilization of intermediates were further investigated by operando electrochemical Raman

spectra. As shown in Fig. 6d, the Raman spectrum of FeCoNiMnRu/CNFs determined at 0 V displays three peaks at 1316, 1590, and 2616 $cm^{-1}$, corresponding to the D band, G band, and 2D band of CNFs, respectively[33]. When a potential of 60 mV was applied, new Raman peaks corresponding to Fe–O bonds were observed at 215 and 290 $cm^{-1}$, while Raman peaks at 585 and 704 $cm^{-1}$ were ascribed to Co-O bonds[34–36]. The Raman peaks located at approximately 446 and 530 $cm^{-1}$ were attributed to the emergence of Ni-O bonds[37]. The newly formed Fe–O, Ni–O, and Co–O bonds suggest the generation of Fe-OH$^*$, Ni-OH$^*$, and Co-OH$^*$ intermediates, which originate from the cleavage of $H_2O$ molecules. Interestingly, two sharp Raman peaks that emerged at 2069 and 2092 $cm^{-1}$ correspond to the Ru-H bonds, strongly suggesting the formation of Ru-H$^*$ intermediates[38,39]. With increasing applied potentials ranging from 60 to 180 mV, the intensities of all characteristic Raman peaks continuously increased, suggesting enhanced HER activity. We further obtained operando electrochemical Raman spectra of FeCoNiMn/CNFs during the HER process (Supplementary Fig. 37). Without the Ru metal, no Raman peaks for Ru–H bonds were observed in the vicinity of 2000–2200 $cm^{-1}$, directly

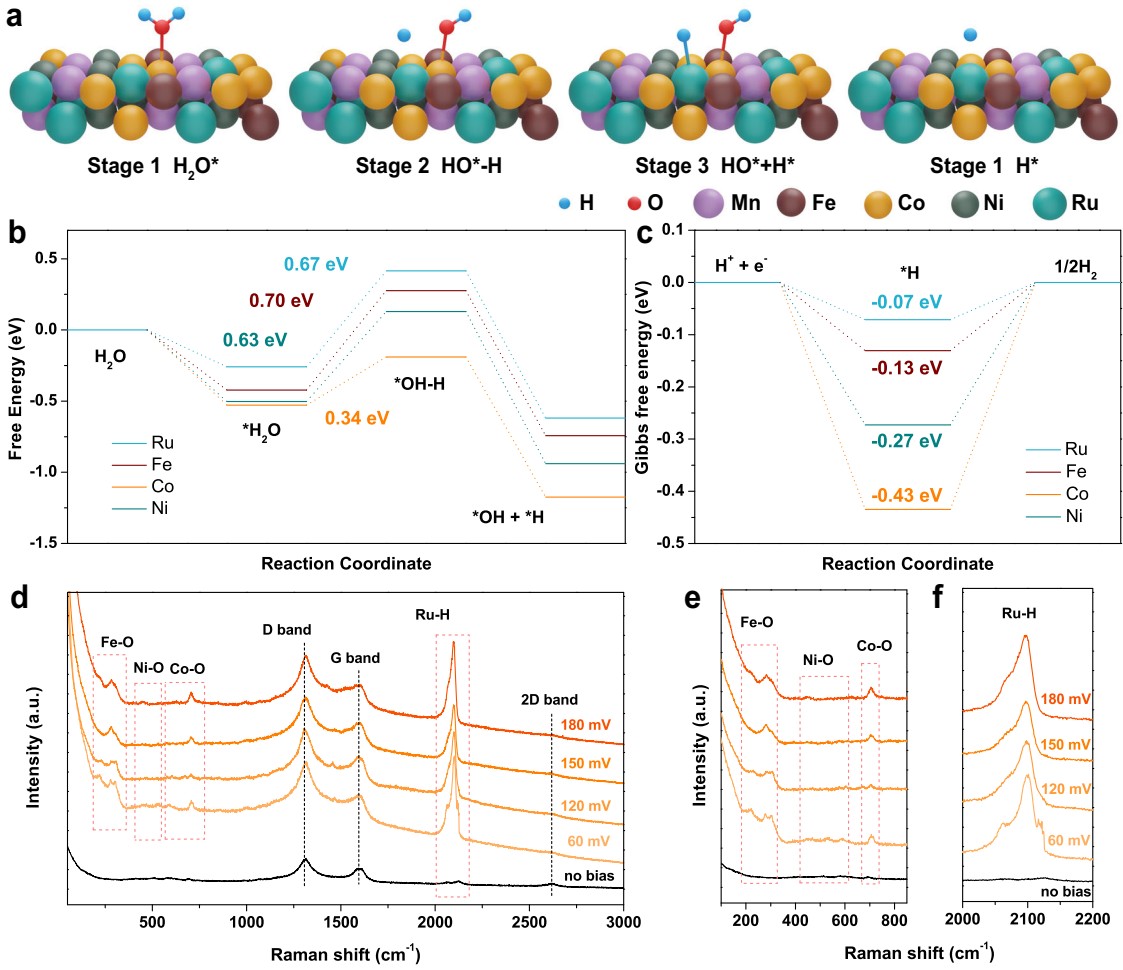

**Fig. 6 Theoretical calculation and in situ electrochemical-Raman characterization. a** The atomic configurations on catalytic sites of FeCoNiMnRu HEA at the four stages during $H_2O$ dissociation. **b** Reaction energy profile for water dissociation on various catalytic sites of the FeCoNiMnRu HEA surface. **c** Gibbs free energy ($\Delta G_{H^*}$) profiles on various catalytic sites of the FeCoNiMnRu HEA surface. **d–f** Operando electrochemical-Raman spectra collected for FeCoNiMnRu/CNFs during the HER process in 1.0 M KOH electrolyte.

confirming the ability of Ru to absorb $H^*$. The Raman results provided direct evidence that Co and Ru active sites in the FeCoNiMnRu HEA stabilize different intermediates. In the typical FeCoNiMnRu HEA, the Co sites facilitate $H_2O$ dissociation, and the Ru sites simultaneously accelerate the combination of $H^*$ to $H_2$. Therefore, the stabilization of multiple intermediates on various active sites in HEA was verified experimentally and theoretically.

The above results indicated that the Fe, Co, Ni, and Ru sites in HEAs play different roles in the HER, and we further present a series of FeCoNiXRu (X = Mn, Cr, Cu) HEAs by using fifth elements (X) with different electronegativities to understand the relationship between electronegativity and HER activity. Considering that the difference in atomic configurations may change the adsorption energy, 8 randomly selected configurations of FeCoNiXRu (X = Mn, Cr, Cu) HEA were analyzed by DFT calculation. The chemical structures of FeCoNiMnRu, FeCoNiCrRu and FeCoNiCuRu HEA with 8 randomly selected configurations are shown in Supplementary Figs. 38–40. As shown in Fig. 7a and b, the overpotentials at $100 \, mA \, cm^{-2}$ for FeCoNiCuRu/CNFs (245 mV), FeCoNiCrRu/CNFs (126 mV), and FeCoNiMnRu/CNFs (71 mV) suggest a strong relationship with the electronegativity of the fifth metal in HEA. For the OER (Supplementary Fig. 41), the low electronegativity of Mn (1.55) compared with those of Cr (1.66) and Cu (1.90) gives

FeCoNiMnRu/CNFs the best OER activity with the lowest overpotential of 308 mV at current density of $100 \, mA \, cm^{-2}$. The d-band center of each metal sites in FeCoNiMnRu and the three HEA structures were calculated and shown in Fig. 7c and d. The d-band orbitals with large spin polarization can be divided into spin up and spin down. As shown in Fig. 7d, a smaller number of spin states occupy the spin down orbitals, which is likely to be a spin-polarized catalytic active center and generally participate in catalytic reactions. As shown in Supplementary Figs. 42–44, there are much more electrons can be observed on Ru after the charge distributions in all three HEA structures. The results indicate the charge transfers from Fe, Co, Ni and Mn metals to Ru metal.

We further calculated the energy profile for water dissociation on Fe (Supplementary Figs. 45, 46), Co (Fig. 7e), Ni (Supplementary Figs. 47, 48), and Ru (Supplementary Fig. 49) sites on different FeCoNiMnRu, FeCoNiCrRu, and FeCoNiCuRu HEA surfaces. As shown in Fig. 7e, the energy profiles for the water dissociation on Co site of the three HEA surfaces demonstrate that the Co sites in FeCoNiMnRu HEA has the lowest energy barrier (0.34 eV) for dissociation of water into $H^*$ and $OH^*$ when compared with those of FeCoNiCrRu (0.47 eV) and FeCoNiCuRu (0.66 eV) HEAs. Figure 7f strongly confirms that the electronegativity of the fifth metal in HEA can regulate the energy barrier for water dissociation at each metal site. In particular,

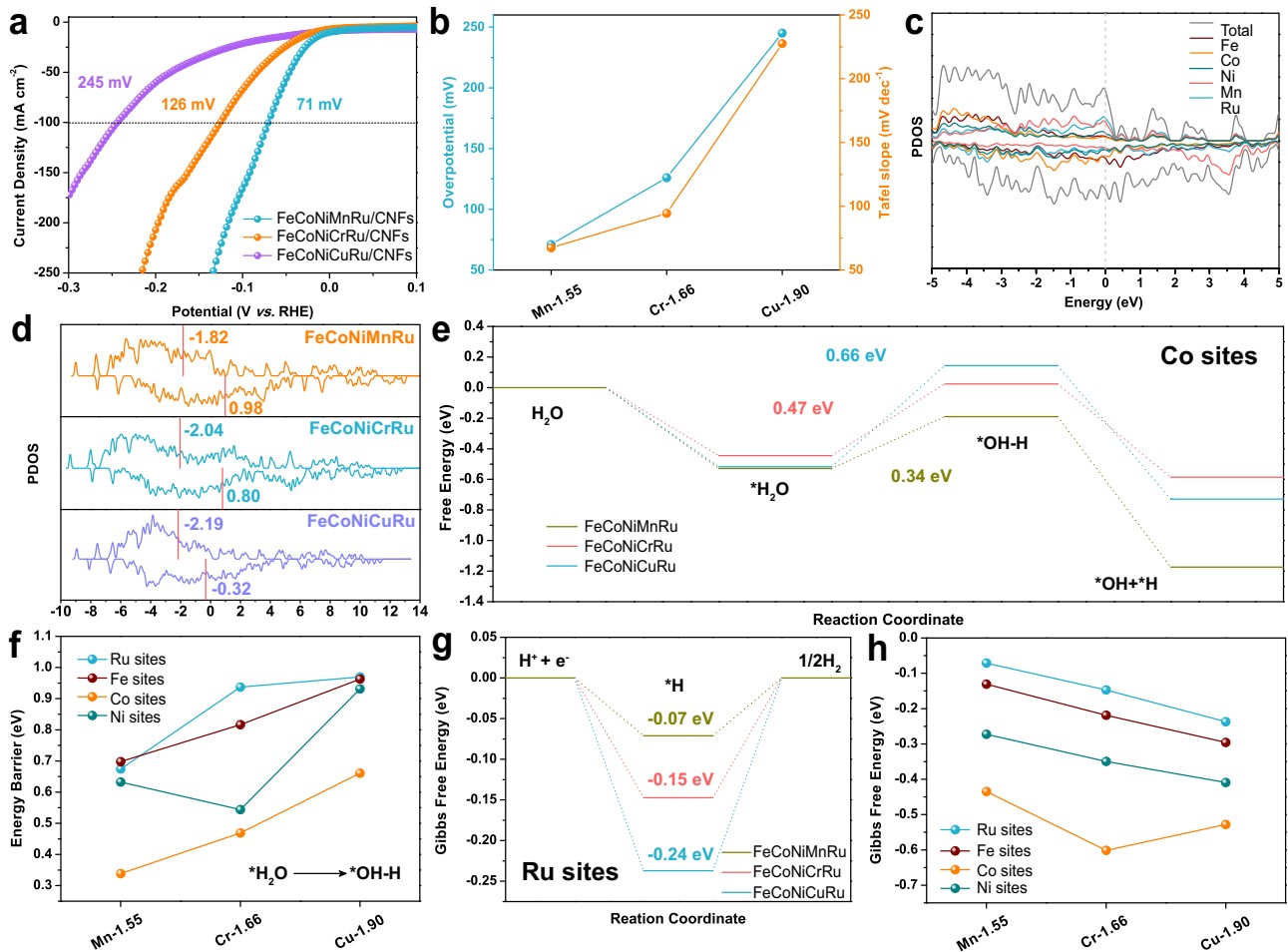

**Fig. 7 Relationship between metal electronegativity and electrochemical performance. a** HER polarization curves obtained on FeCoNiXRu/CNFs (X = Cr, Mn, and Cu) in 1.0 M KOH electrolyte. **b** Correlation between the HER overpotentials at 100 mA cm$^{-2}$, Tafel slopes, and the electronegativities of metals X (Cr, Mn, and Cu). **c** The d-orbital projected density of states (PDOS) of Fe, Co, Ni, Mn, Ru, and FeCoNiMnRu. **d** Comparison of PDOS of FeCoNiXRu (X = Mn, Cr, Cu) HEA. **e** Reaction energy profile for water dissociation at Co sites of FeCoNiMnRu, FeCoNiCrRu, and FeCoNiCuRu HEA surfaces. **f** Correlation between the energy barrier for H$_2$O dissociation at different metal sites and the electronegativities of metals (Cr, Mn, and Cu). **g** Gibbs free energy (ΔG$_{H*}$) profiles at Ru sites on the FeCoNiMnRu, FeCoNiCrRu, and FeCoNiCuRu HEA surfaces. **h** Correlation between ΔG$_{H*}$ at different metal sites and the electronegativities of metals X (Cr, Mn, and Cu).

energy barriers for water dissociation on Fe, Co, and Ru sites are reduced by introducing the less-electronegative Mn, and the Co site still exhibits the lowest value of 0.34 eV, suggesting Co sites are the preferred locations for water disassociation.

Additionally, the calculated ΔG$_{H*}$ values for Ru, Fe, Co, Ni sites in FeCoNiMnRu, FeCoNiCrRu and FeCoNiCuRu are shown in Fig. 7g and Supplementary Figs. 50–52. The atomic configurations for H* adsorption at the Fe, Co, Ni, and Ru catalytic sites of FeCoNiCrRu, and FeCoNiCuRu HEA are shown in Supplementary Figs. 53 and 54. The results also demonstrate that optimized ΔG$_{H*}$ values can be realized by introducing less-electronegative metals (Fig. 7h). Furthermore, all Ru sites in FeCoNiMnRu, FeCoNiCrRu, and FeCoNiCuRu showed the most appealing ΔG$_{H*}$ compared with those of Fe, Co, and Ni sites, indicating that Ru sites are still the preferred sites for H* adsorption. The ΔG$_{H*}$ on Fe, Co, Ni, and Ru sites in FeCoNiXRu (X = Mn, Cr, Cu) with 8 randomly selected configurations were analyzed by DFT calculation, and the values of ΔG$_{H*}$ were summarized in Supplementary Fig. 55 and Tables 9–11. DFT Results suggest that all of the Ru sites in FeCoNiXRu (X = Mn, Cr, Cu) display lower values than Fe, Co, and Ni sites, demonstrating that the atomic configurations may not affect the ΔG$_{H*}$ order on Fe, Co, and Ni. In addition, the Ru sites in FeCoNiMnRu also exhibit the

lowest ΔG$_{H*}$ than those Ru sites in FeCoNiCrRu, and FeCoNiCuMn HEA, further indicating that the five metal with low electronegativity in HEA could reduce the ΔG$_{H*}$. According to the bond length and coordination numbers of Co and Ru in HEA, as investigated by XAS (Fig. 3), we have chosen the atomic configuration 4 accordingly with XAS results as a representative sample to show the relationship between the ΔG$_{H*}$ and electronegativity, unveiling the electrocatalytic mechanism of HER on FeCoNiXRu (X = Mn, Cr, Cu) HEA.

Based on the above results, the relationships between metal electronegativities in HEAs and the energy barriers for water dissociation and H* adsorption at various active metal sites have been established. Charge transfer between different surface atoms occurs in HEAs containing five principal elements with different electronegativities, further leading to significant charge redistribution on the surfaces of alloys (Supplementary Figs. 42–44). In the FeCoNiMnRu HEA, the electronegativity differences in Fe (1.83), Co (1.88), Ni (1.91), Mn (1.55), and Ru (2.20) induce significant charge redistribution and create the most active Co and Ru sites with optimized energy barriers for simultaneously stabilizing OH* and H* intermediates, greatly promoting the HER efficiency in alkaline solution. We prepared a series of HEAs by fixing Fe, Co, Ni, and Ru metals and varying the fifth metal among Mn, Cr, and

Cu. The decrease in electronegativities from Cu (1.90) to Mn (1.55) leads to the reduced energy barriers for water dissociation and $H^*$ adsorption. Changing in the fifth metal in a HEA did not affect the adsorption energy order on active site in HEA; Co site was the most active sites for $OH^*$ adsorption, while Ru site was the most active sites for $H^*$ adsorption. In a FeCoNiMnRu NP, the Co site was the preferred active site with the lowest energy barriers (0.34 eV) for water dissociation when compared with Fe, Ni, and Ru sites. During the subsequent $H^*$ adsorption/desorption process, $H^*$ leaves Co and is absorbed on Ru sites due to its lowest $\Delta G_{H^*}$ of $-0.07$ eV. Adjustments of the HER activities of HEA catalysts were shown experimentally and theoretically by tailoring the electronegativities of the compositions.

## Discussion

In summary, FeCoNiMnRu HEA NPs were designed and synthesized in electrospun CNFs by combining the electrospinning technique and graphitization process. The transformation from a multiphase to a single-phase HEA was evident from the real-time in situ XRD results and demonstrated a thermodynamically driven phase transition. The FeCoNiMnRu/CNFs achieved low overpotentials of 71 and 308 mV to drive a current density of 100 mA cm$^{-2}$ for the HER and OER, respectively. It required only 1.65 V to achieve two-electrode overall water splitting, and the chronoamperometric curve exhibited a stable current density of 1 A cm$^{-2}$ during continuous electrolysis for more than 600 h in 1.0 M KOH, suggesting its remarkable stability. In the FeCoNiMnRu HEA, the Co site facilitated $H_2O$ dissociation, and the Ru sites simultaneously accelerated the combination of $H^*$ to $H_2$. The ability to stabilize multiple intermediates on various active sites in HEAs was verified experimentally and theoretically. Adjustments of the HER activities of HEA catalysts were shown experimentally and theoretically by tailoring the electronegativities of the composition. This work provides an in-depth understanding of the correlation between specific active sites and intermediates, opening up a fascinating approach for overcoming the scaling relation issues seen for multistep reactions.

## Methods

**Materials**. Anhydrous ferric trichloride (FeCl$_3$, ≥97.0%), nickel chloride hexahydrate (NiCl$_2$·6H$_2$O, AR), and N,N-dimethylformamide (DMF, ≥99.0%) were acquired from Sinopharm Chemical Reagent Co., Ltd. Manganese chloride tetrahydrate (MnCl$_2$·4H$_2$O, 99.99%), cobalt chloride hexahydrate (CoCl$_2$·6H$_2$O, AR), anhydrous copper chloride (CuCl$_2$, 98%), commercial Pt/C (20 wt%), ruthenium oxide (RuO$_2$), iridium dioxide (IrO$_2$), and potassium hydroxide (KOH, GR, 95%) were purchased from Shanghai Macklin Biochemical Co., Ltd. Chromium chloride hexahydrate (CrCl$_3$·6H$_2$O, 99%) was acquired from Beijing Innochem Science & Technology Co., Ltd. Ruthenium chloride trihydrate (RuCl$_3$·3H$_2$O, 95%) was provided by Bide Pharmatech Ltd. Polyacrylonitrile (PAN, M$_w$ = 1.49 × 10$^5$, copolymerized with 10 wt% acrylate) was supplied by Sinopec Shanghai Petrochemical Co., Ltd. Nafion117 solution (5 wt%) was obtained from Shanghai Aladdin Biochemical Technology Co., Ltd.

**Synthesis of FeCoNiMnRu HEA nanoparticles supported on CNFs**. Typically, 0.5 mmol of MnCl$_2$·4H$_2$O, 0.5 mmol of FeCl$_3$, 0.5 mmol of CoCl$_2$·6H$_2$O, 0.5 mmol of NiCl$_2$·6H$_2$O, 0.5 mmol of RuCl$_3$·3H$_2$O and 1.5 g of PAN were dissolved in 22 g of dimethyl formamide (DMF). Homogeneous metal salts/PAN polymer solution was acquired after stirred by magnetic stirring apparatus for 8 h at room temperature. Afterwards, the syringes with stainless needles were filled with the as-prepared precursor solution and assembled to the electrospinning machine (YFSP-T, Tianjin Yunfan Technology Co., Ltd.) with an anode voltage of 20 kV, an injection rate of 0.3 mL/h, and a distance between the collector and needle of 18 cm. The as-prepared MnFeCoNiRu/PAN precursor nanofibers were put into the heating section of the home-built chemical vapor deposition (CVD) furnace. For the pre-oxidation process, the membranes were heated to 230 °C in air with a heating rate of 2 °C min$^{-1}$ and maintained for 3 h. Then, the furnace was heated to 1000 °C under Ar atmosphere with a heating rate of 2 °C min$^{-1}$ and maintained for 3 h. Finally, the as-synthesized FeCoNiMnRu HEA/CNFs was obtained after the furnace cooling down to room temperature under Ar atmosphere.

**Synthesis of metal alloy and HEA nanoparticles supported on CNFs**. Ru/CNFs, FeCoNi/CNFs, FeCoNiMn/CNFs, FeCoNiRu/CNFs, FeCoNiCrRu/CNFs, and FeCoNiCuRu/CNFs were also synthesized through the same processes with those of FeCoNiMnRu/CNFs. The precursor solutions of above control samples contained 0.5 mmol of each Fe, Co, Ni, Cu, Mn, Ru, Cr metal salts, 1.5 g PAN, and 22 g DMF. FeCoNiMnRu$_{0.5}$/CNFs and FeCoNiMnRu$_2$/CNFs were prepared by halving and doubling the amount of RuCl$_3$·3H$_2$O. FeCoNiMnRu/CNFs synthesized at 800 and 900 °C were also synthesized to unveil the thermodynamically driven phase transition process of HEA NPs.

**Materials characterizations**. The field-emission scanning electron microscope (FE-SEM, HITACHI S-4800) at an acceleration voltage of 3 kV was applied to collect the FE-SEM images. The transmission electron microscope (TEM, JEM-2100 plus) at an acceleration voltage of 200 kV was used to record the TEM images. Bright field and high-angle annular dark field scanning transmission electron microscopy (STEM) images, energy dispersive X-ray spectroscopy (EDX) mapping images, and line-scan EDX spectra were characterized by a Tecnai G2 F30S-Twin, Philips-FEI at an acceleration voltage of 300 kV. Data of inductively coupled plasma-optical emission spectrometry (ICP-OES) were acquired by Agilent 720ES. X-ray diffraction (XRD) patterns were taken via a Smartlab 9kw advance powder X-ray Cu K$_\alpha$ radiation diffractometer (λ = 1.5406 Å) in the 2θ range of 20~80° at the scanning rate of 0.5 or 10° min$^{-1}$, with the Cu K$_\alpha$ source operating at 40 kV and 40 mA. X-ray photoelectron (XPS) spectra were acquired by Thermo Scientific K-Alpha with the Al (mono) K$_\alpha$ source (1486.6 eV) operating at 12 kV and 6 mA. The binding energies were calibrated with C 1s (284.8 eV) as the standard. The BL08U1-A at the Shanghai Synchrotron Radiation Facility (SSRF) operated at 500 eV with injection currents of 100 mA was used to obtain the Co K-edge and Ru K-edge X-ray absorption near edge structure (XANES) spectra, with radiation monochromatized by a Si (111) double-crystal monochromator. Metal Co, Ru, CoO, Co$_3$O$_4$, and RuO$_2$ were taken as controls. The operando electrochemical Raman test was carried out in a round home-made electrolyzer. Ag/AgCl electrode and Pt wire served as the RE and CE, respectively. The prepared electrocatalysts were dispersed on the glass carbon electrode (GCE). The electrochemical processes were conducted in 1.0 M KOH saturated with Ar and controlled by a CHI660E electrochemical workstation. Raman spectrometer (inVia) used the laser wavelength of 785 nm.

**Electrochemical Characterization**. Electrochemical measurements were all conducted in a typical three-electrode system at 25 °C in 1.0 M KOH with a Autolab electrochemical workstation. Saturated calomel electrode (SCE) and graphite rod were used as reference electrode (RE) and count electrode (CE), respectively. The SCE was calibrated before each test. The self-supported CNFs-based materials were cut into 1 × 1 cm$^{-2}$ and served as working electrode (WE). Potentials were converted to the reversible hydrogen electrode (RHE) by the equation E$_{RHE}$ = E$_{SCE}$ + 0.244 + 0.059 × pH. Pt/C (20 wt%), RuO$_2$ and IrO$_2$ powder were taken as controls and deposited on glassy carbon electrode (GCE) with diameter of 3 mm for measurement. To prepare the electrocatalyst ink, 3 mg of electrocatalyst was dispersed into 1 mL mixed solvent with a volume ratio of V$_{isopropanol}$: V$_{water}$ = 3:1. After 30 min of ultrasonication, 25 μL Nafion117 solution was added. After another 30 min for ultrasonication, 5 μL electrocatalyst ink was casted on GCE and dried in the air naturally. All linear sweep voltammetry (LSV) curves were obtained at a scan rate of 2 mV s$^{-1}$ with 95% iR-compensation. Tafel plots were gained according to the Tafel equation:

$$\eta = a + b \log j \qquad (1)$$

where $\eta$, $b$, and $j$ represent the overpotential, Tafel slope, and current density, respectively. Electrochemical double layer capacitances (C$_{dl}$) were measured by analyzing the cyclic voltammetry (CV) curves at scan rate of 10, 20, 30, 40, and 50 mV s$^{-1}$ in the range of –0.78 to –1.00 V vs. SCE. Plotting the $\Delta i/2$ ($\Delta i = i_a - i_c$, where $i_a$ and $i_c$ represent the positive and negative current, respectively) at –0.89 V vs. SCE against the scan rate ($v$), the C$_{dl}$ can be calculated by the equation: C$_{dl}$ = $\Delta i$/2 $v$. Electrochemical active surface area (ECSA) was estimated by the equation: ECSA = C$_{dl}$/C$_s$, where the specific capacitance value (C$_s$) was taken 0.04 mF cm$^{-2}$. The ECSA-normalized LSV curves were acquired by the equation: $j_{ECSA}$ = i/ECSA, where $j_{ECSA}$ and $i$ is the current density normalized to ECSA and current of the working electrode, respectively. 10,000 CV cycles in the range of –0.4 to 0.05 V vs. RHE at a scan rate of 100 mV s$^{-1}$ and chronoamperometry were performed to evaluate the durability of samples. In the course of the chronoamperometry test, 10 min pauses were inset into the electrolysis process with intervals of about 10 h operating. Meanwhile, the electrolyte was renewed and the RE was calibrated during the pauses. For the comparison of mass activities and eliminating the difference in electrode structure, Pt/C, RuO$_2$, IrO$_2$, and FeCoNiMnRu/CNFs powder were dropped onto GCE, and the calculation process is shown as follow: $J_{mass}$ = $J_{geo}$ × 0.07069 cm$^2$/m, m = 3 mg × ω × (5 μL/1000 μL), where m and ω are the mass loading and mass fraction of noble metal (Ru, Pt, or Ir).

**Computational methods**. We have employed the first-principles[40,41] to perform all Spin-polarization density functional theory (DFT) calculations within the generalized gradient approximation (GGA) using the Perdew-Burke-Ernzerhof

(PBE)[42] formulation. vdW corrections was added. We have chosen the projected augmented wave (PAW) potentials[43,44] to describe the ionic cores and take valence electrons into account using a plane wave basis set with a kinetic energy cutoff of 400 eV. Partial occupancies of the Kohn−Sham orbitals were allowed using the Gaussian smearing method and a width of 0.05 eV. The electronic energy was considered self-consistent when the energy change was smaller than $10^{-6}$ eV. A geometry optimization was considered convergent when the energy change was smaller than 0.04 eV Å$^{-1}$. The vacuum spacing in a direction perpendicular to the plane of the structure is 15 Å. The Brillouin zone integration is performed using 3 × 3 × 1 Monkhorst-Pack k-point sampling for a structure. Finally, the adsorption energies ($E_{ads}$) were calculated as Eq. (2):

$$E_{ads} = E_{ad/sub} - E_{ad} - E_{sub} \qquad (2)$$

where $E_{ad/sub}$, $E_{ad}$, and $E_{sub}$ are the total energies of the optimized adsorbate/substrate system, the adsorbate in the structure, and the clean substrate, respectively. The free energy was calculated using the Eq. (3):

$$G = E + ZPE - TS \qquad (3)$$

where G, E, ZPE, and TS are the free energy, total energy from DFT calculations, zero point energy, and entropic contributions, respectively. The ZPE has been calculation using the 6 × 2 × 2 Monkhorst-Pack k-point and DFT + U correction has been used in our systems. In our alloy structure, the FCC structure has been obtained with the 80 atoms.

## Data availability

Source data are provided with this paper. The data used in this study are presented in the text and Supplementary Information. Additional data and information are available from the corresponding author upon reasonable request. Additionally, data reported herein have been deposited in the Figshare database, and are accessible through https://figshare.com/articles/figure/Source_Data_zip/19513684. Source data are provided with this paper.

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

## Acknowledgements

This study is supported by the National Natural Science Foundation of China (NSFC) (Grant nos. 52073124, 51803077), Natural Science Foundation of Jiangsu Province (Grant nos. BK20180627), Postdoctoral Science Foundation of China (2018M630517, 2019T120389), the MOE & SAFEA, 111 Project (B13025), and the Fundamental Research Funds for the Central Universities. We also thank the characterizations supported by Central Laboratory, School of Chemical and Material Engineering, Jiangnan University.

## Author contributions

H.Z. conceived and supervised the research. H.Z, J.H., and Z. Z. designed the experiments. J. H. performed most of the experiments and data analysis. H.Z., C. W., G. G., F. L., and T. L. performed the DFT calculations and mechanistic analysis. J.H., S.L., P. M., W. D., and F. L. prepared the electrodes and helped with electrochemical measurements. K. C. conducted and analyzed HRTEM micrographs and mapping images. J. H., Z. Z., M. D., and H. Z. wrote the manuscript. All authors discussed the results and commented on the manuscript.

## Competing interests

The authors declare no competing interests.
