## [Peer Review File · Nature Communications]

Title: Unraveling the electronegativity-dominated intermediate adsorption on high-entropy alloy electrocatalystsReviewers' comments:

Reviewer #1 (Remarks to the Author):

This manuscript presented the synthesis of HEA catalysts of FeCoNiXRu (X: Cu, Cr, and Mn) in the matrix of electrospun polymer nanofibers and the characterization of performance for water splitting. I acknowledge they systematically investigated HEA catalysts, but the novelty is still lacking. In fact, I am not so impressed with this study.

The comments are as follows:

1. The author's description of the average size of the HEA particles (in line 105 to line 107) seems ambiguous. It is best to make statistics on a certain number of particles and give a distribution diagram of particle size.
2. The authors proposed a thermodynamically driven phase transition mechanism, and concluded the formation of single phase and uniform composition from the in situ X-ray diffraction (XRD) results. However, the accuracy of XRD is relatively low. Therefore, EDX mappings of 5 nm particles from wider area are suggested to further support the conclusions.
3. In line 116, The "Rh in a single FeCoNiMnRu HEA NP...." should be "Ru in a single FeCoNiMnRu HEA NP...."; Please check the sentence in line 305-307 "The water dissociation into H* and OH* and H* adsorption are potential-determining steps (PDS), which determine the water dissociation rates.", there are two "H*" may be repeated; In line 394, the "H*" should be "H*(UPPER)", besides, in the whole manuscript, the * marks sometimes is big (in line 301, 303, 305, ...), but sometimes is small (in line 326, 348, ...). Please check carefully to avoid such errors.
4. Mass activity is an important parameter for evaluating noble metal-based catalysts. Therefore, please give a comparison of mass activity between the HEAs and commercial Pt/C catalysts.
5. In the manuscript, the overpotential of FeCoNiMnRu/CNF catalyst at current density of 100 mA cm⁻² for OER were mentioned three times (line 260, 372, and in Fig. 6c), but the three values are different (308, 318, 309 mV).
6. The OER stability of FeCoNiMnRu/CNF catalyst was not discussed.
7. Author claims the Volmer–Heyrovsky reaction pathway according to the Tafel slope result. According to Volmer–Heyrovsky pathway, the formed H* intermediate at stage 4 will be detached from the surfaces by combining with released H⁺ ions or a H₂O molecular in the electrolyte. However, the description of the reaction in the manuscript (line 303-305, and line 359) is different (that is Volmer–Tafel pathway) Please carefully check and ensure the accuracy of the description.

8. The synergistic effect and rearranged electronic structure is believed to be the main origin of activity of a multi-element catalyst. However, the accurate identification of the active centers of a HEA catalyst remains challenge, owing to the numerous possibilities of element combinations and atomic configurations. Please explain the rationality of the atomic configuration applied in the manuscript. In fact, a plenty of combinations of atomic configuration exist. Also, the surface is usually curved (high-indexed surface) and the some metallic species (Mn, Fe, Co and Ni) are oxidized with surface oxygen (XPS data) but the effects are not considered at all.

9. Some recent references on the development of HEA catalysts for electrocatalytic reactions, such as ACS Mater. Lett. 2019, 1, 5, 526–533; J. Catal. 2020, 383,164–171; Chem. Mater. 2021, 33, 5, 1771–1780; Chem. Sci., 2021, 12, 11306-11315. can be mentioned and cited in the introduction to give the readers a broader view of this field.

10. Provide all ICP analysis data of FeCoNiMnRu, FeCoNiCrRu, FeCoNiCuRu. The high activity may be explained by the content of Ru, not electronegativity.

Reviewer #2 (Remarks to the Author):

The authors report a study on the function of high-entropy alloys (HEAs) as catalysts for water dissociation and the correlation of the performance with the electronegativity of the included elements. The background and motivation for the study are well described, elemental composition analysis, XPS and Raman are well-performed and reported, but there are some serious shortcoming in the analysis and conclusions. The authors “propose a conceptual and experimental approach to overcome the limitations of single-element catalysts by designing a unique FeCoNiXRu (X: Cu, Cr, and Mn) HEA system with two kinds of active sites that have different adsorption capacities for multiple intermediates.” (line 68-71) and “and successfully identified electronegativity-dependent preferences for active site absorption of the intermediates” (line 74-76). This is not shown from the data.

1. The authors ‘notion of electronegativity is very simplified and needs to be expanded. First, the Pauling electronegativity is defined from the ability to attract electrons when participating in a covalent bond. As the metals have metallic bonds, the notion of Pauling electronegativity is ill-defined. It is instead more common to use d-band center descriptions or other notions to quantify the effect of charge transfer in the system. The first and most simple case presented by Pauling 1932 with e.g. an electronegativity of 2.20 for Ru and Ni (1.91), Co (1.88), Fe (1.83), and Mn (1.55) is since then extended with e.g. Mullikan, Sanderson electronegativity, where for example Ru has an electronegativity of 1.54 in comparison to 1.80-1.88 for Fe, Co, Ni. One can note that the charge transfer with respect to Ru and surrounding elements would then be the opposite of what the authors describe if a more elaborated electronegativity scale is used.

2. The electronegativity given by the authors, Ru (2.20) and Ni (1.91), Co (1.88), Fe (1.83), and Mn (1.55),

are referred to be found in ref 28 (line 203-204), where they cannot be found, (or in ref 42 given in ref 28, or ref 6c, given in ref 42 in ref 28). From the numbers, however, it is apparent that they are from the Pauling electronegativity scale.

3. The electrochemical performance need to be clarified. In Fig 4a, Pt/C show very bad performance while it instead have outstanding performance in other publications. For a fair comparison, the same approach and optimization need to be done for all electrodes (same for Fig 4d). In addition, the large change is from inclusion of Ru, while the difference in-between the two Ru-containing electrocatalysts need to be verified to be from the claimed electronic effect, not from suppression of oxidation by inclusion of Ru or increased surface area due to intermixing of an additional element. From the electrochemical surface areas (ECSAs) normalized reported by the authors at line and Fig 4c, it seem to be a remained improvement, but from the values at line 253 with a Cdl value for FeCoNiMnRu/CNFs of 349.3 mF cm⁻², compared to the significantly lower value for e.g. FeCoNiMn/CNFs (274.5 mF cm⁻²) and the small shift in curves in the un-normalized curves Fig 4a, it does not make sense.

4. The calculations does not seem to include variation in mixing in the high entropy alloys. The effect of different elements at different sites around the active site will naturally change the adsorption energy. This need to be done and is a quite more demanding efforts than the calculations reported in the manuscript.

5. The simple notion of Pauling electronegativity is not advised to use as a descriptor and put on the same scale of accuracy as DFT calculations, here at least Mulliken or Allen electronegativity scale should be used. As the authors have performed DFT calculations, analysis of the charge density is preferred over the notion of electronegativity where e.g. Bader analysis and will give the charge distribution and possible charge transfer from first-principles directly. That is similar to the analysis reported for just one composition in SI Fig 22, but can instead be used to show the charge transfer in-between the chosen compositions in their materials (using fully randomized structures).

6. The PBE functional is not suitable for calculation of correct adsorption energies for the hydrogen evolution reaction, here vdW corrections are needed, and the BEEF + vdW is known to be the approach that correlate with experiments (Wallendorff et al Surface Science 640 (2015): 36-44. Sharada et al. Physical Review B 100.3 (2019): 035439.).

I suggest that the authors update their manuscript with the above considerations and submit to a more technical journal. I cannot recommend publication at this stage.

Reviewer #3 (Remarks to the Author):

In this work, the authors applied a FeCoNiXRu (X: Cu, Cr, and Mn) high-entropy alloy (HEA) system as an alkaline water electrolysis catalyst. The developed FeCoNiCu HEA electrocatalyst showed improved

intrinsic electrochemical performance through transformation from a multiphase to a single-phase HEA with charge redistribution of multi-components and optimized energy barrier of Co and Ru sites, which were attributed to electronegativity differences between mixed elements in HEA (HER overpotential: 82 mV at 100 mA cm⁻², OER overpotential: 308 mV at 100 mA cm⁻², full cell overpotential: 1.65 V at 100 mV cm⁻², half cell cycle stability: 10,000 cycles, full cell chronoamperometric stability: 600 h at 1 A cm⁻²). The demonstrated HER/OER and full cell performance of the proposed material system is noticeable, however, the applied multi-metal in this work has already been introduced in various previous HEA electrocatalyst system. Moreover, the key mechanism for the electrochemical performance enhancement of the developed catalyst does not seem to be distinguishable from those of previous studies (e.g., transformation to single-phase, optimized electronic structure via electron transfer, reduced energy barrier in rate-determining step, etc.). Therefore, the reviewer concludes that this work is not suitable for the wide readership of Nature Communications. Below are some of the issues that should be considered before submitting this work elsewhere.

1. In this work, the origin of enhanced hydrogen/oxygen generation performance in FeCoNiXRu (X: Cu, Cr, and Mn) HEA is attributed to optimized electronic structures of the alloy via combination of high-entropy alloying with multiple elements. However, numerous previous studies have already reported that HEA electrocatalysts comprising similar multi-components (including Fe, Co, Ni, Cu, Cr, and Mn) with synergetic effects can enhance the HER and OER kinetics (Electrochim. Acta, 2021, 378, 138142; Chem. Eng. J., 2021, 425, 131533; J. Mater. Chem. A, 2021, 9, 889-893; Electrochim. Acta, 2018, 279, 19-23; Adv. Mater. Interfaces, 2019, 6(7), 1900015; Nat. Commun., 2020, 11, 2016: <https://doi.org/10.1038/s41467-020-15934-1>; Chem. Sci., 2020, 11, 12731). The authors should clearly demonstrate and discuss the advancement of this work from previous studies.
2. The ICP-OES analysis showed that the molar ratio of Fe:Co:Ni:Mn:Ru was 1:1.01:1.01:0.75:0.75. However, to validate whether the optimal HER/OER performance is obtained from the above molar ratio (1:1.01:1.01:0.75:0.75), the HER/OER performance (overpotential, Tafel slope, EIS, ECSA, etc.) of other control samples with various molar ratios should be provided.
3. Specific surface area of the water electrolysis catalyst is directly related to the water electrolysis performance because it is correlated to the active site, which, however, is missing in this study. The authors are recommended to quantitatively compare the specific surface area of the materials used in the study.
4. For the real-time in situ XRD analysis in Figure 2f-h, a positive shift of (111) plane of FeCoNiMnRu HEA with annealing treatment at 1000 °C for 3 h compared to that prepared at 800-1000 °C was observed, which was attributed to the reduced lattice distortion in the HEA. The authors claimed that this result corroborated the single-phase formation of FeCoNiMnRu HEA. In addition to the (111) plane, additional diffraction peaks of (200) and (220) planes can also be found in FeCoNiMnRu. To better elucidate the single-phase formation of FeCoNiMnRu HEA with prolonged annealing treatment at 1000 °C, the correlation between the peak shift of (200)/(220) planes and thermodynamically driven phase transition should also be discussed.

5. The electronic interaction in the proposed HEA is attributed to the difference in electronegativity between the constituent elements, which was supported by the XPS analysis. However, current XPS analysis on the water electrolysis performance can be further elaborated by providing more in-depth analysis via comparing FeCoNiMnRu HEA with different control samples of FeCoNi, FeCoNiMn, FeCoNiRu. For example, in Figure 3c, the $\text{Co}^{3+}/\text{Co}^{2+}$ ratio can suggest more detailed information for the OER kinetics of the electrocatalyst (Electrochimica Acta, 2017, 254: 14-24). Also, in Figure 3f, O 1s peak is deconvoluted into hydroxyl group and residual oxygen containing group only. Here, the O 1s peak can also be deconvoluted to lattice oxygen, highly oxidative oxygen, surface-active oxygen, and adsorbed water, and the OER performance can be discussed in terms of the surface-active oxygen/lattice oxygen ratio (Electrochimica Acta, 2017, 254: 14-24).

6. In Figure 3, the authors analyzed that Ru has a low oxidation state resulting from the electron transfer. It is well known that oxidation state of active metals, which can modulate the electronic structure of the electrocatalyst, directly affects the water electrolysis performance. Therefore, to improve the reliability of the oxidation state information for the elements analyzed via XPS, other means of analysis such as XANES should be accompanied.

7. In Supplementary Figure 11, the authors noted that FeCoNiMnRu HEA has the lowest R_{ct} compared to the control samples through EIS analysis, which was also used to support the following statement, “The low R_{ct} values of HEAs suggest that high-entropy alloying with multiple elements could allow optimization of the electronic structures of the alloy”. This appears to be rather abstract for explaining the “optimized electronic structure” since R_{ct} only indicates the interfacial resistance between the electrolyte and the electrode.

8. In Supplementary Figure 12, based on Cdl of electrocatalysts, ECSA of control samples and FeCoNiMnRu HEA were inferred. However, actual ECSA values are missing. The ECSA data of the FeCoNiMnRu and control samples should be specifically provided.

9. In Figure 4f, the full cell polarization curve of the noble metal couple (Pt/C || Ru/C) should be added when discussing the performance improvement of FeCoNiMnRu HEA over the Pt/C || Ru/C.

10. In Figure 4g and Supplementary Figure 15, the authors claim the operational stability of FeCoNiMnRu HEA based on chronoamperometric (CA) stability (600 hours at 1 A cm^{-2}) and CV cycle stability (10000 cycles at 1 M KOH). However, to better support the reliability of the stability analysis, additional verification in various harsh environments (high temperature, high electrolyte concentration, high current density, acidic electrolyte, etc.) is recommended (Nat. Commun., 2018, 9, 2609: <https://doi.org/10.1038/s41467-018-05019-5>; Nat. Commun., 2020, 11, 5462: <https://doi.org/10.1038/s41467-020-19214-w>; Nat. Commun., 2021, 12, 4606: <https://doi.org/10.1038/s41467-021-24829-8>).

11. After 600 hours of CA stability test, the authors verified the stability of the electrode via XRD and

STEM-EDS mapping analysis. In addition to the structural and compositional analysis, the status of chemical state analysis for the constituent metals after the stability test is also recommended, e.g., change in the oxidation state analyzed by XPS and XANES.

12. FeCoNiMnRu HEA showed full cell performance that outperformed the noble-metal couple (Pt/C || Ru/C). To better elucidate the water electrolysis performance of FeCoNiMnRu HEA, please calculate and provide the energy efficiency of FeCoNiMnRu (Int. J. Hydrog. Energy, 2010, 35(20), 10851-10858).

Reviewer #4 (Remarks to the Author):

I was asked to review the submitted manuscript “Unraveling the interactions between active sites and intermediates in high-entropy alloy electrocatalysts: How electronegativity matters”, written by Hao & Zhuang et al. and submitted to Nature Communications.

The manuscript is very well written and gives a very comprehensive insight into catalytic procedures of high entropy alloys. The authors produced a network of carbon nanofibers, partially coated with HEA particles and used this compound as catalyst for HER and OER reactions. Additionally, they used operando and computational methods to explain the behavior of the material and drew a connection between electronegativity and electrochemical performance of the individual incorporated elements. Without a doubt, I recommend a publication in Nature Communications after some revisions.

1. Since the surface of the catalyst is the most important “part” of the particle when it comes to catalytic reactions, the authors should explain why they are preparing a CNF/HEA network instead using pure HEA particles. The incorporation of the HEAs in the CNF network reduces the addressable surface of the HEAs, since they are attached (or even fully included sometimes?) to the CNF as shown in the SEM micrographs.

2. When using such a compound, how do the authors determine the catalytic activity? Is the activity normalized to a measured surface, or the mass of the complete material or the mass of the HEA particles?

3. Is the comparison of Figure 4e “fair”? When I understood the experimental part correctly, the authors are using a free standing electrode and compare it to mostly tape casted electrodes. What are the differences in surface between the tape casted and their free standing electrodes? I think that more experimental details about the electrode preparation should be given. Is it possible to compare an “architected” electrode with a nice 3D carbon network to a 2D tape casted electrode?

4. A very important part is the materials characterization as well. In my opinion, the XRD part needs some improvements.

- In both pattern for FeCoNiMnRu (1000°C, 3h) a strong asymmetry of the 111 reflection is observable. Such shoulders often can indicate byphases, also the 200 reflection reveals a very small shoulder. Where do these shoulders come from?

- Between FeCoNiMnRu (1000°C, 3h) and FeCoNiMnRu huge shifts of the reflections can be seen. The authors argue with the introduction of the Mn₃Co₇ phase into the single phase structure. However, as can be seen, even the Mn₃Co₇ phase already seems to be a high entropy structure, since the reflections

are shifted as well. Therefore, can such a huge shift when one high entropy structure with mixed elements is introduced into another one with mixed elements, really be explained using this argument? Most probably all elements are apparent in both phases already, so an incorporation would not really lead to a change of the lattice parameters.

- Even if the incorporation of the Mn_3Co_7 phase leads to a shrinking of the lattice parameters, how do the authors explain the extension of the lattice parameter (between 900 and 1000 °C), although the intensity of the Mn_3Co_7 reflection is shrinking? This would mean that the Mn_3Co_7 phase is vanishing, most probably incorporated as the authors state, and the lattice parameters are growing. This is in contrast to the next step when the same incorporation should lead to the explained lattice shrinking. Therefore, I would strongly recommend doing detailed Rietveld refinements and HR XRD measurements to identify the mechanism and to rule out that additional phases are in the compound.

Response to the Reviewers' comments

Reviewer #1 (Remarks to the Author):

This manuscript presented the synthesis of HEA catalysts of FeCoNiXRu (X: Cu, Cr, and Mn) in the matrix of electrospun polymer nanofibers and the characterization of performance for water splitting. I acknowledge they systematically investigated HEA catalysts, but the novelty is still lacking. In fact, I am not so impressed with this study.

Reply: We are thankful to the reviewer for the kind recommendation and careful review. (All the changes are labeled in red). We have provided more evidence to support the novelty of our manuscript. The application of HEA nano-materials in the field of electrocatalysis has been demonstrated to be viable in recently 3 years (Y. Yao, et al., *Science* 2018, 359, 1489-1494). The HEA nano-material is still in its initial stage as a kind of novel electrocatalyst, while the current researches mainly focused on the synthesis methods and component screening (Y. Yao, et al., *Sci. Adv.*, 2020, 6, eaaz0510; M. W. Glasscott, et al., *Nat. Commun.* 2019, 10, 2650). In regard to HEA nano-materials as electrocatalysts for water splitting (Z. Jia, et al., *Adv. Mater.* 2020, 32, 2000385; H. Qiu, et al., *J. Mater. Chem. A*, 2019, 7, 6499-6506. Z. Jin, et al., *Small*, 2019, 15, 1904180), only preliminary electrochemical tests have been conducted and the origin of the obtained excellent performance were simply ascribed to the "synergistic effect" derived from multicomponent. Therefore, electrocatalytic mechanism on HER are still extremely rare and discussions on the mechanism and the origin of the excellent activities are significantly lack. Meanwhile, the configurational complexity in HEA makes the identification of exact active sites difficult and complicated. How do the multi-element HEA work, how do the different metal sites in HEA adsorb species during electrolysis and what the roles do metal sites play? A lot of questions should be considered and therefore, in-depth investigations on the mechanism understanding and performance regulation are urgent need. In this work, we have elaborately investigated the electrocatalytic mechanism both experimentally and theoretically. We are the first example to unraveling the roles of each metal sites in HEA for electrocatalytic water splitting and correlate the electronegativity with activity. The electronegativity differences between mixed elements in HEA induce significant charge redistribution and create highly active Co and Ru sites with optimized energy barriers for simultaneously stabilizing OH* and H* intermediates, which greatly enhances the efficiency of water dissociation in alkaline conditions. Adjustment of the electrocatalytic activities of HEA catalysts were further shown experimentally and theoretically by tailoring the electronegativities of the compositions. This work provides an in-depth understanding of the interactions between specific active sites and intermediates, which opens up a new direction for breaking scaling relation issues for multistep reactions.

The comments are as follows:

1. The author's description of the average size of the HEA particles (in line 105 to line 107) seems ambiguous. It is best to make statistics on a certain number of particles and give a distribution diagram of particle size.

Reply: We are thankful to the reviewer for the kind recommendation and careful review. The authors have added the distribution diagram of particle size in Supporting Information. As shown

in Supplementary Fig. 2, the average diameter of the HEA NPs is approximately 14.2 ± 9.1 nm.

Supplementary Fig. 2. Average diameters of the FeCoNiMnRu HEA NPs supported on CNFs.

2. The authors proposed a thermodynamically driven phase transition mechanism, and concluded the formation of single phase and uniform composition from the in situ X-ray diffraction (XRD) results. However, the accuracy of XRD is relatively low. Therefore, EDX mappings of 5 nm particles from wider area are suggested to further support the conclusions.

Reply: We are thankful to the reviewer for the kind suggestion. According to your advice, authors have performed the STEM-EDX mapping images of 5 nm HEA NPs from wider area (scale bar is 10 nm). The mapping area contains 10 HEA NPs (Supplementary Fig. 5) and the line-scan STEM-EDX of 3 HEA NPs (Supplementary Fig. 6) have also been conducted. The results exhibit the uniformly distribution of Fe, Co, Ni, Mn and Ru elements among all the HEA NPs, further suggesting the repeatability of HEA NPs with uniform composition distribution. For the XRD accuracy, authors have further performed the high resolution XRD (HRXRD) with a scan rate of 0.5° /min. In addition, we have also performed the XRD detailed Rietveld refinement. The crystal structure of FeCoNiMnRu NPs obtained at and 1000 °C-3h (Figure 2f) were further analyzed by detailed Rietveld refinements. The results confirmed that the FeCoNiMnRu HEA NPs prepared at 1000 °C (3h) is single phase structure with similar component and the simulative components of FeCoNiMnRu NPs models matches well with the ICP results.

Supplementary Fig. 5. STEM-EDS mapping of the HEA NPs supported on CNFs.

Supplementary Fig. 6. Line-scan STEM-EDX spectra of the HEA NPs supported on CNFs.

Figure 2f. XRD pattern of FeCoNiMnRu/CNFs (1000 °C-3 h) with a scan rate of 0.5° /min and the corresponding XRD Rietveld refinement.

3. In line 116, The “Rh in a single FeCoNiMnRu HEA NP...” should be “Ru in a single FeCoNiMnRu HEA NP...”; Please check the sentence in line 305-307 “The water dissociation into H* and OH* and H* adsorption are potential-determining steps (PDS), which determine the water dissociation rates.”, there are two “H*” may be repeated; In line 394, the “H*” should be “H*(UPPER)”, besides, in the whole manuscript, the * marks sometimes is big (in line301, 303, 305, ...), but sometimes is small (in line326, 348, ...). Please check carefully to avoid such errors.

Reply: We are thankful to the reviewer for the careful review. According to your advice, author have carefully revised all the mistakes in the revised manuscript.

4. Mass activity is an important parameter for evaluating noble metal-based catalysts. Therefore, please give a comparison of mass activity between the HEAs and commercial Pt/C catalysts.

Reply: We are thankful to the reviewer for the kind suggestion. According to your advice, author have added the comparison of mass activity between the HEAs and commercial Pt/C catalysts in the revised manuscript. For a fair comparison, the FeCoNiMnRu/CNF, commercial Pt/C, and RuO₂ were all dropped on GCE to evaluate their activities. As shown in Supplementary Fig. 23, the FeCoNiMnRu/CNFs exhibit higher mass activity than the Pt/C catalysts for HER and RuO₂ catalysts for OER at high overpotential, which is benefit for the practical application.

Supplementary Fig. 23. The mass activities of (a) FeCoNiMnRu/CNFs and 20 wt% Pt/C for HER at overpotentials of 200 and 300 mV. (b) FeCoNiMnRu/CNFs and RuO₂ for OER at overpotentials of 400 and 500 mV.

5. In the manuscript, the overpotential of FeCoNiMnRu/CNF catalyst at current density of 100 mA cm⁻² for OER were mentioned three times (line 260, 372, and in Fig. 6c), but the three values are different (308, 318, 309 mV).

Reply: We are thankful to the reviewer for the careful review. According to your advice, author have revised the mistakes. The overpotential of FeCoNiMnRu/CNF catalyst at current density of 100 mA cm⁻² for OER is 308 mV.

6. The OER stability of FeCoNiMnRu/CNF catalyst was not discussed.

Reply: We are thankful to the reviewer for the kind suggestion. According to your advice, author have performed the OER stability of FeCoNiMnRu/CNF catalyst. The OER stability of FeCoNiMnRu/CNFs was conducted at 1.55 V vs. RHE for 10 h (Supplementary Fig. 27). The current density showed no obvious decay during the test, and furthermore, the LSV curves before and after stability display negligible change.

Supplementary Fig. 27. The chronoamperometric curve for FeCoNiMnRu/CNFs measured at 1.55 V vs. RHE for 10 h. Inset shows the LSV curves of FeCoNiMnRu/CNFs before and after stability test.

7. Author claims the Volmer–Heyrovsky reaction pathway according to the tafel slope result. According to Volmer–Heyrovsky pathway, the formed H* intermediate at stage 4 will be detached from the surfaces by combining with released H⁺ ions or a H₂O molecular in the electrolyte. However, the description of the reaction in the manuscript (line 303-305, and line 359) is different

(that is Volmer–Tafel pathway) Please carefully check and ensure the accuracy of the description.

Reply: We are thankful to the reviewer for the careful review. According to your advice, author have revised the mistakes.

8. The synergistic effect and rearranged electronic structure is believed to be the main origin of activity of a multi-element catalyst. However, the accurate identification of the active centers of a HEA catalyst remains challenge, owing to the numerous possibilities of element combinations and atomic configurations. Please explain the rationality of the atomic configuration applied in the manuscript. In fact, a plenty of combinations of atomic configuration exist. Also, the surface is usually curved (high-indexed surface) and the some metallic species (Mn, Fe, Co and Ni) are oxidized with surface oxygen (XPS data) but the effects are not considered at all.

Reply: We are thankful to the reviewer for the kind suggestion. Considering that the difference in atomic configurations may change the adsorption energy, the hydrogen adsorbing energy (ΔG_{H^*}) on Fe-, Co-, Ni-, Ru-sites in FeCoNiXRu (X = Mn, Cr, Cu) with 8 randomly selected configurations were analyzed by DFT calculation. The corresponding atomic configurations and ΔG_{H^*} were shown in Supplementary Fig. 38-40, and Fig. 55 and the values of ΔG_{H^*} were summarized in Supplementary Table 9-11. DFT Results suggest that the all of the Ru sites in FeCoNiMnRu, FeCoNiCuRu and FeCoNiCrRu display the lowest values than Fe, Co, and Ni sites, demonstrating that the atomic configurations could not affect the ΔG_{H^*} orders on Fe, Co, Ni and Ru sites in different HEA NPs (Supplementary Fig. 55). In addition, the Ru sites in FeCoNiMnRu also exhibit the lowest ΔG_{H^*} than those Ru sites in FeCoNiCrRu and FeCoNiCuMn HEA, further indicating that the fifth metal (Mn, Cr and Cu) with low electronegativity in HEA could reduce the ΔG_{H^*} .

We used the X-ray absorption spectroscopy (XAS) to investigate the structure of FeCoNiMnRu/CNFs. The X-ray absorption near-edge structure (XANES) results (Fig. 3a-d) indicate that the pre-absorption edge features for Co and the absorption edge for Ru both are metallicity by comparing with reference metal Co, CoO and Co₃O₄ foils, demonstrating that the Co and Ru elements in FeCoNiMnRu HEA NPs are in metallic states. The post-edge for Co and Ru in HEA exhibits slight deviation in the shape and intensity when compared with the reference metal Co and Ru foils. These features indicate the alloy formation rather than elemental segregation into pure metals, which would show the same length as metal Co and Ru foils. The extended X-ray absorption fine structure (EXAFS) of Co and Ru were determined through the fitting of the Fourier transform (FT) spectra. As shown in Fig. 3b and 3d, the FT-EXAFS spectra indicate that the average bond length of Ru and Ni in HEA NPs is quite different from the metallic bond in bulk Co and Ru references, suggesting that the Co and Ru elements are surrounded by different metallic species (Fe, Mn and Ni). The bond structures of Co and Ru in HEA reveal the similar average bond length without any oxidation when compared CoO and Co₃O₄ foils, further confirming the metallic states of Co and Ru in HEA after stability test. According to the EXAFS fitting (Supplementary Fig. 14), the bond length (R) and coordination numbers of each bond type in the HEA were summarized in Supplementary Table 4. The reliability of the fitting method is supported by smaller R factors. According to the bond length and coordination numbers of Co and Ru in HEA, we have chosen the atomic configuration accordingly with XAS results as represent sample to show the relationship between the ΔG_{H^*} and electronegativity.

Supplementary Figure 55. The summarized values of ΔG_{H^+} on Fe-, Co-, Ni-, and Ru-sites in (a) FeCoNiMnRu, (b) FeCoNiCrRu, and (c) FeCoNiCuRu with different atomic configurations.

Supplementary Figure 38. Different atomic configurations of FeCoNiMnRu HEA.

Supplementary Figure 39. Different atomic configurations of FeCoNiCrRu HEA.

Supplementary Figure 40. Different configurations of FeCoNiCuRu HEA.

Figure 3. (a) The Co K-edge XANES spectra and (b) FT-EXAFS spectra of FeCoNiMnRu/CNFs, Co foil, CoO, and Co₃O₄. (c) The Ru K-edge XANES spectra and (d) FT-EXAFS spectra of FeCoNiMnRu/CNFs, Ru foil, and RuO₂.

Supplementary Figure 14. WT-EXAFS spectra of (a) Co and Ru foil, (b) Co and Ru in FeCoNiMnRu/CNFs.

According to your suggestion, we further used the XAS to investigate the chemical states of FeCoNiMnRu/CNFs before and after the HER stability test. As shown in Fig. 5a-f, the pre-absorption edge features for Co and the absorption edge for Ru both are metallicity by comparing with reference metal Co, CoO and Co₃O₄ foils (Fig. 3a-d), demonstrating that the Co and Ru elements in HEA NPs are in metallic state. Meanwhile, after the stability test, the XANES of Co and Ru in HEA NPs also keep metallic state, suggesting the excellent oxidation resistance during the long-term HER stability test. The FT-EXAFS spectra (Fig. 5b and 5d) indicate that the bond structure of Co and Ru in HEA before and after stability test reveal the similar average bond length without any oxidation when compared CoO and Co₃O₄ foils, further confirming the metallic states of Co and Ru in HEA after stability test. According to the EXAFS fitting (Fig. 5c and 5f), the bond length (R) and coordination numbers of each bond type in the HEA before and after stability test were summarized in Supplementary Table 4. The FT-EXAFS and WT-EXAFS spectra (Fig. 5g-i) of Co and Ru in HEA before and after stability tests have negligible mismatch, which means that the FeCoNiMnRu HEA keep metallic states in long-term stability tests, exhibiting extraordinary durability. The reliability of the fitting method is supported by smaller R factors.

Figure 5. (a) The Co K-edge XANES spectra and (b) FT-EXAFS spectra of FeCoNiMnRu/CNFs before and after stability test, and Co foil. (c) The corresponding FT-EXAFS fitting curves of FeCoNiMnRu/CNFs before and after stability test. (d) The Ru K-edge XANES spectra and (e) FT-EXAFS spectra of FeCoNiMnRu/CNFs before and after stability test, and Ru foil. (f) The corresponding FT-EXAFS fitting curves of FeCoNiMnRu/CNFs before and after stability test. WT-EXAFS spectra of Co and Ru foil (g), Co and Ru in FeCoNiMnRu/CNFs before (h) and after stability test (i).

Supplementary Table 4. EXAFS fitting parameters at the Co and Ru K-edge for various samples.

Sample	Shell	CN^a	$R(\text{Å})^b$	$\sigma^2(\text{Å}^2)^c$	$\Delta E_0(\text{eV})^d$	R factor
Co K-edge ($S_0^2=0.811$)						
Co foil	Co-Co	12*	2.49	0.0063	6.9	0.0023
CoO	Co-O	6.0	2.10	0.0106	1.5	0.0053
	Co-Co	12.0	3.01	0.0080		

	Co-O	6.2	1.91	0.0035		
Co ₃ O ₄	Co-Co	6.1	2.86	0.0048	-8.1	0.0079
	Co-Co	7.4	3.36	0.0048		
FeCoNiMn Ru/CNFs before stability test	Co-Co	7.2	2.51	0.0066		
	Co-Ru	1.4	2.49	0.0066	4.5	0.0059
FeCoNiMn Ru/CNFs after stability test	Co-Co	6.9	2.50	0.0085		
	Co-Ru	0.3	2.52	0.0085	3.0	0.0071
Ru K-edge ($S_o^2=0.816$)						
Ru foil	Ru-Ru	12*	2.67	0.0033	6.6	0.0071
	Ru-O	6.0	1.98	0.0035		
RuO ₂	Ru-Ru	3.0	3.12	0.0045	2.4	0.0085
	Ru-Ru	11.7	3.56	0.0045		
FeCoNiMn Ru/CNFs before stability test	Ru-Co	6.7	2.52	0.0049	-3.0	0.0059

9. Some recent references on the development of HEA catalysts for electrocatalytic reactions, such as ACS Mater. Lett. 2019, 1, 5, 526–533; J. Catal. 2020, 383,164–171; Chem. Mater. 2021, 33, 5, 1771–1780; Chem. Sci., 2021, 12, 11306-11315. can be mentioned and cited in the introduction to give the readers a broader view of this field.

Reply: We are thankful to the reviewer for the kind suggestion. All the suggested literatures have been cited in the revised manuscript.

10. Provide all ICP analysis data of FeCoNiMnRu, FeCoNiCrRu, FeCoNiCuRu. The high activity may be explained by the content of Ru, not electronegativity.

Reply: We are thankful to the reviewer for the kind suggestion. The Ru contents in FeCoNiMnRu/CNFs, FeCoNiCrRu/CNFs, and FeCoNiCuRu/CNFs were measured to be 4.38, 4.07, and 4.13 wt%, which are basically consistent with each other. The metal contents of FeCoNiMnRu, FeCoNiCrRu, and FeCoNiCuRu were shown in Supplementary Table 1-3. Therefore, the differences in electrocatalytic activity of the three kinds of HEA could influenced by the electronegativity.

Supplementary Table 1. The mass loadings of Fe, Co, Ni, Mn, and Ru in FeCoNiMnRu/CNFs measured by ICP-OES and the corresponding atomic percentages.

FeCoNiMnRu	Fe	Co	Ni	Mn	Ru
Mass loading (wt. %)	2.97	3.05	2.96	1.78	4.38
Atomic percentage (at. %)	23.00	22.38	21.82	14.03	18.77

Supplementary Table 2. The mass loadings of Fe, Co, Ni, Cr, and Ru in FeCoNiCrRu/CNFs measured by ICP-OES and the corresponding atomic percentages.

FeCoNiMnRu	Fe	Co	Ni	Cr	Ru
Mass loading (wt. %)	2.57	2.82	2.67	2.68	4.07
Atomic percentage (at. %)	19.92	20.71	19.67	22.29	17.41

Supplementary Table 3. The mass loadings of Fe, Co, Ni, Cu, and Ru in FeCoNiCuRu/CNFs measured by ICP-OES and the corresponding atomic percentages.

FeCoNiMnRu	Fe	Co	Ni	Cu	Ru
Mass loading (wt. %)	2.85	2.95	2.57	2.54	4.13
Atomic percentage (at. %)	22.63	22.21	19.38	17.68	18.10

In addition, the electrocatalytic activities of Ru-containing electrocatalysts have been further evaluated by comparing a series of FeCoNiMnRu/CNFs with different Ru contents. The FeCoNiMnRu/CNFs with different Ru contents were denoted as FeCoNiMnRu_{0.5}/CNFs, FeCoNiMnRu/CNFs and FeCoNiMnRu₂/CNFs and the corresponding Ru contents were 2.41, 4.23 and 7.13 wt%, respectively, which were measured by ICP-OES. As shown in Supplementary Fig. 24, it is shown that the FeCoNiMnRu/CNFs and FeCoNiMnRu₂/CNFs display very similar overpotentials and Tafel slopes at 100 mA cm⁻², while FeCoNiMnRu_{0.5}/CNFs show obviously inferior activity. The ECSA values of FeCoNiMnRu/CNFs, FeCoNiMnRu_{0.5}/CNFs and FeCoNiMnRu₂/CNFs were calculated to be 2614, 2818, and 2093 cm² for FeCoNiMnRu/CNFs, FeCoNiMnRu_{0.5}/CNFs, and FeCoNiMnRu₂/CNFs, respectively. The R_{ct} values of FeCoNiMnRu/CNFs, FeCoNiMnRu_{0.5}/CNFs, and FeCoNiMnRu₂/CNFs were measured to be 11.4, 16.3, and 5.1 Ω, respectively. The OER activities were also evaluated by LSV curves. The overpotentials at 100 mA cm⁻² of FeCoNiMnRu/CNFs, FeCoNiMnRu_{0.5}/CNFs, and FeCoNiMnRu₂/CNFs were determined to be 308, 324, and 299 mV, respectively. The results indicate that FeCoNiMnRu/CNFs achieved higher current density than that of FeCoNiMnRu₂/CNFs at high overpotential (> 300 mV). Therefore, at relative low Ru content range (2-4 wt%), high Ru contents would lead to enhanced activity for water splitting. When the Ru contents increased to very high values (> 4 wt%), the Ru contents could pose negligible or negative effects on improving the electrocatalytic activity. Therefore, the results indicate that at a certain content of Ru, the differences in electrocatalytic activity of the three kinds of HEA could influenced by the electronegativity.

Supplementary Figure 24. (a) LSV curves, corresponding (b) Tafel plots, and (c) EIS spectra of the as-prepared FeCoNiMnRu/CNFs, FeCoNiMnRu_{0.5}/CNFs and FeCoNiMnRu₂/CNFs used for HER in 1 M KOH. (d) LSV curves of the as-prepared FeCoNiMnRu/CNFs, FeCoNiMnRu_{0.5}/CNFs and FeCoNiMnRu₂/CNFs used for OER in 1 M KOH.

Reviewer #2 (Remarks to the Author):

The authors report a study on the function of high-entropy alloys (HEAs) as catalysts for water dissociation and the correlation of the performance with the electronegativity of the included elements. The background and motivation for the study are well described, elemental composition analysis, XPS and Raman are well-performed and reported, but there are some serious shortcoming in the analysis and conclusions. The authors “propose a conceptual and experimental approach to overcome the limitations of single-element catalysts by designing a unique FeCoNiXRu (X: Cu, Cr, and Mn) HEA system with two kinds of active sites that have different adsorption capacities for multiple intermediates.” (line 68-71) and “and successfully identified electronegativity-dependent preferences for active site absorption of the intermediates” (line 74-76). This is not shown from the data.

Reply: We are thankful to the reviewer for the kind recommendation and careful review. We have provided more evidence to support the novelty of our manuscript. The application of HEA nano-materials in the field of electrocatalysis has been demonstrated to be viable in recently 3 years (Y. Yao, et al., *Science* 2018, 359, 1489-1494). The HEA nano-material is still in its initial stage as a kind of novel electrocatalyst, while the current researches mainly focused on the synthesis methods and component screening (Y. Yao, et al., *Sci. Adv.*, 2020, 6, eaaz0510; M. W. Glasscott, et al., *Nat. Commun.* 2019, 10, 2650). In regard to HEA nano-materials as electrocatalysts for water splitting (Z. Jia, et al., *Adv. Mater.* 2020, 32, 2000385; H. Qiu, et al., *J. Mater. Chem. A*, 2019, 7, 6499-6506. Z. Jin, et al., *Small*, 2019, 15, 1904180), only preliminary electrochemical tests have been conducted and the origin of the obtained excellent performance were simply ascribed to the “synergistic effect” derived from multicomponent. Therefore, electrocatalytic mechanism on HER are still extremely rare and discussions on the mechanism and the origin of the excellent activities are significantly lack. Meanwhile, the configurational complexity in HEA makes the identification of exact active sites difficult and complicated. How do the multi-element HEA work, how do the different metal sites in HEA adsorb species during electrolysis and what the roles do metal sites play? A lot of questions should be considered and therefore, in-depth investigations on the mechanism understanding and performance regulation are urgent need. In this work, we have elaborately investigated the electrocatalytic mechanism both experimentally and theoretically. We are the first example to unraveling the roles of each metal sites in HEA for electrocatalytic water splitting and correlate the electronegativity with activity. The electronegativity differences between mixed elements in HEA induce significant charge redistribution and create highly active Co and Ru sites with optimized energy barriers for simultaneously stabilizing OH* and H* intermediates, which greatly enhances the efficiency of water dissociation in alkaline conditions. Adjustment of the electrocatalytic activities of HEA catalysts were further shown experimentally and theoretically by tailoring the electronegativities of the compositions. This work provides an in-depth understanding of the interactions between specific active sites and intermediates, which opens up a new direction for breaking scaling relation issues for multistep reactions.

1. The authors' notion of electronegativity is very simplified and needs to be expanded. First, the Pauling electronegativity is defined from the ability to attract electrons when participating in a covalent bond. As the metals have metallic bonds, the notion of Pauling electronegativity is ill-defined. It is instead more common to use d-band center descriptions or other notions to quantify the effect of charge transfer in the system. The first and most simple case presented by Pauling 1932 with e.g. an electronegativity of 2.20 for Ru and Ni (1.91), Co (1.88), Fe (1.83), and Mn (1.55) is since then extended with e.g. Mulliken, Sanderson electronegativity, where for example Ru has an electronegativity of 1.54 in comparison to 1.80-1.88 for Fe, Co, Ni. One can note that the charge transfer with respect to Ru and surrounding elements would then be the opposite of what the authors describe if a more elaborated electronegativity scale is used.

Reply: We are thankful to the reviewer for the kind suggestion. Thanks for the comments. We were considering the electronegative values on Mulliken and Sanderson scales, which are based both on ionic and covalent bonds. For example, an electronegativity of 1.54 of Ru was calculated using Allen scale, which was also obtained based on one electron ionization energies, which is better to estimate the electronegativity on the formation of chemical bonds than the Pauling scale that is based on covalent bond (J. Am. Chem. Soc. 2000, 122, 5137). However, for metallic alloys, there are only metallic bonds, which is actually the balance between the Coulomb interaction force between the valence electron and the positively charged atomic substance and the repulsive force between the atomic cores. Therefore, only the distribution of electrons can be observed instead of transfer, forming the occupied and unoccupied Brillouin zone. Especially, for high entropy alloys with small number of atoms, the band theory cannot be used directly. The metallic bonds are more like covalent bonds than ionic bonds, and we probably use the method of molecular orbitals combination to solve the functions. Thus, it means Pauling electronegative values may be more suitable in high entropy alloys. And this is in good agreement with the Bader charge distribution results. As shown in Supplementary Fig. 42-44, there are more electrons can be observed on Ru after the charge distributions in all three structures.

Supplementary Figure 42. (a) Charge density difference analysis and (b) Bader charge analysis of FeCoNiMnRu HEA.

Supplementary Figure 43. (a) Charge density difference analysis and (b) Bader charge analysis of FeCoNiCrRu HEA.

Supplementary Figure 44. (a) Charge density difference analysis and (b) Bader charge analysis of FeCoNiCuRu HEA.

The XPS results also supported the electron interaction between Ru and other metals. The Fe 2p, Co 2p, Ni 2p, Mn 2p and Ru 2p XPS spectra of the controlled samples including Ru/CNFs, FeCoNi/CNFs, FeCoNiMn/CNFs, and FeCoNiRu/CNFs were performed to provide more electron interaction information among the metal elements in HEA (Supplementary Fig. 17). The corresponding binding energies (BEs) information were summarized in Supplementary Table 5. The BEs for Co, Ni, Fe and Mn in FeCoNiMnRu/CNFs show negative shifts when compared with FeCoNi/CNFs, FeCoNiRu/CNFs and FeCoNiMn/CNFs, respectively (Supplementary Fig. 17a-d). As shown in Supplementary Fig. 17e, the BEs for Ru in FeCoNiMnRu/CNFs exhibit positive shift when compared with the FeCoNiRu/CNFs and Ru/CNFs, suggesting that the Ru in HEA NPs served as the electron acceptor. The results strongly demonstrate the electron transfers from Fe, Co, Ni and Mn atoms to Ru atoms in HEA NPs that is due to the higher electronegativity of Ru (2.20) than those of Ni (1.91), Co (1.88), Fe (1.83), and Mn (1.55). Therefore, Pauling electronegative values may be more suitable in high entropy alloys.

Supplementary Figure 17. High-resolution (a) Mn 2p XPS spectrum of FeCoNiMn/CNFs and FeCoNiMnRu/CNFs, (b) Fe 2p, (c) Co 2p, (d) Ni 2p XPS spectra of FeCoNi/CNFs, FeCoNiMn/CNFs, FeCoNiRu/CNFs, and FeCoNiMnRu/CNFs, (e) Ru 3p XPS spectrum of Ru/CNFs, FeCoNiRu/CNFs, and FeCoNiMnRu/CNFs.

2. The electronegativity given by the authors, Ru (2.20) and Ni (1.91), Co (1.88), Fe (1.83), and Mn (1.55), are referred to be found in ref 28 (line 203-204), where they cannot be found, (or in ref 42 given in ref 28, or ref 6c, given in ref 42 in ref 28). From the numbers, however, it is apparent that they are from the Pauling electronegativity scale.

Reply: We are thankful to the reviewer for the kind suggestion. We have revised the references. Additionally, we noted that the charge transfer from transition metals (eg. Fe, Co, Ni) to Ru is a common phenomenon and widely observed in articles related to electrocatalysis (eg. Ritu Dhanda, et al. ChemistrySelect 2017, 2, 335-341. Feifei Zhang, et al. J. Mater. Chem. A, 2020, 8, 12810–12820. Xin Xiao, et al. Small Methods 2020, 1900796. Meixuan Li, et al. Adv. Sci. 2020, 7, 1901833. Caihua Zhang, et al. ACS Appl. Mater. Interfaces 2017, 9, 17326–17336). Xiang Xu, et al. (ACS Nano 2020, 14, 17704–17712) also extensively explored the electronic structures in HEA-NPs supported on carbon by XPS and first-principles electronic structure calculations, which found that the electron density redistribution in the NPs occurs from less electronegative elements to more electronegative ones. Interestingly, the authors performed a confirmatory experiment by synthesized an HEA-NP with “the more electronegative element Ru” to examine the behavior of the Cu (Pauling electronegativity: 1.90, less than that of Ru) core level, and “the direction of the core energy level shift of Cu in FeCoNiCuRu@CNF is in the positive direction”, in line with trend observed in our study. Therefore, Pauling electronegative values may be more suitable in high entropy alloys.

3. The electrochemical performance need to be clarified. In Fig 4a, Pt/C show very bad performance while it instead have outstanding performance in other publications. For a fair comparison, the same approach and optimization need to be done for all electrodes (same for Fig 4d). In addition, the large change is from inclusion of Ru, while the difference in-between the two Ru-containing electrocatalysts need to be verified to be from the claimed electronic effect, not from suppression of oxidation by inclusion of Ru or increased surface area due to intermixing of an additional element. From the electrochemical surface areas (ECSAs) normalized reported by the authors at line and Fig 4c, it seem to be a remained improvement, but from the values at line 253 with a Cdl value for FeCoNiMnRu/CNFs of 349.3 mF cm⁻², compared to the significantly lower value for e.g. FeCoNiMn/CNFs (274.5 mF cm⁻²) and the small shift in curves in the un-normalized curves Fig 4a, it does not make sense.

Reply: We are thankful to the reviewer for the kind suggestion. According to your advice, author have added the comparison of mass activity between the HEAs and commercial Pt/C catalysts in the revised manuscript. For a fair comparison, the FeCoNiMnRu/CNF, commercial Pt/C, and RuO₂ were all dropped on GCE to evaluate their activities. As shown in Supplementary Fig. 23, the FeCoNiMnRu/CNFs exhibit higher mass activity than the Pt/C catalysts for HER and RuO₂ catalysts for OER at high overpotential, which is benefit for the practical application. Actually, according to reported literatures, Pt is not excellent electrocatalyst for alkaline HER and OER.

Supplementary Fig. 23. The mass activities of (a) FeCoNiMnRu/CNFs and 20 wt% Pt/C for HER at overpotentials of 200 and 300 mV. (b) FeCoNiMnRu/CNFs and RuO₂ for OER at overpotentials of 400 and 500 mV.

According to your suggestion, we used the X-ray absorption spectroscopy (XAS) to investigate the structure of FeCoNiMnRu/CNFs. The X-ray absorption near-edge structure (XANES) results (Fig. 3a-d) indicate that the pre-absorption edge features for Co and the absorption edge for Ru both are metallicity by comparing with reference metal Co, CoO and Co₃O₄ foils, demonstrating that the Co and Ru elements in FeCoNiMnRu HEA NPs are in metallic states. The post-edge for Co and Ru in HEA exhibits slight deviation in the shape and intensity when compared with the reference metal Co and Ru foils. These features indicate the alloy formation rather than elemental segregation into pure metals, which would show the same length as metal Co and Ru foils. The extended X-ray absorption fine structure (EXAFS) of Co and Ru were determined through the fitting of the Fourier transform (FT) spectra. As shown in Fig. 3b and 3d, the FT-EXAFS spectra indicate that the average bond length of Ru and Ni in HEA NPs is quite different from the metallic bond in bulk Co and Ru references, suggesting that the Co and Ru elements are surrounded by different metallic species (Fe, Mn and Ni). The bond structures of Co and Ru in HEA reveal the similar average bond length without any oxidation when compared CoO and Co₃O₄ foils, further confirming the metallic states of Co and Ru in HEA after stability test. According to the EXAFS fitting (Supplementary Fig. 14), the bond length (R) and coordination numbers of each bond type in the HEA were summarized in Supplementary Table 4. The reliability of the fitting method is supported by smaller R factors.

Figure 3. (a) The Co K-edge XANES spectra and (b) FT-EXAFS spectra of FeCoNiMnRu/CNFs, Co foil,

CoO, and Co₃O₄. (c) The Ru K-edge XANES spectra and (d) FT-EXAFS spectra of FeCoNiMnRu/CNFs, Ru foil, and RuO₂.

Supplementary Figure 14. WT-EXAFS spectra of (a) Co and Ru foil, (b) Co and Ru in FeCoNiMnRu/CNFs.

According to your suggestion, we further used the XAS to investigate the chemical states of FeCoNiMnRu/CNFs before and after the HER stability test. As shown in Fig. 5a-f, the pre-absorption edge features for Co and the absorption edge for Ru both are metallicity by comparing with reference metal Co, CoO and Co₃O₄ foils (Fig. 3a-d), demonstrating that the Co and Ru elements in HEA NPs are in metallic state. Meanwhile, after the stability test, the XANES of Co and Ru in HEA NPs also keep metallic state, suggesting the excellent oxidation resistance during the long-term HER stability test. The FT-EXAFS spectra (Fig. 5b and 5d) indicate that the bond structure of Co and Ru in HEA before and after stability test reveal the similar average bond length without any oxidation when compared CoO and Co₃O₄ foils, further confirming the metallic states of Co and Ru in HEA after stability test. According to the EXAFS fitting (Fig. 5c and 5f), the bond length (R) and coordination numbers of each bond type in the HEA before and after stability test were summarized in Supplementary Table 4. The FT-EXAFS and WT-EXAFS spectra (Fig. 5g-i) of Co and Ru in HEA before and after stability tests have negligible mismatch, which means that the FeCoNiMnRu HEA keep metallic states in long-term stability tests, exhibiting extraordinary durability. The reliability of the fitting method is supported by smaller R factors.

Figure 5. (a) The Co K-edge XANES spectra and (b) FT-EXAFS spectra of FeCoNiMnRu/CNFs before and after stability test, and Co foil. (c) The corresponding FT-EXAFS fitting curves of

FeCoNiMnRu/CNFs before and after stability test. (d) The Ru K-edge XANES spectra and (e) FT-EXAFS spectra of FeCoNiMnRu/CNFs before and after stability test, and Ru foil. (f) The corresponding FT-EXAFS fitting curves of FeCoNiMnRu/CNFs before and after stability test. WT-EXAFS spectra of Co and Ru foil (g), Co and Ru in FeCoNiMnRu/CNFs before (h) and after stability test (i).

Supplementary Table 4. EXAFS fitting parameters at the Co and Ru K-edge for various samples.

Sample	Shell	CN ^a	R(Å) ^b	σ ² (Å ²) ^c	ΔE ₀ (eV) ^d	R factor
Co K-edge (S₀²=0.811)						
Co foil	Co-Co	12*	2.49	0.0063	6.9	0.0023
CoO	Co-O	6.0	2.10	0.0106	1.5	0.0053
	Co-Co	12.0	3.01	0.0080		
Co ₃ O ₄	Co-O	6.2	1.91	0.0035	-8.1	0.0079
	Co-Co	6.1	2.86	0.0048		
FeCoNiMn Ru/CNFs before stability test	Co-Co	7.4	3.36	0.0048	4.5	0.0059
	Co-Co	7.2	2.51	0.0066		
FeCoNiMn Ru/CNFs after stability test	Co-Ru	1.4	2.49	0.0066	3.0	0.0071
	Co-Co	6.9	2.50	0.0085		
Ru K-edge (S₀²=0.816)						
Ru foil	Ru-Ru	12*	2.67	0.0033	6.6	0.0071
	Ru-O	6.0	1.98	0.0035		
RuO ₂	Ru-Ru	3.0	3.12	0.0045	2.4	0.0085
	Ru-Ru	11.7	3.56	0.0045		
FeCoNiMn Ru/CNFs before stability test	Ru-Co	6.7	2.52	0.0049	-3.0	0.0059

Supplementary Figure 20. Double-layer capacitance per geometric area (C_{dl}) of the as-prepared electrodes.

To confirm that the performance improvement of FeCoNiMnRu/CNFs is not dominated by the increased surface area due to intermixing of an additional element, ECSA-normalized LSV curves were used to estimate the intrinsic activity of FeCoNiMnRu/CNFs and their medium- and low-entropy counterparts. The values of C_{dl} and ECSA of prepared materials have been remeasured. As shown in Supplementary Fig. 20, the C_{dl} values of FeCoNi/CNFs, FeCoNiMn/CNFs, FeCoNiRu/CNFs, and FeCoNiMnRu/CNFs are measured to be 16.4, 62.07, 110.53, 104.57 mF, with corresponding ECSA of 410, 1552, 2763, and 2614 cm^2 . ECSA-normalized LSV curves (Fig. 4c) indicate that the FeCoNiMnRu/CNFs show the highest intrinsic activity than those of FeCoNi/CNFs, FeCoNiMn/CNFs, and FeCoNiRu/CNFs.

Figure 4c. The ECSA-normalized LSV curves of FeCoNi/CNFs, FeCoNiMn/CNFs, FeCoNiRu/CNFs, and FeCoNiMnRu/CNFs in 1 M KOH.

Supplementary Figure 24. (a) LSV curves, corresponding (b) Tafel plots, and (c) EIS spectra of the as-prepared FeCoNiMnRu/CNFs, FeCoNiMnRu_{0.5}/CNFs and FeCoNiMnRu₂/CNFs used for HER in 1 M KOH. (d) LSV curves of the as-prepared FeCoNiMnRu/CNFs, FeCoNiMnRu_{0.5}/CNFs and FeCoNiMnRu₂/CNFs used for OER in 1 M KOH.

In addition, the electrocatalytic activities of Ru-containing electrocatalysts have been further evaluated by comparing a series of FeCoNiMnRu/CNFs with different Ru contents. The FeCoNiMnRu/CNFs with different Ru contents were denoted as FeCoNiMnRu_{0.5}/CNFs, FeCoNiMnRu/CNFs and FeCoNiMnRu₂/CNFs and the corresponding Ru contents were 2.41, 4.23 and 7.13 wt%, respectively, which were measured by ICP-OES. As shown in Supplementary Fig. 24, it is shown that the FeCoNiMnRu/CNFs and FeCoNiMnRu₂/CNFs display very similar overpotentials and Tafel slopes at 100 mA cm⁻², while FeCoNiMnRu_{0.5}/CNFs show obviously inferior activity. The ECSA values of FeCoNiMnRu/CNFs, FeCoNiMnRu_{0.5}/CNFs and FeCoNiMnRu₂/CNFs were calculated to be 2614, 2818, and 2093 cm² for FeCoNiMnRu/CNFs, FeCoNiMnRu_{0.5}/CNFs, and FeCoNiMnRu₂/CNFs, respectively. The R_{ct} values of FeCoNiMnRu/CNFs, FeCoNiMnRu_{0.5}/CNFs, and FeCoNiMnRu₂/CNFs were measured to be 11.4, 16.3, and 5.1 Ω, respectively. The OER activities were also evaluated by LSV curves. The overpotentials at 100 mA cm⁻² of FeCoNiMnRu/CNFs, FeCoNiMnRu_{0.5}/CNFs, and FeCoNiMnRu₂/CNFs were determined to be 308, 324, and 299 mV, respectively. The results indicate that FeCoNiMnRu/CNFs achieved higher current density than that of FeCoNiMnRu₂/CNFs at high overpotential (> 300 mV). Therefore, at relative low Ru content range (2-4 wt%), high Ru contents would lead to enhanced activity for water splitting. When the Ru contents increased to very high values (> 4 wt%), the Ru contents could pose negligible or negative effects on improving the electrocatalytic activity. Therefore, the results indicate that at a certain content of Ru, the differences in electrocatalytic activity of the three kinds of HEA could be influenced by the electronegativity.

4. The calculations does not seem to include variation in mixing in the high entropy alloys. The effect of different elements at different sites around the active site will naturally change the adsorption energy. This need to be done and is a quite more demanding efforts than the calculations reported in the manuscript.

Reply: We are thankful to the reviewer for the kind suggestion. Yes, the surrounding of each element position will change the effect of adsorption. Considering that the difference in atomic configurations may change the adsorption energy, the hydrogen adsorbing energy (ΔG_{H^*}) on Fe-, Co-, Ni-, Ru-sites in FeCoNiXRu (X = Mn, Cr, Cu) with 8 randomly selected configurations were analyzed by DFT calculation. The corresponding atomic configurations and ΔG_{H^*} were shown in Supplementary Fig. 38-40, and Fig. 55 and the values of ΔG_{H^*} were summarized in Supplementary Table 9-11. DFT Results suggest that the all of the Ru sites in FeCoNiMnRu, FeCoNiCuRu and FeCoNiCrRu display the lowest values than Fe, Co, and Ni sites, demonstrating that the atomic configurations could not affect the ΔG_{H^*} orders on Fe, Co, Ni and Ru sites in different HEA NPs (Supplementary Fig. 55). In addition, the Ru sites in FeCoNiMnRu also exhibit the lowest ΔG_{H^*} than those Ru sites in FeCoNiCrRu and FeCoNiCuMn HEA, further indicating that the fifth metal (Mn, Cr and Cu) with low electronegativity in HEA could reduce the ΔG_{H^*} . According to the bond length and coordination numbers of Co and Ru in HEA, we have chosen the atomic configuration accordingly with XAS results as represent sample to show the relationship between the ΔG_{H^*} and electronegativity.

Supplementary Figure 38. Different atomic configurations of FeCoNiMnRu HEA.

Supplementary Figure 39. Different atomic configurations of FeCoNiCrRu HEA.

Supplementary Figure 40. Different configurations of FeCoNiCuRu HEA.

Supplementary Figure 55. The summarized values of ΔG_{H^+} on Fe-, Co-, Ni-, and Ru-sites in (a) FeCoNiMnRu, (b) FeCoNiCrRu, and (c) FeCoNiCuRu with different atomic configurations.

Supplementary Table 4. EXAFS fitting parameters at the Co and Ru K-edge for various samples.

Sample	Shell	CN ^a	R(\AA) ^b	$\sigma^2(\text{\AA}^2)$ ^c	$\Delta E_0(\text{eV})$ ^d	R factor
Co K-edge (S₀²=0.811)						
Co foil	Co-Co	12*	2.49	0.0063	6.9	0.0023
CoO	Co-O	6.0	2.10	0.0106	1.5	0.0053
	Co-Co	12.0	3.01	0.0080		
Co ₃ O ₄	Co-O	6.2	1.91	0.0035	-8.1	0.0079
	Co-Co	7.4	3.36	0.0048		
FeCoNiMn Ru/CNFs before stability test	Co-Co	7.2	2.51	0.0066	4.5	0.0059
	Co-Ru	1.4	2.49	0.0066		
FeCoNiMn	Co-Co	6.9	2.50	0.0085	3.0	0.0071

Ru/CNFs after stability test	Co-Ru	0.3	2.52	0.0085		
			Ru K-edge ($S_{\sigma}^2=0.816$)			
Ru foil	Ru-Ru	12*	2.67	0.0033	6.6	0.0071
	Ru-O	6.0	1.98	0.0035		
RuO ₂	Ru-Ru	3.0	3.12	0.0045	2.4	0.0085
	Ru-Ru	11.7	3.56	0.0045		
FeCoNiMn Ru/CNFs before stability test	Ru-Co	6.7	2.52	0.0049	-3.0	0.0059

5. The simple notion of Pauling electronegativity is not advised to use as a descriptor and put on the same scale of accuracy as DFT calculations, here at least Mulliken or Allen electronegativity scale should be used. As the authors have performed DFT calculations, analysis of the charge density is preferred over the notion of electronegativity where e.g. Bader analysis and will give the charge distribution and possible charge transfer from first-principles directly. That is similar to the analysis reported for just one composition in SI Fig 22, but can instead be used to show the charge transfer in-between the chosen compositions in their materials (using fully randomized structures).

Reply: We are thankful to the reviewer for the kind suggestion. Moreover, the d-band center of each metal sites in FeCoNiMnRu and the three HEA structures were re-calculated and shown in Fig. 7c and 7d. The d-band orbitals with large spin polarization can be divided into spin up and spin down. As shown in Fig.7d, a smaller number of spin states occupy the spin down orbitals, which is likely to be a spin-polarized catalytic active center and generally participate in more catalytic reactions (Nat Commun 9, 1610 (2018)). As shown in Supplementary Fig. 42-44, there are more electrons can be observed on Ru after the charge distributions in all three HEA structures. The results indicate the charge transfers from Fe, Co, Ni and Mn metals to Ru metal.

Figure 7. (c) The d-orbital projected density of states (PDOS) of Fe, Co, Ni, Mn, Ru, and FeCoNiMnRu. (d) Comparison of PDOS of FeCoNiXRu (X = Mn, Cr, Cu).

Supplementary Figure 42. (a) Charge density difference analysis and (b) Bader charge analysis of FeCoNiMnRu HEA.

Supplementary Figure 43. (a) Charge density difference analysis and (b) Bader charge analysis of FeCoNiCrRu HEA.

Supplementary Figure 44. (a) Charge density difference analysis and (b) Bader charge analysis of FeCoNiCuRu HEA.

6. The PBE functional is not suitable for calculation of correct adsorption energies for the hydrogen evolution reaction, here vdW corrections are needed, and the BEEF + vdW is known to be the approach that correlate with experiments (Wallendorff et al Surface Science 640 (2015): 36-44. Sharada et al. Physical Review B 100.3 (2019): 035439.).

Reply: We are thankful to the reviewer for the kind suggestion. Thanks for the comments. vdW corrections were added. Similar results were obtained.

I suggest that the authors update their manuscript with the above considerations and submit to a more technical journal. I cannot recommend publication at this stage.

Reply: We are thankful to the reviewer for the kind suggestion. Those comments are all valuable and very helpful for revising and improving our manuscript, as well as the important guiding significance to our research. We have done a thorough revision of our manuscript according to reviewers' valuable comments. Please reconsider your suggestion. Thank you.

Reviewer #3 (Remarks to the Author):

In this work, the authors applied a FeCoNiXRu (X: Cu, Cr, and Mn) high-entropy alloy (HEA) system as an alkaline water electrolysis catalyst. The developed FeCoNiCu HEA electrocatalyst showed improved intrinsic electrochemical performance through transformation from a multiphase to a single-phase HEA with charge redistribution of multi-components and optimized energy barrier of Co and Ru sites, which were attributed to electronegativity differences between mixed elements in HEA (HER overpotential: 82 mV at 100 mA cm⁻², OER overpotential: 308 mV at 100 mA cm⁻², full cell overpotential: 1.65 V at 100 mA cm⁻², half cell cycle stability: 10,000 cycles, full cell chronoamperometric stability: 600 h at 1 A cm⁻²). The demonstrated HER/OER and full cell performance of the proposed material system is noticeable, however, the applied multi-metal in this work has already been introduced in various previous HEA electrocatalyst system. Moreover, the key mechanism for the electrochemical performance enhancement of the developed catalyst does not seem to be distinguishable from those of previous studies (e.g., transformation to single-phase, optimized electronic structure via electron transfer, reduced energy barrier in rate-determining step, etc.). Therefore, the reviewer concludes that this work is not suitable for the wide readership of Nature Communications. Below are some of the issues that should be considered before submitting this work elsewhere.

Reply: We are thankful to the reviewer for the review. The application of HEA nano-materials in the field of electrocatalysis has been demonstrated in recently years (Y. Yao, et al. *Science* 2018, 359, 1489-1494). The HEA nano-material is still in its initial stage as a kind of novel electrocatalyst, while the current concern mainly focused on the synthesis methods and component screening (Y. Yao, et al. *Sci. Adv.*, 2020, 6; M. W. Glasscott, et al. *Nat Commun* 2019, 10, 2650), and only preliminary electrochemical tests have been conducted. Further researches on the electrocatalytic mechanism are still extremely rare. In regard to HEA nano-materials as electrocatalysts for hydrogen evolution reaction (HER), configurational complexity renders the identification of exact active sites difficult and complicated, and thus discussions on the mechanism or the origin of the excellent activities are usually deficient or simply ascribe to the “synergistic effect” derived from multicomponent (Zhe Jia, et al. *Adv. Mater.* 2020, 32, 2000385; H. Qiu, et al. *J. Mater. Chem. A* 2019, 7, 6499–6506; Small 2019, 15, 1904180; Miaomiao Liu, et al. *Adv. Mater. Interfaces* 2019, 1900015). Therefore, in addition to synthesis methods, electrocatalytic mechanism on HER are still extremely rare and discussions on the mechanism and the origin of the excellent activities are usually deficient. Meanwhile, the configurational complexity makes the identification of exact active sites difficult and complicated. How the multi-element HEA works, how do the different metal sites in HEA adsorb species during electrolysis and what the roles do metal sites play? A lot of questions should be considered and therefore, in-depth investigations on the mechanism understanding and performance regulation are urgently deserved.

In this work, we have elaborately investigated the electrocatalytic mechanism both experimentally and theoretically. We are the first example to unraveling the role of each metal sites in HEA for electrocatalytic water splitting and correlate the electronegativity with activity. The transformation from multiple phases to a single phase was evident from real-time *in situ* XRD results and demonstrated a thermodynamically driven phase transition. The thermodynamically driven phase transition of FeCoNiMnRu NPs in CNF nanofiber reactors was firstly evidenced by the *in situ* XRD characterizations. The electronegativity differences between mixed elements in HEA induce significant charge redistribution and this “self-optimizing” process create highly active Co and Ru

sites. We have figured out the roles and functions of each metal sites in HEA for electrocatalytic water splitting. We have elaborately investigated the electrocatalytic mechanism both experimentally and theoretically, *i.e.*, the *in situ* Raman provided direct evidence for the reaction intermediate selectively adsorbed on specific sites and DFT calculation also discussed the influence of atomic configuration. Furthermore, the effect of electronegativity of X on the intrinsic activity of HEA is also explored for providing new paths for the design of more efficient HEA catalyst. To the best of our knowledge, no research has showed the in-depth understanding of the role of HEA metal sites on water splitting.

1. In this work, the origin of enhanced hydrogen/oxygen generation performance in FeCoNiXRu (X: Cu, Cr, and Mn) HEA is attributed to optimized electronic structures of the alloy via combination of high-entropy alloying with multiple elements. However, numerous previous studies have already reported that HEA electrocatalysts comprising similar multi-components (including Fe, Co, Ni, Cu, Cr, and Mn) with synergetic effects can enhance the HER and OER kinetics (*Electrochim. Acta*, 2021, 378, 138142; *Chem. Eng. J.*, 2021, 425, 131533; *J. Mater. Chem. A*, 2021, 9, 889-893; *Electrochim. Acta*, 2018, 279, 19-23; *Adv. Mater. Interfaces*, 2019, 6(7), 1900015; *Nat. Commun.*, 2020, 11, 2016: <https://doi.org/10.1038/s41467-020-15934-1>; *Chem. Sci.*, 2020, 11, 12731). The authors should clearly demonstrate and discuss the advancement of this work from previous studies.

Reply: We are thankful to the reviewer for the kind suggestion. These articles (*Adv. Mater. Interfaces*, 2019, 6, 1900015; *Nat. Commun.*, 2020, 11, 2016) mainly focus on the synthesis method of HEA through ultrasonication-assisted wet chemistry method and fast-moving bed pyrolysis process, and however, the exploration of HEA for electrocatalysis is lack of. As mentioned above, the HEA nano-material is still in its initial stage as a kind of novel electrocatalyst, the in-depth understanding of electrocatalytic mechanism and identifying of real active sites are of great interest. Therefore, on the basis of extensively exploring the thermodynamically driven phase transition of FeCoNiMnRu NPs in CNF nanofiber reactors by the *in situ* XRD characterizations, we also perform systematically electrochemical characterizations and further unveil the active sites by *in situ* Raman and elaborate DFT calculation (8 randomly selected configurations) to reveal the operational principle of HEA as electrocatalysts.

Another three articles (*Electrochim. Acta*, 2021, 378, 138142; *Chem. Eng. J.*, 2021, 425, 131533; *Electrochim. Acta*, 2018, 279, 19-23) mainly focus on the application of HEA in addition to the material preparation, which provide exhaustive performance test and achieve technical advances. However, the discussion of mechanism understanding and identification of exact active sites are lacking, which is urgently demanded for the development of HEA-based electrocatalysts as an important scientific issue. In this work, we have elaborately investigated the electrocatalytic mechanism both experimentally and theoretically. To the best of our knowledge, we are the first example to unraveling the role of each metal sites in HEA for electrocatalytic water splitting and correlate the electronegativity with activity.

Similarly, Dongshuang Wu et al. analyzed the electronic structure of IrPdPtRhRu HEA NPs by HAXPES and indicated that “the HEA NPs possessed a broad valence band spectrum without any obvious peaks”, which “implies that the HEA NPs have random atomic configurations leading to a variety of local electronic structures” (*Chem. Sci.*, 2020, 11, 12731). This article revealed the characteristics of electronic structure of the HEA-based electrocatalysts. As indicated by the authors, however, the HEA-based electrocatalysts possess random atomic configurations, which

hinder the further investigation of electrocatalytic mechanism and identification of exact active sites. In this regard, we have revealed the different roles of multi-sites in HEA system by *in situ* Raman and elaborate DFT calculation (8 randomly selected configurations), providing a more comprehensive and precise understanding of the electrocatalytic mechanism of HEA-based electrocatalysts.

Dan Zhang et al. also analyzed the role of each element in PdFeCoNiCu HEA NPs by DFT calculations, indicating that “Fe enhances the d – d electron coupling and compensates the electrons at the Co position; electron-rich Cu promotes overall electron transfer between metal sites; the pinning of electroactive Co sites favors water splitting, while nearby Ni facilitates *OH stabilization from water splitting; highly active Pd sites for H₂ formation determine the efficient HER process in alkaline media” (*J. Mater. Chem. A*, 2021, 9, 889-893). These results are nice and exciting, but direct experimental evidence is lacking, especially the more convincing *in situ* characterization. Moreover, the DFT calculation did not consider the influence of complicated atomic configurations of HEA on the adsorption of the reactants and key intermediate, which is a nonnegligible issue for fundamental understanding the electrocatalytic mechanism for HEA-based electrocatalysts. To provide direct evidence for identifying real active sites in HEA systems, *in situ* Raman has been conducted in our study. Furthermore, elaborate DFT calculations of FeCoNiXRu (X = Mn, Cr, Cu) with 8 randomly selected configurations have been performed to theoretically unveil the different role of different sites in HEA system and effect of electronegativity on activity, providing a reference for further understanding of HEA-based electrocatalysts.

In this work, we have elaborately investigated the electrocatalytic mechanism both experimentally and theoretically. We are the first example to unraveling the role of each metal sites in HEA for electrocatalytic water splitting and correlate the electronegativity with activity. The transformation from multiple phases to a single phase was evident from real-time *in situ* XRD results and demonstrated a thermodynamically driven phase transition. The thermodynamically driven phase transition of FeCoNiMnRu NPs in CNF nanofiber reactors was firstly evidenced by the *in situ* XRD characterizations. The electronegativity differences between mixed elements in HEA induce significant charge redistribution and this “self-optimizing” process create highly active Co and Ru sites. We have figured out the roles and functions of each metal sites in HEA for electrocatalytic water splitting. We have elaborately investigated the electrocatalytic mechanism both experimentally and theoretically, *i.e.*, the *in situ* Raman provided direct evidence for the reaction intermediate selectively adsorbed on specific sites and DFT calculation also discussed the influence of atomic configuration. Furthermore, the effect of electronegativity of X on the intrinsic activity of HEA is also explored for providing new paths for the design of more efficient HEA catalyst. To the best of our knowledge, no research has showed the in-depth understanding of the role of HEA metal sites on water splitting.

2. The ICP-OES analysis showed that the molar ratio of Fe:Co:Ni:Mn:Ru was 1:1.01:1.01:0.75:0.75. However, to validate whether the optimal HER/OER performance is obtained from the above molar ratio (1:1.01:1.01:0.75:0.75), the HER/OER performance (overpotential, Tafel slope, EIS, ECSA, etc.) of other control samples with various molar ratios should be provided.

Reply: We are thankful to the reviewer for the kind suggestion. The electrocatalytic activities of Ru-containing electrocatalysts have been further evaluated by comparing a series of FeCoNiMnRu/CNFs with different Ru contents. The FeCoNiMnRu/CNFs with different Ru contents

were denoted as FeCoNiMnRu_{0.5}/CNFs, FeCoNiMnRu/CNFs and FeCoNiMnRu₂/CNFs and the corresponding Ru contents were 2.41, 4.23 and 7.13 wt%, respectively, which were measured by ICP-OES. As shown in Supplementary Fig. 24, it is shown that the FeCoNiMnRu/CNFs and FeCoNiMnRu₂/CNFs display very similar overpotentials and Tafel slopes at 100 mA cm⁻², while FeCoNiMnRu_{0.5}/CNFs show obviously inferior activity. The ECSA values of FeCoNiMnRu/CNFs, FeCoNiMnRu_{0.5}/CNFs and FeCoNiMnRu₂/CNFs were calculated to be 2614, 2818, and 2093 cm² for FeCoNiMnRu/CNFs, FeCoNiMnRu_{0.5}/CNFs, and FeCoNiMnRu₂/CNFs, respectively. The R_{ct} values of FeCoNiMnRu/CNFs, FeCoNiMnRu_{0.5}/CNFs, and FeCoNiMnRu₂/CNFs were measured to be 11.4, 16.3, and 5.1 Ω, respectively. The OER activities were also evaluated by LSV curves. The overpotentials at 100 mA cm⁻² of FeCoNiMnRu/CNFs, FeCoNiMnRu_{0.5}/CNFs, and FeCoNiMnRu₂/CNFs were determined to be 308, 324, and 299 mV, respectively. The results indicate that FeCoNiMnRu/CNFs achieved higher current density than that of FeCoNiMnRu₂/CNFs at high overpotential (> 300 mV). Therefore, at relative low Ru content range (2-4 wt%), high Ru contents would lead to enhanced activity for water splitting. When the Ru contents increased to very high values (> 4 wt%), the Ru contents could pose negligible or negative effects on improving the electrocatalytic activity. Therefore, high-entropy alloys formed by approximately equal amounts of metals possess the best electrocatalytic activity and the results indicate that at a certain content of Ru, the differences in electrocatalytic activity of the three kinds of HEA could be influenced by the electronegativity.

Supplementary Figure 24. (a) LSV curves, corresponding (b) Tafel plots, and (c) EIS spectra of the as-prepared FeCoNiMnRu/CNFs, FeCoNiMnRu_{0.5}/CNFs and FeCoNiMnRu₂/CNFs used for HER in 1 M KOH. (d) LSV curves of the as-prepared FeCoNiMnRu/CNFs, FeCoNiMnRu_{0.5}/CNFs and FeCoNiMnRu₂/CNFs used for OER in 1 M KOH.

3. Specific surface area of the water electrolysis catalyst is directly related to the water electrolysis performance because it is correlated to the active site, which, however, is missing in this study. The authors are recommended to quantitatively compare the specific surface area of the materials used in the study.

Reply: We are thankful to the reviewer for the kind suggestion. The electrochemical active area is a concept relative to the geometric area. For traditional electrodes, the active site is considered to

be the part of the outer surface of the catalyst electrode that is in contact with the electrolyte. Therefore, it can be considered that the geometric area is approximately equal to the electrochemically active area. But for some nano-material electrodes, especially some porous electrodes, such as foamed nickel electrodes, gas diffusion layer electrodes, etc., after the nano-catalyst is loaded, the specific surface area is very large, and abundant active sites are exposed, which will inevitably lead to increased electrochemical performance. At this time, using the geometric area to calculate the current density obviously cannot really reflect the activity of the catalyst, nor can it compare the performance of different catalysts well, so that a scientific conclusion can be obtained. The specific surface area measured by BET is based on the adsorption and desorption of N_2 , while the areas that adsorb N_2 may not be active for electrochemical process. So to put it plainly, an ideal electrochemical active area (ECSA) is the area where the active sites of the catalyst are in contact with the electrolyte. Therefore, we used the electrochemical approach to simulate and calculate the electrochemically active area of the samples to show the intrinsic activity. ECSA-normalized LSV curves were used to estimate the intrinsic activity of FeCoNiMnRu/CNFs and their medium- and low-entropy counterparts. The values of C_{dl} and ECSA of prepared materials have been remeasured. As shown in Supplementary Fig. 20, the C_{dl} values of FeCoNi/CNFs, FeCoNiMn/CNFs, FeCoNiRu/CNFs, and FeCoNiMnRu/CNFs are measured to be 16.4, 62.07, 110.53, 104.57 mF, which corresponding to ECSA of 410, 1552, 2763, and 2614 cm^2 . ECSA-normalized LSV curves (Fig. 4c) indicate that the FeCoNiMnRu/CNFs show the highest intrinsic activity than those of FeCoNi/CNFs, FeCoNiMn/CNFs, and FeCoNiRu/CNFs.

Supplementary Figure 20. Double-layer capacitance per geometric area (C_{dl}) of the as-prepared electrodes.

Figure 4c. The ECSA-normalized LSV curves of FeCoNi/CNFs, FeCoNiMn/CNFs, FeCoNiRu/CNFs, and FeCoNiMnRu/CNFs in 1 M KOH.

4. For the real-time in situ XRD analysis in Figure 2f-h, a positive shift of (111) plane of FeCoNiMnRu HEA with annealing treatment at 1000 °C for 3 h compared to that prepared at 800-1000 °C was observed, which was attributed to the reduced lattice distortion in the HEA. The authors claimed that this result corroborated the single-phase formation of FeCoNiMnRu HEA. In addition to the (111) plane, additional diffraction peaks of (200) and (220) planes can also be found in FeCoNiMnRu. To better elucidate the single-phase formation of FeCoNiMnRu HEA with prolonged annealing treatment at 1000 °C, the correlation between the peak shift of (200)/(220) planes and thermodynamically driven phase transition should also be discussed.

Reply: We are thankful to the reviewer for the kind suggestion. The conclusion that single-phase HEA was supported by the XRD pattern, and high resolution XRD patterns with detailed Rietveld refinements. As shown in Fig. 2f, the FeCoNiMnRu/CNFs exhibits three main diffraction peaks at $2\theta=43^\circ$, 50° and 74° , which can be indexed to the (111), (200), and (220) planes of the *fcc* phases (PDF#47-1417), respectively. No separated XRD peaks from pure/unary metals or metal oxides were observed, suggesting the formation of a single-phase HEA. The crystal structure of FeCoNiMnRu NPs obtained at 800 °C-3h (Supplementary Fig. 13) and 1000 °C-3h (Fig. 2f) were further analyzed by detailed Rietveld refinements. The results confirmed that both of the FeCoNiMnRu HEA NPs prepared at 800 and 1000 °C are single phase structure with similar component and the simulative components of FeCoNiMnRu NPs models matches well with the results of ICP.

Figure 2f. XRD pattern of FeCoNiMnRu/CNFs (1000 °C-3 h) with a scan rate of 0.5° /min and the corresponding XRD Rietveld refinement.

Supplementary Fig. 13. XRD pattern of FeCoNiMnRu/CNFs (800 °C-3 h) with a scan rate of 0.5°/min and the corresponding XRD Rietveld refinement.

Supplementary Fig. 11. The enlarged *in situ* XRD patterns of FeCoNiMnRu/CNFs showing the peak shift of (200) and (220) planes.

The temperature-dependent *in situ* XRD patterns display two kinds of shifted peak positions. The negatively shifted peak position correlated with the increased treatment temperatures from 400 to 1000 °C were attributed to the thermodynamically driven phase transition from multiple-phase to single-phase of HEA. In addition, the prolonged annealing (0-3h) treatment at 1000 °C induced positively peak shift, which was attributed to the reduced lattice distortion of the HEA crystal. The peak of (111) plane was selected to analyze the phase transformation process due to its largest peak intensity. In regard of (200) and (220) planes (Supplementary Fig. 13), the peak shifts display the same trend as that of (111) planes. The transformation from multiple phases to a single phase was evident by the real-time *in situ* XRD results.

5. The electronic interaction in the proposed HEA is attributed to the difference in electronegativity between the constituent elements, which was supported by the XPS analysis. However, current XPS analysis on the water electrolysis performance can be further elaborated by providing more

in-depth analysis via comparing FeCoNiMnRu HEA with different control samples of FeCoNi, FeCoNiMn, FeCoNiRu. For example, in Figure 3c, the $\text{Co}^{3+}/\text{Co}^{2+}$ ratio can suggest more detailed information for the OER kinetics of the electrocatalyst (Electrochimica Acta, 2017, 254: 14-24). Also, in Figure 3f, O 1s peak is deconvoluted into hydroxyl group and residual oxygen containing group only. Here, the O 1s peak can also be deconvoluted to lattice oxygen, highly oxidative oxygen, surface-active oxygen, and adsorbed water, and the OER performance can be discussed in terms of the surface-active oxygen/lattice oxygen ratio (Electrochimica Acta, 2017, 254: 14-24).

Reply: We are thankful to the reviewer for the kind suggestion. The electron transfers among the five metal elements induced by their different electronegativities were investigated by the XPS and Bader charge distribution calculated by DFT. The Fe 2p, Co 2p, Ni 2p, Mn 2p and Ru 2p XPS spectra of the controlled samples including Ru/CNFs, FeCoNi/CNFs, FeCoNiMn/CNFs, and FeCoNiRu/CNFs were performed to provide more electron interaction information among the metal elements in HEA (Supplementary Fig. 17). The corresponding binding energies (BEs) information were summarized in Supplementary Table 5. The BEs for Co, Ni, Fe and Mn in FeCoNiMnRu/CNFs show negative shifts when compared with FeCoNi/CNFs, FeCoNiRu/CNFs and FeCoNiMn/CNFs, respectively (Supplementary Fig. 17a-d). As shown in Supplementary Fig. 17e, the BEs for Ru in FeCoNiMnRu/CNFs exhibit positive shift when compared with the FeCoNiRu/CNFs and Ru/CNFs, suggesting that the Ru in HEA NPs served as the electron acceptor. The results strongly demonstrate the electron transfers from Fe, Co, Ni and Mn atoms to Ru atoms in HEA NPs that is due to the higher electronegativity of Ru (2.20) than those of Ni (1.91), Co (1.88), Fe (1.83), and Mn (1.55).

Supplementary Figure 17. High-resolution (a) Mn 2p XPS spectrum of FeCoNiMn/CNFs and FeCoNiMnRu/CNFs, (b) Fe 2p, (c) Co 2p, (d) Ni 2p XPS spectra of FeCoNi/CNFs, FeCoNiMn/CNFs, FeCoNiRu/CNFs, and FeCoNiMnRu/CNFs, (e) Ru 3p XPS spectrum of Ru/CNFs, FeCoNiRu/CNFs, and FeCoNiMnRu/CNFs.

As shown in Supplementary Fig. 42-44, there are more electrons can be observed on Ru after the charge distributions in all three HEA structures. The results indicate the charge transfers from Fe, Co, Ni and Mu metals to Ru metal.

Supplementary Figure 42. (a) Charge density difference analysis and (b) Bader charge analysis of FeCoNiMnRu HEA.

Supplementary Figure 43. (a) Charge density difference analysis and (b) Bader charge analysis of FeCoNiCrRu HEA.

Supplementary Figure 44. (a) Charge density difference analysis and (b) Bader charge analysis of FeCoNiCuRu HEA.

The XPS spectra of FeCoNiMnRu/CNFs in Fig. 3 have confirmed the existence of metallic states of Fe, Co, Ni Mn and Ru in HEA NPs. According to the O 1s XPS spectra in Fig. 3f, there were only BEs for hydroxyl group and residual oxygen containing group, indicating that some of the high valences of these elements were attributed to the absorbed oxygen species or the surface oxidation during the XPS preparing. In order to show the metallic states of HEA NPs, we used the X-ray absorption spectroscopy (XAS) to investigate the structure of FeCoNiMnRu/CNFs. The X-ray absorption near-edge structure (XANES) results (Fig. 3a-d) indicate that the pre-absorption edge features for Co and the absorption edge for Ru both are metallicity by comparing with reference metal Co, CoO and Co₃O₄ foils, demonstrating that the Co and Ru elements in FeCoNiMnRu HEA NPs are in metallic states. The post-edge for Co and Ru in HEA exhibits slight deviation in the shape and intensity when compared with the reference metal Co and Ru foils. These features indicate the alloy formation rather than elemental segregation into pure metals, which would show the same length as metal Co and Ru foils. The extended X-ray absorption fine structure (EXAFS) of Co and Ru were determined through the fitting of the Fourier transform (FT) spectra. As shown in Fig. 3b and 3d, the FT-EXAFS spectra indicate that the average bond length of Ru and Ni in HEA NPs is quite

different from the metallic bond in bulk Co and Ru references, suggesting that the Co and Ru elements are surrounded by different metallic species (Fe, Mn and Ni). The bond structures of Co and Ru in HEA reveal the similar average bond length without any oxidation when compared CoO and Co₃O₄ foils, further confirming the metallic states of Co and Ru in HEA after stability test. According to the EXAFS fitting (Supplementary Fig. 14), the bond length (R) and coordination numbers of each bond type in the HEA were summarized in Supplementary Table 4. The reliability of the fitting method is supported by smaller R factors.

Figure 3. (a) The Co K-edge XANES spectra and (b) FT-EXAFS spectra of FeCoNiMnRu/CNFs, Co foil, CoO, and Co₃O₄. (c) The Ru K-edge XANES spectra and (d) FT-EXAFS spectra of FeCoNiMnRu/CNFs, Ru foil, and RuO₂.

Supplementary Figure 14. WT-EXAFS spectra of (a) Co and Ru foil, (b) Co and Ru in FeCoNiMnRu/CNFs.

According to your suggestion, we further used the XAS to investigate the chemical states of FeCoNiMnRu/CNFs before and after the HER stability test. As shown in Fig. 5a-f, the pre-absorption edge features for Co and the absorption edge for Ru both are metallicity by comparing with reference metal Co, CoO and Co₃O₄ foils (Fig. 3a-d), demonstrating that the Co and Ru elements in HEA NPs are in metallic state. Meanwhile, after the stability test, the XANES of Co and Ru in HEA NPs also keep metallic state, suggesting the excellent oxidation resistance during the long-term HER stability test. The FT-EXAFS spectra (Fig. 5b and 5d) indicate that the bond structure of Co and Ru in HEA before and after stability test reveal the similar average bond length without any oxidation when compared CoO and Co₃O₄ foils, further confirming the metallic states of Co and Ru

in HEA after stability test. According to the EXAFS fitting (Fig. 5c and 5f), the bond length (R) and coordination numbers of each bond type in the HEA before and after stability test were summarized in Supplementary Table 4. The FT-EXAFS and WT-EXAFS spectra (Fig. 5g-i) of Co and Ru in HEA before and after stability tests have negligible mismatch, which means that the FeCoNiMnRu HEA keep metallic states in long-term stability tests, exhibiting extraordinary durability. The reliability of the fitting method is supported by smaller R factors.

Figure 5. (a) The Co K-edge XANES spectra and (b) FT-EXAFS spectra of FeCoNiMnRu/CNFs before and after stability test, and Co foil. (c) The corresponding FT-EXAFS fitting curves of FeCoNiMnRu/CNFs before and after stability test. (d) The Ru K-edge XANES spectra and (e) FT-EXAFS spectra of FeCoNiMnRu/CNFs before and after stability test, and Ru foil. (f) The corresponding FT-EXAFS fitting curves of FeCoNiMnRu/CNFs before and after stability test. WT-EXAFS spectra of Co and Ru foil (g), Co and Ru in FeCoNiMnRu/CNFs before (h) and after stability test (i).

Supplementary Table 4. EXAFS fitting parameters at the Co and Ru K-edge for various samples.

Sample	Shell	CN^a	$R(\text{Å})^b$	$\sigma^2(\text{Å}^2)^c$	$\Delta E_0(\text{eV})^d$	R factor
Co K-edge ($S\sigma^2=0.811$)						
Co foil	Co-Co	12*	2.49	0.0063	6.9	0.0023
CoO	Co-O	6.0	2.10	0.0106	1.5	0.0053
	Co-Co	12.0	3.01	0.0080		
Co ₃ O ₄	Co-Co	6.1	2.86	0.0048	-8.1	0.0079
	Co-O	6.2	1.91	0.0035		
FeCoNiMn Ru/CNFs before stability test	Co-Co	7.2	2.51	0.0066	4.5	0.0059
	Co-Ru	1.4	2.49	0.0066		
FeCoNiMn	Co-Co	6.9	2.50	0.0085	3.0	0.0071

Ru/CNFs after stability test	Co-Ru	0.3	2.52	0.0085		
Ru K-edge ($S_0^2=0.816$)						
Ru foil	Ru-Ru	12*	2.67	0.0033	6.6	0.0071
	Ru-O	6.0	1.98	0.0035		
RuO ₂	Ru-Ru	3.0	3.12	0.0045	2.4	0.0085
	Ru-Ru	11.7	3.56	0.0045		
FeCoNiMn Ru/CNFs before stability test	Ru-Co	6.7	2.52	0.0049	-3.0	0.0059

Through our systematic investigation including *in situ* Raman, DFT, XAS, and electrochemical evaluation, the electronegativity differences between mixed elements in HEA induce significant charge redistribution and create highly active metal sites with optimized energy barriers for simultaneously stabilizing OH* and H* intermediates, which greatly enhances the efficiency of water dissociation in alkaline conditions.

6. In Figure 3, the authors analyzed that Ru has a low oxidation state resulting from the electron transfer. It is well known that oxidation state of active metals, which can modulate the electronic structure of the electrocatalyst, directly affects the water electrolysis performance. Therefore, to improve the reliability of the oxidation state information for the elements analyzed via XPS, other means of analysis such as XANES should be accompanied.

Reply: We are thankful to the reviewer for the kind suggestion.

Figure 5. (a) The Co K-edge XANES spectra and (b) FT-EXAFS spectra of FeCoNiMnRu/CNFs before and after stability test, and Co foil. (c) The corresponding FT-EXAFS fitting curves of FeCoNiMnRu/CNFs before and after stability test. (d) The Ru K-edge XANES spectra and (e) FT-

EXAFS spectra of FeCoNiMnRu/CNFs before and after stability test, and Ru foil. (f) The corresponding FT-EXAFS fitting curves of FeCoNiMnRu/CNFs before and after stability test. WT-EXAFS spectra of Co and Ru foil (g), Co and Ru in FeCoNiMnRu/CNFs before (h) and after stability test (i).

According to your suggestion, we further used the XAS to investigate the chemical states of FeCoNiMnRu/CNFs before and after the HER stability test. As shown in Fig. 5a-f, the pre-absorption edge features for Co and the absorption edge for Ru both are metallicity by comparing with reference metal Co, CoO and Co₃O₄ foils (Fig. 3a-d), demonstrating that the Co and Ru elements in HEA NPs are in metallic state. Meanwhile, after the stability test, the XANES of Co and Ru in HEA NPs also keep metallic state, suggesting the excellent oxidation resistance during the long-term HER stability test. The FT-EXAFS spectra (Fig. 5b and 5d) indicate that the bond structure of Co and Ru in HEA before and after stability test reveal the similar average bond length without any oxidation when compared CoO and Co₃O₄ foils, further confirming the metallic states of Co and Ru in HEA after stability test. According to the EXAFS fitting (Fig. 5c and 5f), the bond length (R) and coordination numbers of each bond type in the HEA before and after stability test were summarized in Supplementary Table 4. The FT-EXAFS and WT-EXAFS spectra (Fig. 5g-i) of Co and Ru in HEA before and after stability tests have negligible mismatch, which means that the FeCoNiMnRu HEA keep metallic states in long-term stability tests, exhibiting extraordinary durability. The reliability of the fitting method is supported by smaller R factors.

7. In Supplementary Figure 11, the authors noted that FeCoNiMnRu HEA has the lowest R_{ct} compared to the control samples through EIS analysis, which was also used to support the following statement, "The low R_{ct} values of HEAs suggest that high-entropy alloying with multiple elements could allow optimization of the electronic structures of the alloy". This appears to be rather abstract for explaining the "optimized electronic structure" since R_{ct} only indicates the interfacial resistance between the electrolyte and the electrode.

Reply: We are thankful to the reviewer for the kind suggestion. Actually, the R_{ct} values were used to demonstrate the excellent performance of FeCoNiMnRu/CNFs together with other electrochemical characterization. According to your advice, authors have revised the inappropriate conclusion.

8. In Supplementary Figure 12, based on C_{dl} of electrocatalysts, ECSA of control samples and FeCoNiMnRu HEA were inferred. However, actual ECSA values are missing. The ECSA data of the FeCoNiMnRu and control samples should be specifically provided.

Reply: We are thankful to the reviewer for the kind suggestion.

Supplementary Figure 20. Double-layer capacitance per geometric area (C_{dl}) of the as-prepared electrodes.

The corresponding electrochemical data have been provided, and we used the electrochemical approach to simulate and calculate the electrochemically active area of the samples to show the intrinsic activity. ECSA-normalized LSV curves were used to estimate the intrinsic activity of FeCoNiMnRu/CNFs and their medium- and low-entropy counterparts. The values of C_{dl} and ECSA of prepared materials have been remeasured. As shown in Supplementary Fig. 20, the C_{dl} values of FeCoNi/CNFs, FeCoNiMn/CNFs, FeCoNiRu/CNFs, and FeCoNiMnRu/CNFs are measured to be 16.4, 62.07, 110.53, 104.57 mF, corresponding to ECSA of 410, 1552, 2763, and 2614 cm^2 . ECSA-normalized LSV curves (Fig. 4c) indicate that the FeCoNiMnRu/CNFs show the highest intrinsic activity than those of FeCoNi/CNFs, FeCoNiMn/CNFs, and FeCoNiRu/CNFs.

Figure 4c. The ECSA-normalized LSV curves of FeCoNi/CNFs, FeCoNiMn/CNFs, FeCoNiRu/CNFs, and FeCoNiMnRu/CNFs in 1 M KOH.

9. In Figure 4f, the full cell polarization curve of the noble metal couple (Pt/C || Ru/C) should be added when discussing the performance improvement of FeCoNiMnRu HEA over the Pt/C || Ru/C.

Reply: We are thankful to the reviewer for the kind suggestion. According to your advice, we have performed the full cell polarization curve of the noble metal couple (Pt/C || Ru/C). However, the Ru/C displayed poor OER activity in 1 M KOH due to the dissolution of metallic Ru at oxidation potential. Instead, we have applied the state-of-art electrocatalyst of OER (RuO_2) as the anode to

acquire the full cell polarization curve. As shown in Supplementary Fig. 25, the FeCoNiMnRu/CNFs || FeCoNiMnRu/CNFs couple only need 400 mV to reach the current density of 50 mA cm⁻², which is superior lower than that of Pt/C || RuO₂.

Supplementary Figure 25. Polarization curves for overall water splitting by FeCoNiMnRu/CNFs || FeCoNiMnRu/CNFs and Pt/C || RuO₂ in a two-electrode configuration in 1 M KOH at a scan rate of 2 mV/s.

10. In Figure 4g and Supplementary Figure 15, the authors claim the operational stability of FeCoNiMnRu HEA based on chronoamperometric (CA) stability (600 hours at 1 A cm⁻²) and CV cycle stability (10000 cycles at 1 M KOH). However, to better support the reliability of the stability analysis, additional verification in various harsh environments (high temperature, high electrolyte concentration, high current density, acidic electrolyte, etc.) is recommended (Nat. Commun., 2018, 9, 2609: <https://doi.org/10.1038/s41467-018-05019-5>; Nat. Commun., 2020, 11, 5462: <https://doi.org/10.1038/s41467-020-19214-w>; Nat. Commun., 2021, 12, 4606: <https://doi.org/10.1038/s41467-021-24829-8>).

Reply: We are thankful to the reviewer for the kind suggestion. According to your advices, authors have performed the stability tests in various harsh environments. All stability tests in this study were performed without iR- compensation and in an intermittent manner, the electrolyte was renewed and the reference electrode was calibrated after every 10 h continuous working. As shown in Supplementary Fig. 28, at 60 °C, the FeCoNiMnRu/CNFs can maintain a high current density of 1000 mA cm⁻² at -2.22 V vs. RHE for 100 h, suggesting no obvious current density degradation. In 10 M KOH, the FeCoNiMnRu/CNFs also can afford a high current density of 1000 mA cm⁻² at -0.77 V vs. RHE for 100 h without current density degradation (Supplementary Fig. 29). For stability test at high current density, we consider that the 1000 mA cm⁻² is high enough for chronoamperometry, which is not lower than the values given in the above provided reference articles. Additionally, the stability tests at 60 °C or in 10 M KOH also apply the high current density of 1000 mA cm⁻², which is even higher than that reported in Nat. Commun., 2021, 12, 4606.

Supplementary Figure 28. The chronoamperometric curve for FeCoNiMnRu/CNFs measured at 60°C and -2.22 V vs. RHE for 100 h in 1 M KOH.

Supplementary Figure 29. The chronoamperometric curve for FeCoNiMnRu/CNFs measured at -0.77 V vs. RHE for 100 h in 10 M KOH.

11. After 600 hours of CA stability test, the authors verified the stability of the electrode via XRD and STEM-EDS mapping analysis. In addition to the structural and compositional analysis, the status of chemical state analysis for the constituent metals after the stability test is also recommended, e.g., change in the oxidation state analyzed by XPS and XANES.

Reply: We are thankful to the reviewer for the kind suggestion.

Figure 5. (a) The Co K-edge XANES spectra and (b) FT-EXAFS spectra of FeCoNiMnRu/CNFs before and after stability test, and Co foil. (c) The corresponding FT-EXAFS fitting curves of FeCoNiMnRu/CNFs before and after stability test. (d) The Ru K-edge XANES spectra and (e) FT-EXAFS spectra of FeCoNiMnRu/CNFs before and after stability test, and Ru foil. (f) The corresponding FT-EXAFS fitting curves of FeCoNiMnRu/CNFs before and after stability test. WT-EXAFS spectra of Co and Ru foil (g), Co and Ru in FeCoNiMnRu/CNFs before (h) and after stability test (i).

According to your suggestion, we further used the XAS to investigate the chemical states of FeCoNiMnRu/CNFs before and after the HER stability test. As shown in Fig. 5a-f, the pre-absorption edge features for Co and the absorption edge for Ru both are metallicity by comparing with reference metal Co, CoO and Co₃O₄ foils (Fig. 3a-d), demonstrating that the Co and Ru elements in HEA NPs are in metallic state. Meanwhile, after the stability test, the XANES of Co and Ru in HEA NPs also keep metallic state, suggesting the excellent oxidation resistance during the long-term HER stability test. The FT-EXAFS spectra (Fig. 5b and 5d) indicate that the bond structure of Co and Ru in HEA before and after stability test reveal the similar average bond length without any oxidation when compared CoO and Co₃O₄ foils, further confirming the metallic states of Co and Ru in HEA after stability test. According to the EXAFS fitting (Fig. 5c and 5f), the bond length (R) and coordination numbers of each bond type in the HEA before and after stability test were summarized in Supplementary Table 4. The FT-EXAFS and WT-EXAFS spectra (Fig. 5g-i) of Co and Ru in HEA before and after stability tests have negligible mismatch, which means that the FeCoNiMnRu HEA keep metallic states in long-term stability tests, exhibiting extraordinary durability. The reliability of the fitting method is supported by smaller R factors.

12. FeCoNiMnRu HEA showed full cell performance that outperformed the noble-metal couple (Pt/C || Ru/C). To better elucidate the water electrolysis performance of FeCoNiMnRu HEA, please calculate and provide the energy efficiency of FeCoNiMnRu (Int. J. Hydrog. Energy, 2010, 35(20), 10851-10858).

Reply: We are thankful to the reviewer for the kind suggestion. According to your suggestion, we used the Faradaic efficiency to show the water electrolysis performance of FeCoNiMnRu HEA. The gas products of working electrode compartment were analyzed by directly injected into the gas chromatograph (GC, Agilent 7890B), which was equipped with thermal conductivity detector (TCD) for H₂. The electrolysis at each potential was repeated three times, and the average value of experiments was adapted as the gas concentration for Faradaic efficiency (FE) calculation. The Faradaic efficiency (FE) of H₂ production were calculated as 99.6 %, suggesting the excellent efficiency for H₂ production.

Reviewer #4 (Remarks to the Author):

I was asked to review the submitted manuscript “Unraveling the interactions between active sites and intermediates in high-entropy alloy electrocatalysts: How electronegativity matters”, written by Hao & Zhuang et al. and submitted to Nature Communications.

The manuscript is very well written and gives a very comprehensive insight into catalytic procedures of high entropy alloys. The authors produced a network of carbon nanofibers, partially

coated with HEA particles and used this compound as catalyst for HER and OER reactions. Additionally, they used operando and computational methods to explain the behavior of the material and drew a connection between electronegativity and electrochemical performance of the individual incorporated elements. Without a doubt, I recommend a publication in Nature Communications after some revisions.

Reply: We are thankful to the reviewer for the kind recommendation and careful review. (All the changes are labeled in red).

1. Since the surface of the catalyst is the most important “part” of the particle when it comes to catalytic reactions, the authors should explain why they are preparing a CNF/HEA network instead using pure HEA particles. The incorporation of the HEAs in the CNF network reduces the addressable surface of the HEAs, since they are attached (or even fully included sometimes?) to the CNF as shown in the SEM micrographs.

Reply: We are thankful to the reviewer for the kind suggestion. The referee’s concerns are reasonable. Bulk HEA are excellent structural material and the first thing for HEA using as electrocatalyst is to reduce the size of HEA. In recently years, the nanostructured HEAs usually synthesized on substrates, such as carbon support. It is because that alloying five or more elements into a single-phase HEA is very difficult in liquid phase synthesis, and the as-synthesized multicomponent metal alloys nanoparticles are usually phase separated due to the near equilibrium reaction. In this work, the CNFs can serve as nano-reactor for the confined growth of HEA nanocrystals. FeCoNiMnRu/CNFs is a kind of self-supported electrocatalyst.

The integration of metal nanocatalysts into electrode materials is of great significance. 3D self-supporting electrodes have many advantages: (1) supported nanocrystals can greatly increase the specific surface area of the material, expose active sites, and facilitate electrolyte diffusion and gas release; (2) *In situ* growth of nanocrystal is conducive to electron transport, can improve electrode conductivity, while preventing the catalyst from peeling off, improving catalyst stability. (3) CNFs network possess porous structure, which can facilitate the diffusion of electrolyte and provide a large area for the contact between the electrode and electrolyte.

2. When using such a compound, how do the authors determine the catalytic activity? Is the activity normalized to a measured surface, or the mass of the complete material or the mass of the HEA particles?

Reply: We are thankful to the reviewer for the kind suggestion. According to your advice, authors have determined the catalytic activity of HEA/CNFs by three different methods. Firstly, the polarization curves of the self-supported HEA/CNFs electrodes were normalized by geometric area, and the catalytic activity of all the electrocatalysts were estimated by the overpotentials at 10 or 100 mA cm⁻². Secondly, we have calculated the electrochemically active surface area (ECSA) from C_{dl} measured by CV. The ECSA-normalized LSV curves were acquired to which intrinsic activity of electrocatalysts can be evaluated. Thirdly, the mass activity of FeCoNiMnRu/CNFs was calculated by ICP results of HEA NPs. The catalytic activity of the samples indicates the same trend that HEA NPs show the best activity.

3. Is the comparison of Figure 4e “fair”? When I understood the experimental part correctly, the authors are using a free standing electrode and compare it to mostly tape casted electrodes. What

are the differences in surface between the tape casted and their free standing electrodes? I think that more experimental details about the electrode preparation should be given. Is it possible to compare an “architected” electrode with a nice 3D carbon network to a 2D tape casted electrode?

Reply: We are thankful to the reviewer for the kind suggestion. The HEA or other alloy NPs were synthesized on substrate CNFs and the NP and CNFs were a whole self-supported electrode. In order to fairly compare the catalytic activity of reported electrocatalysts, all of the compared electrodes in Figure 4e were replaced by free standing electrode. The study of self-supporting electrodes is very important in electrocatalysis. Most of the powder catalysts usually applied the 2D tape casted electrode (such as Au and GCE) and a coupling agent such as Nafion or PVDF is required to connect the catalyst and the electrode. This causes a part of the active sites of the catalyst to be wrapped by the coupling agent and hinders the transmission of gas, electron ions, etc., thereby reducing the catalytic efficiency. In addition, self-supporting electrodes have many advantages, such as conducive to electron conduction, no coupling agent, more active sites, and not easy to fall off.

Figure 4. (e) Comparison of HER and OER overpotentials at 10 mA cm⁻² in 1.0 M KOH for different catalysts.

The establishment of a reasonable activity measurement and the correct evaluation of the electrocatalytic performance of the catalyst are of great significance to designing advanced electrocatalysts. We usually define intrinsic activity as specific activity, that is, the current passing through a unit of catalyst surface area. Therefore, an accurate assessment of the specific activity of an electrocatalyst is highly dependent on a reliable measurement of the surface area of the catalyst. The so-called electrochemically active area is a concept relative to the geometric area. For traditional 2D tape casted electrode, the active site is considered to be the part of the outer surface of the catalyst electrode that is in contact with the electrolyte. Therefore, it can be considered that the geometric area is approximately equal to the electrochemically active area. But for some nano-material electrodes, especially some porous electrodes, such as self-supported electrodes, gas diffusion layer electrodes, etc., after loading nano-catalysts, their specific surface area is very large, and abundant active sites are exposed, which will inevitably lead to rapidly enhanced electrochemical performance. At this time, if the geometric area is used to calculate the current density, the activity of the catalyst cannot be truly reflected, and the performance of different catalysts cannot be compared well. So to put it plainly, an ideal electrochemical active area is the area where the active sites of the catalyst are in contact with the electrolyte. However, since many active sites are not a simple layer, it is more difficult to measure by physical means. Therefore, there are many electrochemical approaches to simulate and calculate the electrochemically active area. More experimental details about the electrode preparation have been added in the

experimental section. Comparing an “architected” electrode with a nice 3D carbon network to a 2D tape casted electrode need a combined evaluation test of electrocatalytic activity: electrochemical surface area (ECSA), mass specific activity (MA), area specific activity (SA).

4. A very important part is the materials characterization as well. In my opinion, the XRD part needs some improvements.

- In both pattern for FeCoNiMnRu (1000°C, 3h) a strong asymmetry of the 111 reflection is observable. Such shoulders often can indicate byphases, also the 200 reflection reveals a very small shoulder. Where do these shoulders come from?

Reply: We are thankful to the reviewer for the kind suggestion. High-entropy alloys (High-entropy alloys) referred to as HEAs are alloys formed by five or more equal or approximately equal amounts of metals. For example, the five-fold elements of A, B, C, D, and E are mixed in close to 20%. The original people generally would not design materials in this way, because the content of all elements is too much, it is easy to segregate, it is easy to generate complex harmful phases, and the cost will also be significantly increased. However, from the perspective of basic research, in another metal solid solution, there is no distinction between which element is the solute and which element is the solvent. All elements are "equal". This structure sounds interesting. When a variety of alloying elements are mixed at nearly equal atomic ratios to form a single-phase solid solution structure, the lattice structure of HEA is determined by the difference in the atomic radius between the constituent atoms and the mismatch between the physical properties such as elastic modulus. According to the empirical criteria proposed by Hume-Rothery, the formation of a single-phase solid solution in traditional binary alloys requires that the difference between the solute and the solvent's atomic radius should be less than 15%. Similarly, in high-entropy alloys, when the radii of different constituent atoms are too different, lattice distortion is inevitable. Too high lattice distortion will destabilize the single-phase solid solution structure, leading to the precipitation of the second phase or even amorphization.

In HEAs, the interaction between atoms of different components often makes the arrangement of atoms deviate from strict periodic symmetry. Therefore, the local unit cell structure in the distorted lattice will change with different spatial positions. The degree of deviation from the ideal unit cell reflects the distortion of the local lattice. The lattice type and lattice constant information obtained in the experiment often reflect the overall average effect of the crystal space group, which is not enough to reveal the local distortion information. In addition, the "distortion" of the lattice refers to the degree of deviation of the position of each constituent atom from the lattice of the "ideal lattice". In this work, *in situ* XRD characterization is a fast process with heating rate of 50 °C min⁻¹, which can help us to investigate the early stage of HEA formation mechanism. According to your advice, we have performed the HRXRD of FeCoNiMnRu/CNFs-800-3h and FeCoNiMnRu/CNFs-1000-3h with a scan rate of 0.5 °C min⁻¹ and the corresponding detailed Rietveld refinements. As shown in Fig. 2f and Supplementary Fig. 13, both of the FeCoNiMnRu HEA NPs obtained at 800 °C and 1000 °C under 3h prolonged treatment are single phase structure with similar component, and the simulative components of FeCoNiMnRu NPs models matches well with the results of ICP. Therefore, the asymmetry of the 111 reflection and shoulders were attributed to the lattice distortion caused by radii of different constituent atoms. The lattice distortion in HEA can not be completely eliminated and however, in this work, we show a prolonged heating treatment approach that can reduce the lattice distortion in HEA NPs.

Figure 2f. XRD pattern of FeCoNiMnRu/CNFs (1000 °C-3 h) with a scan rate of 0.5° /min and the corresponding XRD Rietveld refinement.

Supplementary Fig. 13. XRD pattern of FeCoNiMnRu/CNFs (800 °C-3 h) with a scan rate of 0.5° /min and the corresponding XRD Rietveld refinement.

- Between FeCoNiMnRu (1000°C, 3h) and FeCoNiMnRu huge shifts of the reflections can be seen. The authors argue with the introduction of the Mn_3Co_7 phase into the single phase structure. However, as can be seen, even the Mn_3Co_7 phase already seems to be a high entropy structure, since the reflections are shifted as well. Therefore, can such a huge shift when one high entropy structure with mixed elements is introduced into another one with mixed elements, really be explained using this argument? Most probably all elements are apparent in both phases already, so an incorporation would not really lead to a change of the lattice parameters.

- Even if the incorporation of the Mn_3Co_7 phase leads to a shrinking of the lattice parameters, how do the authors explain the extension of the lattice parameter (between 900 and 1000 °C), although the intensity of the Mn_3Co_7 reflection is shrinking? This would mean that the Mn_3Co_7 phase is vanishing, most probably incorporated as the authors state, and the lattice parameters are growing. This is in contrast to the next step when the same incorporation should lead to the explained lattice shrinking.

Therefore, I would strongly recommend doing detailed Rietveld refinements and HR XRD measurements to identify the mechanism and to rule out that additional phases are in the compound.

Reply: We are thankful to the reviewer for the kind suggestion. According to your advice, we Therefore, we proposed a possible growth process of HEA NPs. During graphitization, the Fe/Co/Ni/Mn/Ru mixed metal precursors decomposed first, and then the reduced metal clusters were bonded and confined within the PAN-derived CNFs. At relative low temperature 600-800 °C, the metal elements with small atom radii differences prefer to form alloy phase and insufficient

heating energy at low temperature cause the slightly atom diffusion, which make both of HEA and Mn_3Co_7 phases co-exist. At high temperature 1000 °C, sufficient dynamic energy caused the metal atoms to diffuse dramatically, leading to homogeneous formation of single-phase HEA alloy. It is concluded that the high temperature coupled with prolonged heating treatment provided the activation energy that drove complete mixing of multiple metal element atoms. Therefore, negatively shifted peak position through *in situ* XRD results from 800 to 1000 °C suggest the growth process of single-phase HEA. The XRD results from 1000 °C to 1000-3h exhibit a positively shifted peaks position, further show a structural symmetry optimization by reducing the lattice distortion in HEA NPs. The XRD patterns of FeCoNiMnRu/CNFs synthesized at 800, 900, and 1000 °C under prolonged heat treatment for 3 h were also performed. As shown in Supplementary Fig. 12, compared with the *in situ* XRD patterns of FeCoNiMnRu/CNFs without prolonged heat treatment, all the diffraction peaks for *fcc* HEA NPs ((111), (200), (220) planes) exhibit positive shifts to high values, suggesting the reduced lattice parameters and lattice distortion. It is indicated that after the prolonged heat treatment for 3 h, all of the diffraction peaks for Mn_3Co_7 phase vanished, suggesting the complete formation of FeCoNiMnRu HEA.

Supplementary Figure 12. XRD patterns of FeCoNiMnRu/CNFs obtained after prolonged heat treatment (upper) and during *in situ* characterization (lower) at (a) 800 °C, (b) 900 °C, and 1000 °C.

REVIEWER COMMENTS

Reviewer #1 (Remarks to the Author):

In the revised manuscript, the authors provided many supporting data to strength their claim of study. However, this seems to be more technical paper for HEA community. The synthesis method and characterization for water splitting are not new at all. I suggest a more technical journal.

The technical comments are as follows;

- (1) The performance of Pt/C for HER is underestimated too much (Fig. 4a and Fig. S23). Reviewer#2 also pointed out although the data is not corrected. In addition, the performance of their sample such as FeCoNiMnRu is over-evaluated (too good). The authors need to do fair experiments and analysis.
- (2) The mass activity of their FeCoNiMnRu/CNFs is too high. The details of calculation procedure such as loading mass and so on must be shown.
- (3) Why is the chemical structure (Fig. S31) of FeCoNiMnRu HEA random ?? The explanation is needed.
- (4) No details on RuO₂. Powder ? RuO₂/C ? What is the weight %?
It is unfair to compare with the powder samples (the performance is of course bad).
- (5) IrO₂/C is more common benchmark for OER than RuO₂ powders.

Reviewer #2 (Remarks to the Author):

The authors have added a substantial amount of extra data and analysis in response to concerns from the reviewers. From my side, adding reference samples and reanalyzing the X-ray spectroscopy data, adding a Bader analysis comparison to support their choice of electronegative scale in rationalizing their results, and adding randomized /mixed high-entropy structures are important additions. The authors have also added d-band calculations, electrochemical surface area (ECSA) normalized catalytic currents, as well as other improvements and reference experiments. In summary, my view is that the manuscript is now much improved and warrants publication in Nature communications after some minor corrections.

1. Ref 28 is still not the correct reference for the stated Pauling electronegativities, the study do not mention Ru, and is instead performing Bader analysis to prove their charge transfer towards Co (in their Ru free system). I advise the authors to instead base their results on the Delta G calculations and the Bader analysis instead as of now stating "...due to the higher electronegativity of Ru ...". The authors are of course free to state that this is in line with the simpler notion of Pauling electronegativity, but should then be references to a study including these.

2. Some minor formulations and spellings, e.g. "... there are more electrons can be observed" (line 500), Mu instead of Mn (line 502), "...display the lowest values than ..." instead of "...display lower values than

..." (line 540-541), and "...as represent sample to show ..." instead of "as a representative sample to show..." (line 547), but these and others can also be handled in a proof-process.

Reviewer #3 (Remarks to the Author):

In this manuscript, the authors reported a study on the functionality of HEA as an electrocatalyst for water splitting. The reviewer's concerns/comments regarding the electrochemical performance and stability, oxidation states, local electronic structures, and thermodynamically driven phase transition depending on the temperature of annealing treatment of HEA were properly addressed in the revised manuscript. Therefore, it is considered that the manuscript is now ready for publication in Nature Communications.

Response to the comments

Reviewer #1 (Remarks to the Author):

In the revised manuscript, the authors provided many supporting data to strengthen their claim of study. However, this seems to be more technical paper for HEA community. The synthesis method and characterization for water splitting are not new at all.

I suggest a more technical journal.

Reply: We are thankful to the reviewer for the kind suggestion. The application of HEA nano-materials in the field of electrocatalysis has been demonstrated to be viable in recent 4 years (Y. Yao, *et al.*, *Science* 2018, 359, 1489-1494). Fundamental understandings of the generation process of solid solution phase and the specific role of different sites in HEA system are urgently desired. As reported by Y. Yao, *et al.*, the HEA NPs can be synthesized by the advanced “carbothermal shock” method which employs flash heating and cooling (temperature of ~2000 K, shock duration of ~55 ms, and ramp rates on the order of 10^5 K/s) of metal precursors on oxygenated carbon support. However, the metal precursors are loaded onto the carbon support by dipping the CNF into the mixed metal salt solutions, causing the uncontrollable stoichiometric ratio of HEA NPs resulting from the different adsorption strength of CNF to different metal ions. In addition, the fine structure (*e.g.*, strain, morphology, nanointerfaces) are difficult to be adjusted and the detailed generation process of HEA are hardly detectable. Based on the work of our group in the past decade (Han Zhu, *et al.*, *J. Mater. Chem.*, 2012, 22, 9301; Han Zhu, *et al.*, *Adv. Mater.* 2015, 27, 4752; Han Zhu, *et al.*, *Energy Environ. Sci.*, 2017, 10, 321; Han Zhu, *et al.*, *Adv. Mater.* 2018, 30, 1707301; Han Zhu*, *et al.*, *Adv. Energy Mater.*, 2021, 11, 2102138; Han Zhu*, *et al.*, *ACS Nano*, 2022 DOI: 10.1021/acsnano.1c11145), various electrocatalysts with delicate nanostructure have been acquired by utilizing the CNFs as *in-situ* nanoreactor. Therefore, we proposed the novel “polymer-chain confining” strategy for the synthesis of HEA NPs, which is distinguished from Y. Yao’s work. Specifically, various metal ions are coordinated to the polymer chain in precursor nanofibers and subsequently converted into HEA NPs with the assistance of confining effect origins from polymer-derived graphitic layers. This strategy can provide the excellent stoichiometric control in HEA NPs and the load amounts of the HEA on polymer derived CNFs can be accurately adjusted. In addition, although various novel methods have been reported for the preparation of nanoscale HEAs (Han Zhu, *et al.*, *Small Struct.* 2020, 2000033), few reports have shed light on the growth process of HEA. Herein, by virtue of the “polymer-chain confining” strategy, the selective evolution of nanocrystals with specific phases can be realized driven by the controllable temperature program. With the assistance of *in-situ* and *ex-situ* XRD technique, the thermodynamically driven phase transition process and the strain-relaxation effect of cooling process were unveiled, demonstrating that our approach ensure the characterization of the transition state and the fine adjustment of the microstructure of the HEA NPs. We have reported the strain relaxation phenomena in HEA NPs driven by the controllable

temperature program. Therefore, the novel “polymer-chain confining” strategy can provide guidance for the designing and synthesis of HEA-based electrocatalysts.

In regard to HEA nano-materials as electrocatalysts for water splitting (Z. Jia, *et al.*, *Adv. Mater.* 2020, 32, 2000385; H. Qiu, *et al.*, *J. Mater. Chem. A*, 2019, 7, 6499. Z. Jin, *et al.*, *Small*, 2019, 15, 1904180), only preliminary electrochemical tests have been conducted and the origin of the obtained excellent performance were simply ascribed to the “synergistic effect” or “high-entropy effect” derived from multicomponent. Therefore, electrocatalytic mechanism on HEA are still extremely rare and discussions on the mechanism and the origin of the excellent activities are usually deficient. Meanwhile, the configurational complexity makes the identification of exact active sites difficult and complicated. How do the multi-element HEA work, how do the different metal sites in HEA adsorb species during electrolysis and what the roles do metal sites play? A lot of questions should be considered and therefore, in-depth investigations on the mechanism understanding and performance regulation are urgently deserved. In this work, we have elaborately investigated the electrocatalytic mechanism both experimentally and theoretically. We are the first example to unraveling the role of each metal sites in HEA for electrocatalytic water splitting and correlate the electronegativity with activity, which opens up a new direction for breaking scaling relation issues for multistep reactions.

With the development of electrocatalysis, products with more and more complex chemical structures have been synthesized through complicated electrochemical processes which involve multiple intermediates. For example, carbon dioxide reduction (CRR) featured with a wide product distribution can produce alkane, alkene, alcohol, and so on. Combining with electrochemical reduction of nitrogen-containing precursors (*e.g.* NH_3 , N_2 , NO_x , NO_2^- , NO_3^-), diverse N-containing products (*e.g.* urea, amine, and amide) can be acquired through electrochemical C–N coupling (Shuangyin Wang, *et al.*, *Nat. Chem.* 2020, 12, 717; Hailiang Wang, *et al.*, *Nat Sustain* 2021, 4, 725; Feng Jiao, *et al.*, *Nat. Chem.* 2019, 11, 846). The gradual complication of electrochemical reaction and final products (*e.g.* 14 electrons and 15 protons are needed for forming a methylamine molecule from CO_2 and NO_3^-) is predictable. Therefore, the new generation of electrocatalysts with more sophisticated design and multiple active sites for selectively absorbing special intermediates are urgently demanded. However, HEA nano-materials, as an ideal platform for the designing of multisite electrocatalysts, have seldomly applied in the field of CRR, NRR, NO_3^- -RR, or the electrosynthesis of N-containing products due to the difficult of electrocatalytic mechanism analysis caused by the complexity of configuration. In this work, the water splitting was used as model reaction to unravel the role of each metal sites in HEA, and we provide the basic roles for design advanced electrocatalysts for other reactions, such as CRR, NRR and so on. This work will attract more attentions in nanoscience, materials science, electrochemistry, catalysis science and it will lead to hot topics in active sites recognition in HEA catalysts.

Compared with very recently reported noble metal based HEA works for water splitting in Nature Synthesis (Zhigang Zou, *et al.*, *Nat Synth* 2022, 1, 138) and JACS (Dongshuang Wu *et al.*, *J. Am. Chem. Soc.* 2022. DOI: 10.1021/jacs.1c13616), we obtained

comparable electrochemical performances with very low loading amounts of noble metal. Furthermore, our work provides the excellent stoichiometric control in HEA NPs and the load amounts of the HEA on polymer derived CNFs can be accurately adjusted. We have reported the strain relaxation in HEA NPs driven by the controllable temperature program. We have unraveled the role of each metal sites in HEA for electrocatalytic water splitting and correlate the electronegativity with activity, which opens up a new direction for breaking scaling relation issues for multistep reactions. The activities of our FeCoNiMnRu/CNFs are comparable to the PtIrCuNiCr and the RuRhPdAgOsIrPtAu, which can be considered to be one of the best electrocatalysts for water splitting. Noted that the mass fraction of noble metal in FeCoNiMnRu/CNFs is far lower than those of reported PtIrCuNiCr and RuRhPdAgOsIrPtAu electrocatalysts. We highly believe that the additional revisions substantially strengthen and improve our manuscript, and hope that you could kindly reconsider this revised version for the publication in Nature Communications.

The technical comments are as follows;

(1) The performance of Pt/C for HER is underestimated too much (Fig. 4a and Fig. S23). Reviewer#2 also pointed out although the data is not corrected. In addition, the performance of their sample such as FeCoNiMnRu is over-evaluated (too good). The authors need to do fair experiments and analysis.

Reply: We are thankful to the reviewer for the kind recommendation and careful review. According to your advice, author have checked the electrochemical data carefully. The great distinction between FeCoNiMnRu/CNFs and Pt/C in Fig. 4a is caused by the difference in electrode structure beyond the difference in intrinsic activity. The former is the self-supported materials with HEA NPs supported on porous CNFs member, while the latter, *i.e.* commercial Pt/C, is dropped on GCE. The porous CNFs member with 3D networks can facilitate the electron transfer and diffusion of electrolyte, which can significantly boost the e efficiency of electrocatalysts.

Actually, we have evaluated the activities of electrocatalysts in three ways for the objective comparison in former revised version to reply reviewer #2 and 3. Except the LSV curves normalized by the geometric areas of electrodes, ECSA-normalized LSV curves and mass activities were also measured. The reviewer 2 and 3 have been satisfied with our revised evaluation for electrochemical performance. In addition, the mass activities of GCE supported FeCoNiMnRu/CNFs, Pt/C, RuO₂, and IrO₂ at selected potential were displayed in Supplementary Fig. 23. After extensive investigation, we found that the mass activities of state-of-art electrocatalysts in alkaline electrolyte in other reports are on the same level with recently literatures (Zhigang Zou, *et al.*, *Nat Synth* 2022, 1, 138–146; Yongwen Tan, *et al.*, *ACS Energy Lett.* 2020, 5, 192–199; Shaojun Dong, *et al.*, *Nano Energy* 2020, 68, 104296; Qianwang Chen, *et al.*, *ACS Catal.* 2020, 10, 1152–1160; Ji Yang, *et al.*, *Sci Rep* 2016, 6, 38429; Daniel Böhm, *et al.*, *Adv. Funct. Mater.* 2020, 30, 1906670). Therefore, the performance of Pt/C for HER is reasonable and it was not underestimated. To be more convincing, the complete LSV curves of GCE supported FeCoNiMnRu/CNFs, Pt/C, RuO₂, and IrO₂ normalized by geometric area (0.07065 cm²) and mass loading of noble metal are presented as follow.

The detailed calculation process is clearly shown as equation 1-2:

$$J_{\text{mass}} = J_{\text{geo}} \times 0.07065 \text{ cm}^2 / \text{m} \quad \text{equ (1),}$$

$$m = 3 \text{ mg} \times \omega \times (5 \mu\text{L} / 1000 \mu\text{L}) \quad \text{equ (2),}$$

where m is the mass loading of noble metal (Ru, Pt, or Ir), 3 mg is the mass of electrocatalysts used in ink preparation, ω is the mass fraction of noble metal, 5 μL is the electrocatalyst ink dropped onto the GCE, 1000 μL is the total volume of the as-prepared electrocatalyst ink. The ω for FeCoNiMnRu/CNFs, Pt/C, RuO₂, and IrO₂ are 4.383, 20.000, 75.958, and 85.727 wt.%, respectively. Noted that the mass fraction of noble metal in FeCoNiMnRu/CNFs is far lower than that in the state-of-art electrocatalysts.

Supplementary Fig. 23. LSV curves of GCE supported FeCoNiMnRu/CNFs and 20 wt% Pt/C for HER normalized by (a) geometric area and (b) mass loading of noble metal (Ru or Pt). (c) The mass activities of FeCoNiMnRu/CNFs and 20 wt% Pt/C for HER at overpotentials of 200 and 300 mV. LSV curves of GCE supported FeCoNiMnRu/CNFs, RuO₂ powder, and IrO₂ powder for OER normalized by (d) geometric area and (e) mass loading of noble metal (Ru or Ir). (f) The mass activities of FeCoNiMnRu/CNFs, RuO₂ powder, and IrO₂ powder for OER at overpotentials of 400 and 450 mV.

As shown in Supplementary Fig. 23, the HER and OER LSV curves normalized by geometric area and mass loading of noble metal indicate that the FeCoNiMnRu/CNFs exhibit outstanding electrocatalytic activities and even prominently surpass the state-of-art Pt/C (20 wt%), IrO₂ and RuO₂ electrocatalysts at high current density. At overpotential of 300 mV for HER the FeCoNiMnRu/CNFs show higher mass activity of 2666 than that of Pt/C (1688 mA mg_{Pt}⁻¹). In addition, at overpotential of 450 mV for OER, the FeCoNiMnRu/CNFs obtain the highest mass activity of 487 mA mg_{Ru}⁻¹, which is significantly higher than RuO₂ (146 mA mg_{Ru}⁻¹), and IrO₂ (49 mA mg_{Ir}⁻¹). Recently, Zhigang Zou, *et al.* also estimated the mass activities of PtAuPdFeNi for HER and OER (Zhigang Zou, *et al.*, *Nat Synth* 2022, 1, 138). Obviously, the reported PtIrCuNiCr reaches 8560 mA mg⁻¹ at -0.5 V for HER and 6080 mA mg⁻¹ at 1.6 V for OER, which are 10.8- and 11.9-fold higher than those of Pt/C and Ir/C. Similar to our FeCoNiMnRu, the PtIrCuNiCr also dramatically surpasses the state-of-art electrocatalysts for both HER and OER. Dongshuang Wu *et al.* also reported that the NM-HEA NPs

(RuRhPdAgOsIrPtAu) showed 10.8-times higher intrinsic activity for HER than commercial Pt/C (Dongshuang Wu *et al.*, *J. Am. Chem. Soc.* 2022. DOI: 10.1021/jacs.1c13616). The activities of our FeCoNiMnRu/CNFs are comparable to the PtIrCuNiCr and the RuRhPdAgOsIrPtAu, which can be considered to be one of the best electrocatalysts for water splitting. Therefore, the performance of Pt/C for HER is not underestimated and the performance of their sample such as FeCoNiMnRu is not over-evaluated. It is a fact that HEA have advantages in high current density for water splitting.

The chronoamperometric curve (Fig. 4g) for FeCoNiMnRu/CNFs was measured at -1.16 V vs. RHE for more than 600 h in 1.0 M KOH. The current density at 1 A cm⁻² displays no evident changes, also suggesting its remarkable stability. This is ascribed to the unique corrosion resistance of HEA structures. Additionally, our group have explored advanced HER and OER electrocatalysts for nearly 10 years and many excellent works have published (Han Zhu, *et al.*, *J. Mater. Chem.*, 2012, 22, 9301; Han Zhu, *et al.*, *Adv. Mater.* 2015, 27, 4752; Han Zhu, *et al.*, *Energy Environ. Sci.*, 2017, 10, 321; Han Zhu, *et al.*, *Adv. Mater.* 2018, 30, 1707301; Han Zhu*, *et al.*, *Adv. Energy Mater.*, 2021, 11, 2102138; Han Zhu*, *et al.*, *ACS Nano*, 2022 DOI: 10.1021/acsnano.1c11145), which can guarantee the authenticity of this work.

(2) The mass activity of their FeCoNiMnRu/CNFs is too high. The details of calculation procedure such as loading mass and so on must be shown.

Reply: We are thankful to the reviewer for the kind suggestion. As mentioned above, the complete LSV curves normalized by mass loading of noble metal of GCE supported FeCoNiMnRu/CNFs, Pt/C, RuO₂, and IrO₂ have been displayed, and the calculation process is clearly shown as equation 1-2:

$$J_{\text{mass}} = J_{\text{geo}} \times 0.07065 \text{ cm}^2 / \text{m} \quad \text{equ (1),}$$

$$\text{m} = 3 \text{ mg} \times \omega \times (5 \text{ } \mu\text{L} / 1000 \text{ } \mu\text{L}) \quad \text{equ (2),}$$

where m is the mass loading of noble metal (Ru, Pt, or Ir), 3 mg is the mass of electrocatalysts used in ink preparation, ω is the mass fraction of noble metal, 5 μL is the electrocatalyst ink dropped onto the GCE, 1000 μL is the total volume of the as-prepared electrocatalyst ink. The ω for FeCoNiMnRu/CNFs, Pt/C, RuO₂, and IrO₂ are 4.383, 20.000, 75.958, and 85.727 wt.%, respectively. Noted that the mass fraction of noble metal in FeCoNiMnRu/CNFs is far lower than that in the state-of-art electrocatalysts.

As shown in Supplementary Fig. 23, the HER and OER LSV curves normalized by geometric area and mass loading of noble metal indicate that the FeCoNiMnRu/CNFs exhibit outstanding electrocatalytic activities and even prominently surpass the state-of-art Pt/C (20 wt%), IrO₂ and RuO₂ electrocatalysts at high current density. At overpotential of 300 mV for HER the FeCoNiMnRu/CNFs show higher mass activity of 2666 than that of Pt/C (1688 mA mg_{Pt}⁻¹). In addition, at overpotential of 450 mV for OER, the FeCoNiMnRu/CNFs obtain the highest mass activity of 487 mA mg_{Ru}⁻¹, which is significantly higher than RuO₂ (146 mA mg_{Ru}⁻¹), and IrO₂ (49 mA mg_{Ir}⁻¹). Recently, Zhigang Zou, *et al.* also estimated the mass activities of PtAuPdFeNi for HER and OER (Zhigang Zou, *et al.*, *Nat Synth* 2022, 1, 138). Obviously, the reported PtIrCuNiCr reaches 8560

mA mg⁻¹ at -0.5 V for HER and 6080 mA mg⁻¹ at 1.6 V for OER, which are 10.8- and 11.9-fold higher than those of Pt/C and Ir/C. Similar to our FeCoNiMnRu, the PtIrCuNiCr also dramatically surpasses the state-of-art electrocatalysts for both HER and OER. Dongshuang Wu *et al.* also reported that the NM-HEA NPs (RuRhPdAgOsIrPtAu) showed 10.8-times higher intrinsic activity for HER than commercial Pt/C (Dongshuang Wu *et al.*, *J. Am. Chem. Soc.* 2022. DOI: 10.1021/jacs.1c13616). The activities of our FeCoNiMnRu/CNFs are comparable to the PtIrCuNiCr and the RuRhPdAgOsIrPtAu, which can be considered to be one of the best electrocatalysts for water splitting. Therefore, the performance of Pt/C for HER is not underestimated and the performance of their sample such as FeCoNiMnRu is not over-evaluated. It is a fact that HEA have advantages in high current density for water splitting.

Supplementary Fig. 23. LSV curves of GCE supported FeCoNiMnRu/CNFs and 20 wt% Pt/C for HER normalized by (a) geometric area and (b) mass loading of noble metal (Ru or Pt). (c) The mass activities of FeCoNiMnRu/CNFs and 20 wt% Pt/C for HER at overpotentials of 200 and 300 mV. LSV curves of GCE supported FeCoNiMnRu/CNFs, RuO₂ powder, and IrO₂ powder for OER normalized by (d) geometric area and (e) mass loading of noble metal (Ru or Ir). (f) The mass activities of FeCoNiMnRu/CNFs, RuO₂ powder, and IrO₂ powder for OER at overpotentials of 400 and 450 mV.

Furthermore, due to the multiply components of HEA, the mass fraction of noble metal (Ru) in HEAs-based materials are obviously lower than Pt/C, RuO₂, and IrO₂. Therefore, the mass activity of FeCoNiMnRu/CNFs measured in this paper is reasonable and . Our aim is to reduce the loading amounts of Ru contents and simultaneously improve the electrochemical performance. The results suggested that design HEA structures is an intriguing approach to realize these goals.

For example, Zhigang Zou, *et al.* also estimated the mass activities of PtAuPdFeNi for HER and OER (Zhigang Zou, *et al.*, *Nat Synth* 2022, 1, 138). Obviously, the PtIrCuNiCr reaches 8560 mA mg⁻¹ at -0.5 V for HER and 6080 mA mg⁻¹ at 1.6 V for OER, which are 10.8- and 11.9-fold higher than those of Pt/C and Ir/C. Similar to our FeCoNiMnRu,

PtIrCuNiCr also dramatically surpasses the state-of-art electrocatalysts for both HER and OER. The activities of FeCoNiMnRu/CNFs are comparable to the PtIrCuNiCr and the RuRhPdAgOsIrPtAu, which can be considered to be one of the best electrocatalysts for water splitting. In our work, the chronoamperometric curve (Fig. 4g) for FeCoNiMnRu/CNFs was measured at -1.16 V vs. RHE for more than 600 h in 1.0 M KOH. The current density at 1 A cm⁻² displays no evident changes, also suggesting its remarkable stability. This is ascribed to the unique corrosion resistance of HEA structures.

Fig. S24 Mass activities of a, PtIrCuNiCr and Pt/C for HER, and b, PtIrCuNiCr and Ir/C for OER. [Zhigang Zou, *et al.*, *Nat Synth* 2022, 1, 138]

Dongshuang Wu *et al.* also reported that the NM-HEA NPs (RuRhPdAgOsIrPtAu) showed 10.8-times higher intrinsic activity for hydrogen evolution reaction than commercial Pt/C (Dongshuang Wu *et al.*, *J. Am. Chem. Soc.* 2022. DOI: 10.1021/jacs.1c13616).

Figure S10. Electrochemical data. (a) Geometric activities in 0.15 M H₂SO₄ into HER range. Scan rate: 5 mV/s. [Dongshuang Wu *et al.*, *J. Am. Chem. Soc.* 2022. DOI: 10.1021/jacs.1c13616]

(3) Why is the chemical structure (Fig. S31) of FeCoNiMnRu HEA random?? The explanation is needed.

Reply: We are thankful to the reviewer for the kind suggestion. XRD patterns suggest

the fcc structures of the HEA. STEM-EDS mapping indicate that the atomic distributions in HEA are random and uniform (the precursors are uniformly blended before alloy formation). In the calculation model, the atomic arrangement of the alloy in the HEA model structure is difficult to construct, and the configuration of the alloy is not unique. In the calculation, AIMD method is adopted to randomly construct the structure to simplify the representation of the HEA model. The HEA models were constructed on fcc phase and the random distribution of the five elements. Random construction is just random initial structure, and the atomic distribution of the alloy is random. Therefore, through randomly building the model, a stable HEA structure model is obtained by MD. It is a common method to sue up a model for subsequent calculations

(4) No details on RuO₂. Powder ? RuO₂/C ? What is the weight %?

It is unfair to compare with the powder samples (the performance is of course bad).

Reply: We are thankful to the reviewer for the careful review. The “RuO₂” denotes the RuO₂ powder. To eliminate the difference in electrode structure, the FeCoNiMnRu/CNFs, Pt/C, RuO₂, and IrO₂ were all ground to fine powder and dropped onto GCE. For the fair comparison of mass activities, the “weight % (ω)” of RuO₂ and IrO₂ represent the mass fraction of noble metal elements in the corresponding oxides. The ω for FeCoNiMnRu/CNFs, Pt/C, RuO₂, and IrO₂ are 4.383, 20.000, 75.958, and 85.727 wt.%, respectively. Noted that the mass fraction of noble metal in FeCoNiMnRu/CNFs is far lower than that in the state-of-art electrocatalysts.

Takashi Hibino et al., (J. Electrochem. Soc., 2016, 163, A1420) have reported that “We conducted preliminary tests using these commercially available materials as cathode materials for rechargeable fuel-cell batteries and found that RuO₂ showed a higher cathode performance than IrO₂.” The HRTEM image of nanocrystalline RuO₂ with crystallite size 1.1 nm was shown in Figure S6.

Figure S6 TEM images of commercial RuO₂. [Takashi Hibino et al., J. Electrochem. Soc., 2016, 163, A1420]

(5) IrO₂/C is more common benchmark for OER than RuO₂ powders.

Reply: We are thankful to the reviewer for the kind suggestion. The OER activities of IrO₂ powder supported on GCE are measured and displayed in Supplementary Fig. 23. Both of the IrO₂ and RuO₂ were usually used as the benchmark for OER (Zhigang Zou, et al., *Nat Synth* 2022, 1, 138; Yongwen Tan, et al., *ACS Energy Lett.* 2020, 5, 192;

Shaojun Dong, *et al.*, *Nano Energy* 2020, 68, 104296; Qianwang Chen, *et al.*, *ACS Catal.* 2020, 10, 1152; Ji Yang, *et al.*, *Sci Rep* 2016, 6, 38429; Daniel Böhm, *et al.*, *Adv. Funct. Mater.* 2020, 30, 1906670).

Reviewer #2 (Remarks to the Author):

The authors have added a substantial amount of extra data and analysis in response to concerns from the reviewers. From my side, adding reference samples and reanalyzing the X-ray spectroscopy data, adding a Bader analysis comparison to support their choice of electronegative scale in rationalizing their results, and adding randomized /mixed high-entropy structures are important additions. The authors have also added d-band calculations, electrochemical surface area (ECSA) normalized catalytic currents, as well as other improvements and reference experiments. In summary, my view is that the manuscript is now much improved and warrants publication in Nature communications after some minor corrections.

Reply: We are thankful to the reviewer for the kind recommendation and careful review.

1. Ref 28 is still not the correct reference for the stated Pauling electronegativities, the study do not mention Ru, and is instead performing Bader analysis to prove their charge transfer towards Co (in their Ru free system). I advise the authors to instead base their results on the Delta G calculations and the Bader analysis instead as of now stating "...due to the higher electronegativity of Ru ...". The authors are of course free to state that this is in line with the simpler notion of Pauling electronegativity, but should then be references to a study including these.

Reply: We are thankful to the reviewer for the kind recommendation and careful review. Actually, the ref 18 at "Two satellite peaks (marked as "Sat.") appear at binding energies (BEs) of 648.5 and 659.7 eV²⁸" is cited to support the XPS data. The ref 18 at "...due to the higher electronegativity of Ru (2.20) than those of Ni (1.91), Co (1.88), Fe (1.83), and Mn (1.55)²⁸" is corrected to be ref 31 (Xiang Xu, *et al.*, *ACS Nano* 2020, 14, 17704–17712) and the following references have also been modified to correct order.

In ref 31, Xiang Xu, *et al.* extensively explored the electronic structures in HEA-NPs supported on carbon by XPS and first-principles electronic structure calculations, which found that the electron density redistribution in the NPs occurs from less electronegative elements to more electronegative ones. Interesting, the authors performed a confirmatory experiment by synthesized an HEA-NP with "the more electronegative element Ru" to examine the behavior of the Cu (Pauling electronegativity: 1.90, less that of Ru) core level, and "the direction of the core energy level shift of Cu in FeCoNiCuRu@CNF is in the positive direction", in line with trend observed in our study. Therefore, Pauling electronegative values may be more suitable in high entropy alloys.

2. Some minor formulations and spellings, e.g. "... there are more electrons can be

observed" (line 500), Mu instead of Mn (line 502), "...display the lowest values than ..." instead of "...display lower values than ..." (line 540-541), and " ...as represent sample to show ..." instead of "as a representative sample to show..." (line 547), but these and others can also be handled in a proof-process.

Reply: We are thankful to the reviewer for the careful review. The mentioned mistakes have all been corrected, and we have carefully checked the whole manuscript to eliminate mistakes in formulations and spellings.

Reviewer #3 (Remarks to the Author):

In this manuscript, the authors reported a study on the functionality of HEA as an electrocatalyst for water splitting. The reviewer's concerns/comments regarding the electrochemical performance and stability, oxidation states, local electronic structures, and thermodynamically driven phase transition depending on the temperature of annealing treatment of HEA were properly addressed in the revised manuscript. Therefore, it is considered that the manuscript is now ready for publication in Nature Communications.

Reply: We are thankful to the reviewer for the kind recommendation and careful review.

REVIEWERS' COMMENTS

Reviewer #1 (Remarks to the Author):

The paper was properly revised.

The quality definitely reaches that of Nat. Comm., I believe.

I recommend the acceptance.

Reviewer #2 (Remarks to the Author):

The authors have carefully revised the manuscript, and is in my view now ready for publication in Nature Communications.

Response to the comments

REVIEWERS' COMMENTS

Reviewer #1 (Remarks to the Author):

The paper was properly revised.

The quality definitely reaches that of Nat. Comm., I believe.

I recommend the acceptance.

Reply: We are thankful to the reviewer for the kind suggestion.

Reviewer #2 (Remarks to the Author):

The authors have carefully revised the manuscript, and is in my view now ready for publication in Nature Communications.

Reply: We are thankful to the reviewer for the kind suggestion.